# Directive giant upconversion by supercritical bound states in the continuum

Chiara Schiattarella[1], Silvia Romano[1], Luigi Sirleto[1], Vito Mocella[1], Ivo Rendina[2], Vittorino Lanzio[3], Fabrizio Riminucci[3], Adam Schwartzberg[3], Stefano Cabrini[3], Jiaye Chen[4], Liangliang Liang[4], Xiaogang Liu[4,5,6 ✉] & Gianluigi Zito[1 ✉]

Photonic bound states in the continuum (BICs), embedded in the spectrum of free-space waves[1,2] with diverging radiative quality factor, are topologically non-trivial dark modes in open-cavity resonators that have enabled important advances in photonics[3,4]. However, it is particularly challenging to achieve maximum near-field enhancement, as this requires matching radiative and non-radiative losses. Here we propose the concept of supercritical coupling, drawing inspiration from electromagnetically induced transparency in near-field coupled resonances close to the Friedrich–Wintgen condition[2]. Supercritical coupling occurs when the near-field coupling between dark and bright modes compensates for the negligible direct far-field coupling with the dark mode. This enables a quasi-BIC field to reach maximum enhancement imposed by non-radiative loss, even when the radiative quality factor is divergent. Our experimental design consists of a photonic-crystal nanoslab covered with upconversion nanoparticles. Near-field coupling is finely tuned at the nanostructure edge, in which a coherent upconversion luminescence enhanced by eight orders of magnitude is observed. The emission shows negligible divergence, narrow width at the microscale and controllable directivity through input focusing and polarization. This approach is relevant to various physical processes, with potential applications for light-source development, energy harvesting and photochemical catalysis.

Bound states in the continuum (BICs) have been investigated in photonic-crystal nanoslabs (PCNSs) and metasurfaces[5,6], single-particle resonators[7] and hybrid systems[8], with applications in sensing[9,10], lasing[11,12] and nonlinear optics[13,14]. However, as with all other resonators, the achievable cavity enhancement is fundamentally limited by cavity losses and input coupling. The single-resonance intensity enhancement between the local field $E_{loc}$ and the input field $E_i$ can be written as:

$$G = \frac{|E_{loc}|^2}{|E_i|^2} \approx \kappa_i^2 \frac{Q^2}{V_{eff}} = \frac{Q^2}{Q_r V_{eff}} = \frac{Q_r^2 Q_a^2}{Q_r (Q_r + Q_a)^2 V_{eff}}. \quad (1)$$

In this equation, $\kappa_i$ is the input coupling coefficient. Because it depends on the radiation channel, the coupling becomes $\kappa_i = \sqrt{2\gamma_r}$, with $\gamma_r = \omega/(2Q_r)$ radiative loss[15], where $\omega$ is the angular frequency and $Q_r$ is the radiative quality factor (Methods section 'TCMT: critical coupling for an isolated mode'). The intrinsic quality factor $Q = (1/Q_a + 1/Q_r)^{-1}$ combines radiation channel loss ($1/Q_r$) and non-radiative loss ($1/Q_a$), which encompasses all dissipation channels (finite sizes, imperfections and material absorption), with $Q_a$ being the non-radiative quality factor. The normalized effective mode volume $V_{eff}$ measures the local field superposition with the material of interest. Although $Q$ measures the storable energy of the resonator, $Q_r$ also defines the coupling between external drive and resonator, enabling optical energy pumping. When the radiative loss becomes negligible (diverging $Q_r$), the storable energy becomes limited only by unavoidable non-radiative losses. However, when $Q_r \to \infty$ (ideal BICs and other dark states), no far-field light would couple with the resonator, resulting in $G \to 0$. A trade-off maximizes the cavity enhancement in equation (1) at the critical coupling condition, in which radiative coupling balances non-radiative dissipation ($Q_r = Q_a$)[15,16] (Methods section 'TCMT: critical coupling for an isolated mode'). Coupling strategies for BICs at present are mainly based on perturbing the ideal geometry and constructing quasi-BIC resonators with broken symmetry[8,13] and finite $Q_r$. However, real structures exhibit greatly reduced $Q_r$ on the order of $10^2$ (refs. 5,9), and quantifying balance with non-radiative losses is challenging.

In this work, we show that a Friedrich–Wintgen (FW)[2,17–19] quasi-BIC can be achieved through supercritical coupling, which can be related to coupled-resonance-induced transparency[20–22]. This condition overcomes the negligible direct far-field coupling with the quasi-BIC and restores the maximum level of enhancement imposed by the non-radiative loss. This occurs even though the quasi-BIC has a divergent and mismatched radiative quality factor. By reaching supercritical coupling at the edge point in which a tailored nanostructure meets the surrounding unpatterned slab (Fig. 1a), a giant enhancement of

[1]Institute of Applied Sciences and Intelligent Systems, National Research Council, Naples, Italy. [2]Institute of Applied Sciences and Intelligent Systems, National Research Council, Pozzuoli, Italy. [3]Molecular Foundry, Lawrence Berkeley National Laboratory, Berkeley, CA, USA. [4]Department of Chemistry, National University of Singapore, Singapore, Singapore. [5]Institute of Materials Research and Engineering, Agency for Science, Technology and Research (A*STAR), Singapore, Singapore. [6]Centre for Functional Materials, National University of Singapore Suzhou Research Institute, Suzhou, China. ✉e-mail: chmlx@nus.edu.sg; gianluigi.zito@cnr.it

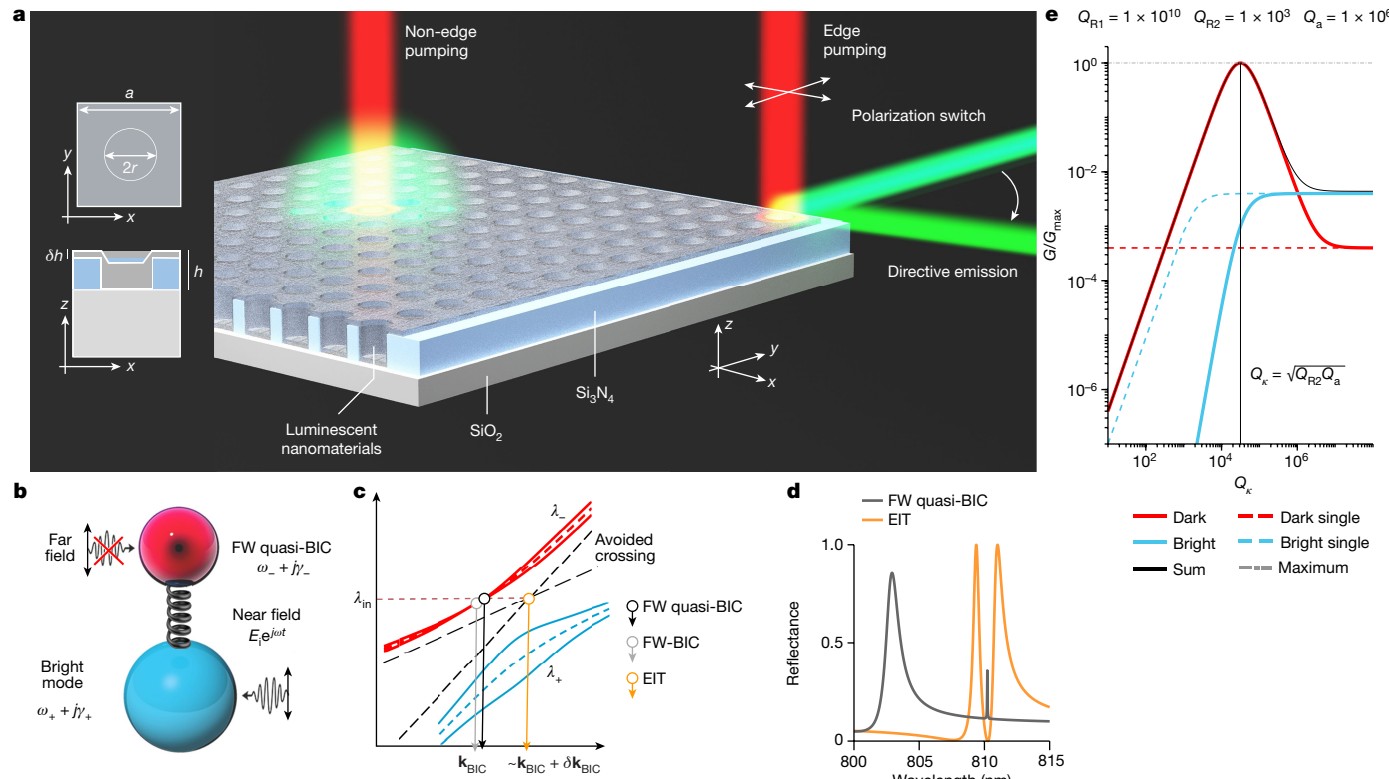

**Fig. 1 | Principle of supercritical coupling and directive upconversion emission. a**, Layout of the PCNS unit cell geometry and collimated upconversion generated through supercritical coupling tuned at edge pumping. **b**, Schematic depicting internally coupled resonances and their external coupling to the far field. **c**, Avoided crossing between waves $\lambda_-$ and $\lambda_+$ (dashed lines, peaks; solid lines, linewidths) owing to intercoupling $\kappa_{12}$: $\lambda_-$ becomes an FW quasi-BIC and can evolve with momentum to the dark mode of the EIT window at the crossing point. The laser wavelength $\lambda_{in}$ is set to $\lambda_-$. **d**, The energy–momentum dispersion of the system is tailored to achieve FW quasi-BIC and EIT with minimal phase mismatch $\delta\mathbf{k}$ at $\lambda_{in}$. **e**, Normalized intensity enhancement $G/G_{max}$ of the bright and dark modes having representative highly mismatched parameters, $Q_{R2} = 10^3 \ll Q_a = 10^6 \ll Q_{R1} = 10^{10}$. The tuning of $Q_\kappa$ enables the supercritical coupling condition: the far-field energy coupled with the bright mode (denoted by the solid cyan line) is diverted to the near-field drive exciting the high-$Q_{R1}$ FW quasi-BIC (solid red line) up to the maximum achievable level, $G_{max}$. The threshold of the single dark mode (dashed red line) is surpassed by several orders of magnitude. When $Q_\kappa \to \infty$ ($\kappa_{12} \to 0$), the coupled dark-mode intensity decreases to the uncoupled resonance threshold.

upconversion photoluminescence is experimentally demonstrated, which exceeds single-dark-resonator coupling by orders of magnitude. Furthermore, experimental results reveal that upconversion photons propagate in plane, forming a microscale coherent beam with a spatial width of less than 100 μm and a divergence of less than 0.07° over a centimetre distance. This combined with supercritical coupling leads to an enhancement of upconversion by eight orders of magnitude.

## Theoretical model

We designed a transparent holey PCNS covered with a conformal layer of upconversion nanoparticles (NPs)[23,24] (Fig. 1a). The system is described by the non-Hermitian Hamiltonian $\hat{H}_{\mathbf{k}} = \hat{\Omega}(\mathbf{k}) + j\hat{\Gamma}_r(\mathbf{k})$, which models transverse electric-like (TE-like) and transverse magnetic-like (TM-like) modes coupled to a single independent radiation channel, thus originally non-orthogonal[17,25]. In the energy–momentum space, these modes evolve and eventually approach an FW-BIC at a specific wavevector $\mathbf{k} = \mathbf{k}_{BIC}$ if the FW condition that diagonalizes $\hat{H}_{\mathbf{k}_{BIC}}$ is satisfied by the parameters (Methods section 'Open-resonator TCMT'). The initial modes 1 and 2 have radiative loss rates $\gamma_{r1}$ and $\gamma_{r2}$, respectively. The coupled final modes of wavelengths $\lambda_-$ and $\lambda_+$ split apart because of strong coupling (Fig. 1b,c). One of these waves becomes a perfect dark mode (ideal FW-BIC) with zero linewidth ($\gamma_- = 0$) and diverging lifetime ($\tau_{R1} = 1/\gamma_- = 2Q_{R1}/\omega_- = \infty$), at a specific wavevector $\mathbf{k}_{BIC}$ near the avoided crossing. The bright mode acquires all radiative losses with $\gamma_+ = \gamma_{r1} + \gamma_{r2}$, which provides a final mode with low $Q_{R2}$. At the asymptotic

FW condition, $\hat{\Omega}(\mathbf{k}_{BIC})$ and $\hat{\Gamma}_r(\mathbf{k}_{BIC})$ are simultaneously diagonalized, resulting in orthogonal modes. This is allowed by energy-conservation balance of input drive and system modes because the dark mode totally decouples from the radiation channel. However, for wavevectors close to but not at the FW-BIC, the perturbed FW quasi-BIC (with $\gamma_- \neq 0$) takes on non-zero coupling with the radiation channel ($\sqrt{2\gamma_-}$), thus the perturbed Hamiltonian $\hat{H}_{\mathbf{k} \simeq \mathbf{k}_{BIC}}$ must be represented with non-zero off-diagonal terms $\kappa_{12}$ in $\hat{\Omega}(\mathbf{k} \simeq \mathbf{k}_{BIC})$ to obey energy conservation[21,26]. When two modes are coupled with a single radiation channel, coupled-resonance-induced transparency can also take place. This is the analogue of electromagnetically induced transparency (EIT) in photonic/plasmonic systems and can provide exceptionally slow light and enhanced local optical field[27]. For suitable PCNS geometries, EIT and ideal FW-BIC may occur with small phase mismatch ($\mathbf{k}_{EIT} = \mathbf{k}_{BIC} + \delta\mathbf{k}$). Essentially, the quasi-BIC may evolve into the transparency frequency of the EIT process (Fig. 1d). Temporal coupled-mode theory (TCMT) and final mode amplitudes for the dark (FW quasi-BIC) and bright partners are explicitly provided as equations (36) and (37) in Methods section 'Supercritical coupling'.

To clarify the consequences of resonance coupling, the intensity enhancement $G$ of the final dark and bright modes, normalized to the maximum enhancement $G_{max} = Q_a/V_{eff}$, is plotted in Fig. 1e for highly unmatched quality factors, representing an unsatisfactory scenario for single resonances. Having defined the near-field coupling quality factor as $Q_\kappa = \omega/(2\kappa_{12}) = \tau_\kappa\omega/2$, an optimum condition that we term 'supercritical coupling' can be reached when $\overline{Q}_\kappa = \sqrt{Q_{R2}Q_a}$, which avoids

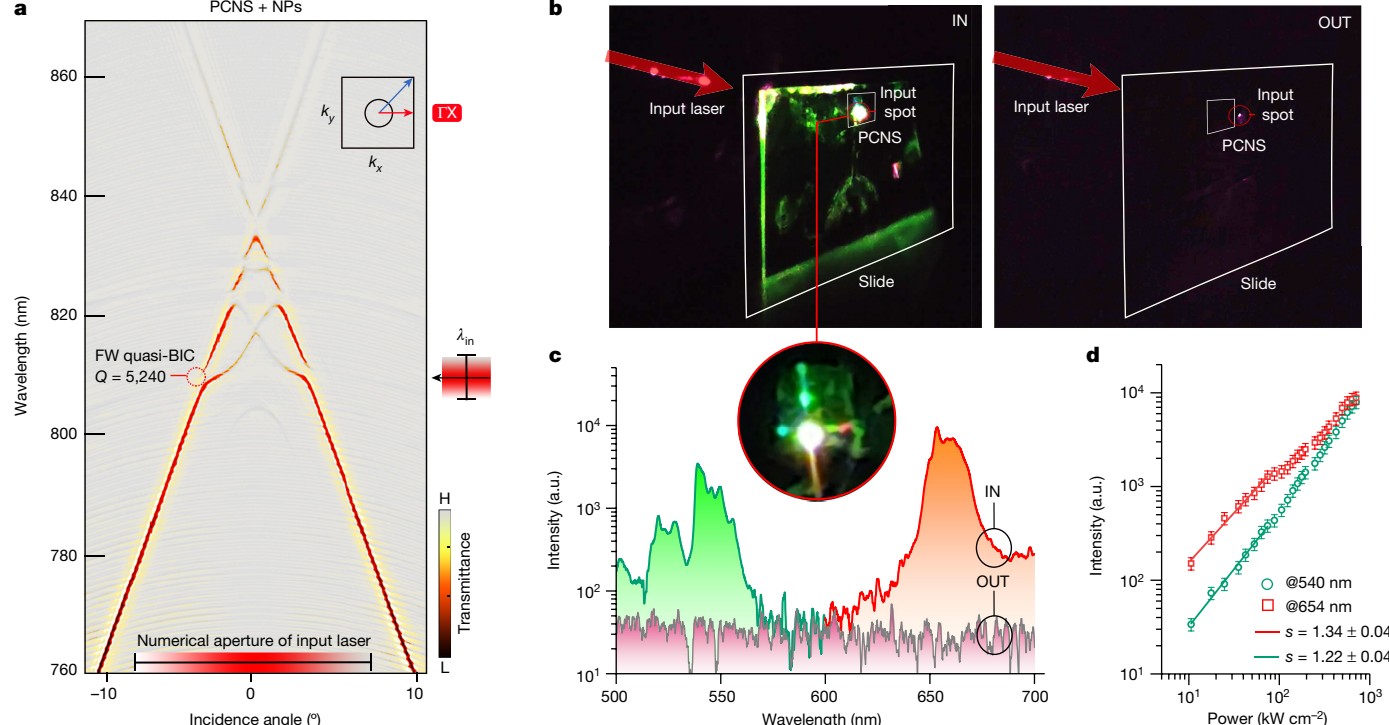

**Fig. 2 | Experimental characterization of forward upconversion radiation.**
**a**, Experimental TE band diagram of the PCNS resonator, highlighting the high-$Q$ FW quasi-BIC. The pulsed Ti:Sa oscillator ($\lambda_{in}$ = 810 nm, linewidth) is tuned to the FW quasi-BIC and focused to a 6-μm spot on the PCNS for upconversion pumping. H, high; L, low. **b**, Experimental upconversion emission, with the left image focusing inside (IN) the PCNS area and the right image focusing outside (OUT). The red circle indicates the pump spot and the inset provides a magnified view of the PCNS region. **c**, Corresponding upconversion spectrum along the forward direction inside (green and red curves) and outside the PCNS area (grey curve). **d**, Upconversion intensity excited inside the PCNS versus input power (error bars: intensity standard deviation). a.u., arbitrary units.

the bottleneck of the narrow input radiation channel ($Q_{R1} = 10^{10}$) by diverting the input energy of the low-$Q_{R2}$ mode to the dark mode. The maximum level of local field enhancement, always in the highest $Q$-factor mode, can be reached even under extremely unfavourable conditions for single isolated modes. Extended Data Fig. 1 shows the off-resonance behaviour and the parameters for which the coupled dark mode has an advantage. At the EIT transparency frequency (dark mode), the field is enhanced because of slow light, possibly reaching a maximal level as explained by supercritical coupling of the dark mode (Methods section 'Supercritical coupling').

The TCMT model is validated by rigorous-coupled-wave analysis (RCWA) (Methods section 'RCWA validation'), including energy–momentum dispersion, full vector modes and their symmetry inversion, FW-BIC formation, EIT condition at the nearby avoided crossing with incidence-angle separation of approximately 0.5° (Extended Data Figs. 2–5, respectively) and tuning of $\kappa_{12}$ towards supercritical coupling (Extended Data Fig. 5). Further details are in Supplementary Figs. 1–4.

## Experimental demonstration

Upconversion nanocrystals convert infrared to visible light by cascade photon absorption through long-lived intermediate energy states[28–30] and have found applications in display technology and lasers[31–33], energy conversion[34], imaging probes[35,36] and metasurface resonators[37,38]. Our PCNSs consist of square holey patterns of 1 mm² in a Si₃N₄ slab on SiO₂ substrate, coated for an area of 1.25 mm² with conformal claddings of either NaErF₄@NaYF₄ or NaGdF₄:Nd/Yb(40/5%)@NaGdF₄:Yb/Tm(49/1%)@NaGdF₄:Eu(15%) nanocrystals (Extended Data Fig. 6). The NPs fill the holes and homogeneously cover the slab (Supplementary Figs. 5 and 6).

Figure 2a shows the measured **ΓX** dispersion-band diagram (Methods and Supplementary Fig. 7). The FW quasi-BIC has intrinsic quality factor ($Q$) of 5,240 and is spectrally overlapped with the NP absorption band. Hybridization between the TM-like and TE-like bands is observed in Supplementary Figs. 8 and 9. The experimental set-up for upconversion measurements is described in Supplementary Fig. 10. Figure 2b compares the upconversion emission generated inside the PCNS with the signal produced by the same number of NPs outside the PCNS (Methods). The forward emission spectra excited by focusing the pump inside and outside the PCNS are shown in Fig. 2c. The intensity scaling is shown in Fig. 2d. Because $n$-photon scaling only occurs for small absorption cross-section (proportional to the local field), the nearly unitary exponent $s$ points out single-photon promotion to excited states[39,40] because of strongly enhanced local field. To estimate the upconversion enhancement factor, which was EF$_{exp}$ ≈ 300, the intensity was compared with a bulk sample of NPs (Methods and Supplementary Fig. 11). However, most of the visible emission propagated in the transverse plane rather than the forward direction.

The emission greatly increases when the input beam crosses the PCNS edge. Figure 3a shows a continuous transformation, measured with 6-μm spot resolution (Methods), causing inner bands to merge into boundary bands with progressively closing gap. The increase in lateral emission correlates with decreasing gap ($\kappa_{12}$), indicating increasing values of $Q_\kappa$. The beam becomes more visible as it crosses a silicone layer (Fig. 3b and Supplementary Video 1). The bands become nearly overlapped at −2.9°, at the FW quasi-BIC wavelength (−3.4°, dashed line) with a momentum mismatch of 17% ($\Delta\theta \simeq 0.5°$). The gap decreases from 3.5 to <0.7 nm, consistent with the calculated coupling based on RCWA modes, reaching $\kappa_{12} = 0.5(\Delta\lambda/\lambda)\omega_{in} \simeq 4.3 \times 10^{-4}\omega_{in}$, close to the estimated supercritical coupling value $\overline{\kappa}_{12} = 0.5\omega_{in}/\overline{Q}_\kappa \simeq 4.7 \times 10^{-4}\omega_{in}$, as described below.

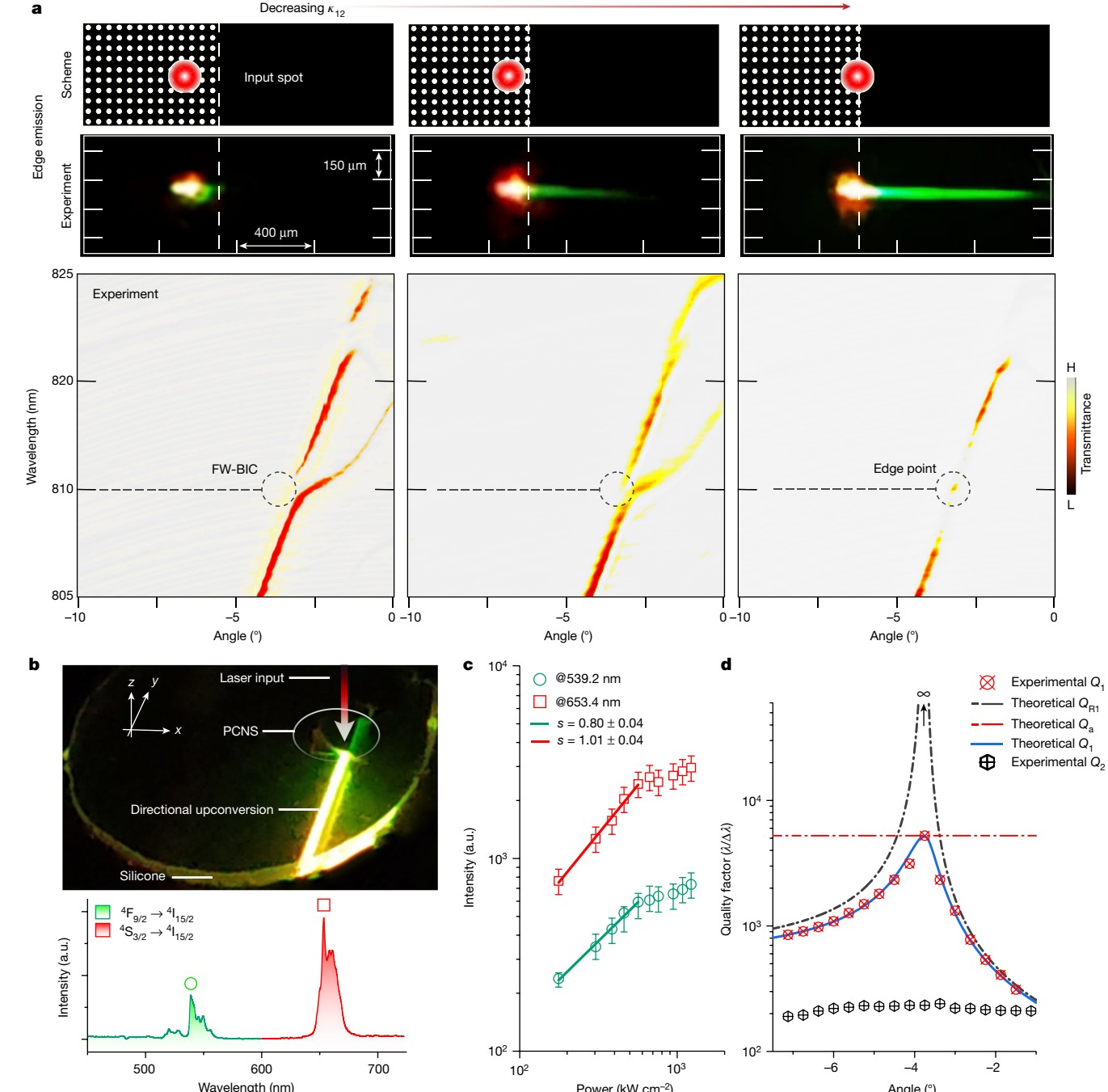

**Fig. 3 | Experimental characterization of in-plane upconversion radiation.**
**a**, Top: the experimental upconversion emission excited by the pulsed laser
focused at a 6-µm spot, $\lambda_{in} = 810$ nm. The edge emission reaches its maximum
when crossing the PCNS boundary. A filter is placed on the camera to observe
only the green emission from core–shell NPs. Bottom: the corresponding TE
band diagrams, evolving with input position and measured with white laser
probe equally focused to a 6-µm spot. This demonstrates the correlation
between the enhanced edge emission and the band-structure modification,
with modes 1 and 2 overlapping as $\kappa_{12}$ decreases (Supplementary Fig. 2).

**b**, Photograph of the visible edge emission taken without a filter, along with
the corresponding photoluminescence spectrum. **c**, Corresponding intensity
scaling: saturation is readily achieved at relatively low input power (error bar:
intensity standard deviation). **d**, Total Q factors of the two modes, $Q_1$ and $Q_2$,
extracted from the transmittance in **a**. $Q_{R1}$ is numerically calculated and predicts
the experimental $Q_1$, assuming a constant $Q_a$. $Q_2$ is mainly influenced by radiative
loss, resulting in $Q_2 \simeq Q_{R2}$ (error bars: linewidth fit standard deviation, within
experimental point size).

The relative intensity, statistically sampled, for all notable upconver-
sion wavelengths is reported in Supplementary Fig. 12. At low incident
power, the scaling $s$ was found to be about 1.0 (Fig. 3c). When corrected
and normalized to the isotropic bulk emission, the estimated enhance-
ment was $EF_{exp} = 3.6 \times 10^4$ (Methods). This value can be compared with
the enhancement factor $EF_{th}$ expected for the single-resonance

equation (1) and with supercritical coupling. The experimental intrinsic
$Q_1$ in the FW quasi-BIC dispersion curve, and calculated $Q_{Ri}$ compensat-
ing for the flat $Q_a$, are shown in Fig. 3d. Assuming a power scaling $s$
in the range (0.8, 1.2), the more favourable enhancement factor for
the single-resonance model can be determined by integrating $Q_1$ over
the excitation angles, giving $EF_{th} = \kappa_o G_{av}^s \simeq (1.1 \times 10^3)^s \in (0.3 \times 10^3, 4.5 \times 10^3)$

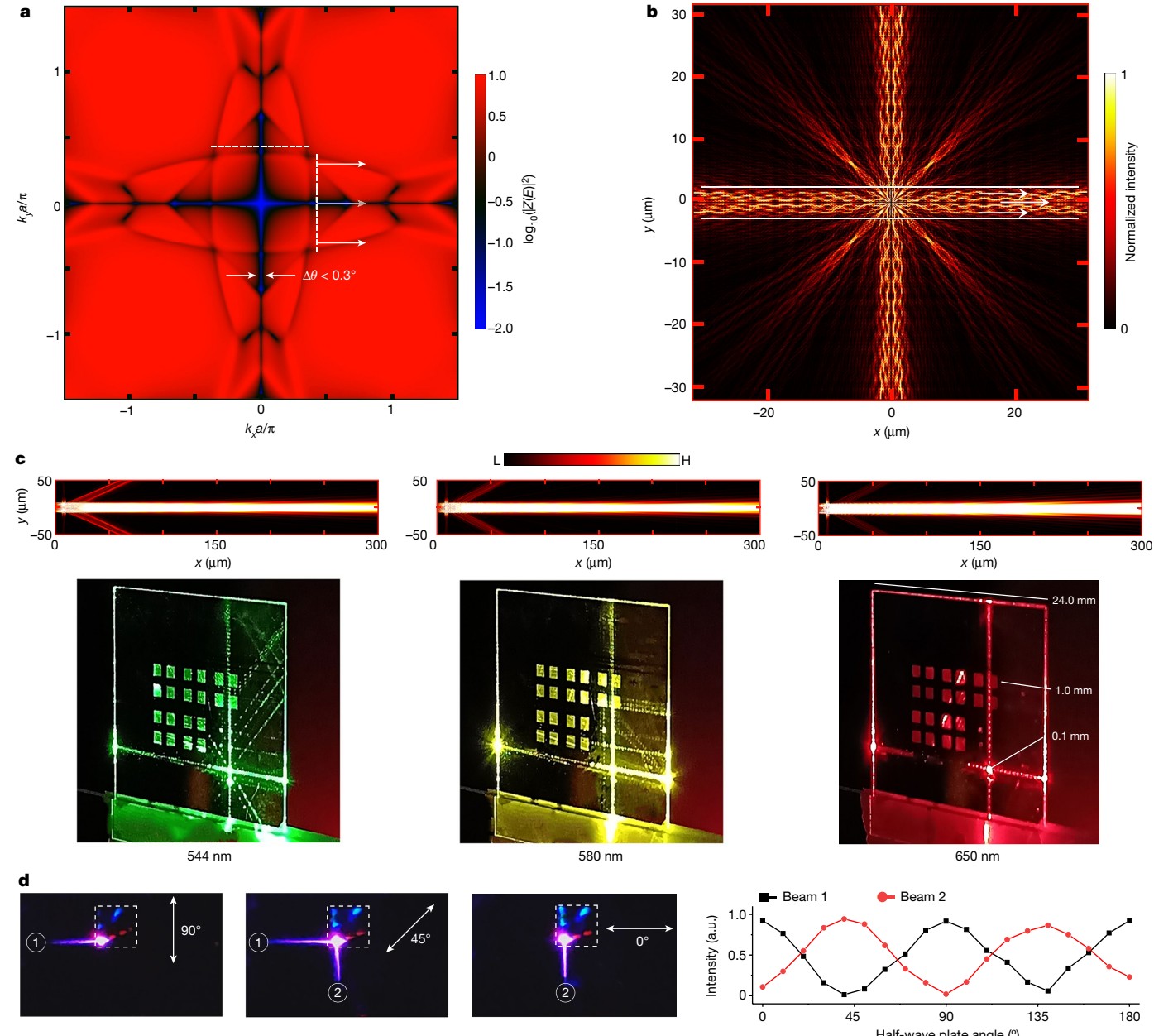

**Fig. 4 | Directivity and polarization-based switching. a**, Isofrequency far-field intensity map in the momentum space, revealing the presence of non-trivial vanishing strips. **b**, Near-field intensity map showing self-collimation along the symmetry axes. **c**, Simulated (top) and experimental (bottom) collimated emission in the visible range at 544 nm, 580 nm and 650 nm over the centimetre scale from a sample of side length 100 μm. On a glass slide, there is an array of 24 photonic-crystal slabs with variable side lengths of 1 mm, 300 μm and 100 μm. The input light is focused to a spot of 10 μm inside a patterned area of 100 μm.

Some cross-talk beams are generated by the propagation of the main beams from the illuminated area when they cross surrounding patterned areas. **d**, Snapshots of controlled light propagation near the edge (left), demonstrating that both horizontal and vertical side beams can be generated at the corner of the photonic-crystal slab, indicated in the dashed box (side length 1 mm): the intensity of the output is determined by the orientation of the input polarization, which is varied with the orientation of the half-wave plate axis (right).

(Methods). The extraction coefficient $\kappa_o$ is approximately 1 at upconversion wavelengths[41]. Even in this case, the single-resonance enhancement underestimates the experimental $EF_{exp} = 3.6 \times 10^4$ by at least one order of magnitude. By contrast, with the coupled-resonance model, the dark mode closely approaches the maximum field for a certain finite range of $\kappa_{12}$ and wavevectors (Methods section 'Supercritical coupling'). With nearly optimal $\overline{Q}_K = \sqrt{Q_{R2}Q_a}$, in which $Q_{R2} \simeq Q_2 = 213$ and $Q_a \simeq 5,240$, the upconversion enhancement factor can reach the value $\kappa_o (G_{max})^s = (6.0 \times 10^3)^s \in (0.1 \times 10^4, 3.4 \times 10^4)$, which is in good agreement with the experimental value ($EF_{exp} = 3.6 \times 10^4$).

## Self-collimation and radiance enhancement

The directive emission, normal to the PCNS edge, can be translated with continuity by correspondingly translating the input beam. This emission propagates for several millimetres while preserving a collimated width <100 μm. Diffraction-free guiding with BICs has been observed in the microwave range[42] and associated with the phenomenon of self-collimation inside the structured waveguide owing to flat-band dispersion in the momentum space[43,44]. In our experiments, however, not only does the propagation start from the

PCNS edge and continue in the slab but the negligible divergence is observed in the upconversion, spectrally far from the FW quasi-BIC. Therefore, finite-difference-time-domain (FDTD) simulations were performed to investigate this complex scenario computing the isofrequency map using the $Z$ transform of the local field (Methods). This approach was preliminarily validated testing the case of conventional flat-band self-collimation[44] (Supplementary Fig. 13). By contrast, the isofrequency map of our system shows non-trivial vanishing strips along the symmetry axes of the geometry, intersecting squared flat bands (Fig. 4a). The associated real-space intensity map also shows self-collimation characteristics. However, in this case, it is the low coupling with the far field along the strips that leads to negligible divergence (Fig. 4b). This was also confirmed experimentally in a PCNS geometry scaled to have visible FW quasi-BIC at 532 nm, confirming self-collimation over a centimetre distance (Extended Data Fig. 7).

Second, an array of dipole point sources covering an area consistent with the experimental pump spot was placed at the boundary between the finite PCNS and the homogeneous waveguide slab. The point sources collectively radiate in the slab with coherent phased-array emission, resulting in minimal divergence, as low as 0.02°, across a large part of the visible spectrum (Extended Data Fig. 8). This was experimentally verified and compared with simulations at representative wavelengths in Fig. 4c. Microscopy analysis of the beam revealed a high degree of spatial coherence, as evidenced by the visibility of the interference pattern (Extended Data Fig. 9a).

Experimental characterization in Extended Data Fig. 9b,c yielded a solid-angle divergence of $\Omega_1 \simeq 1.2 \times 10^{-3}$ srad. The radiance enhancement $R_{EF}$ given by enhanced directive emission, normalized to the isotropic bulk emission ($\Omega_{sphere} = 4\pi$), yields $R_{EF} = \frac{\Omega_{sphere}}{\Omega_1} \times EF_{exp} \simeq 3.8 \times 10^8$ (Methods), an extraordinary value resulting from the combination of supercritical field enhancement and directivity enhancement.

Moreover, by positioning the focused input beam and rotating the input polarization at the corner of the PCNS, it is possible to switch between vertically and horizontally emitted beams (Fig. 4d and Supplementary Video 2). The switch is determined by aligning the input polarization perpendicular to the $x$ wave (or $y$ wave) depicted in Fig. 4b, thereby selectively exciting only one of them (beam 1 or 2) or both when at 45° of inclination.

To investigate emission properties at a microscale level, the structure was reduced in size by factors of 3 and 10. This scaling down resulted in a manageable logarithmic slow variation of the output signal (Supplementary Fig. 14).

## Discussion

The mechanism of near-field coupling between FW quasi-BIC and bright mode can divert the input source from the bright to the dark mode, breaking the limits of single-dark-mode coupling with orders-of-magnitude improvement, a condition referred to as supercritical coupling. This phenomenon is also related to the EIT process, which can arise from similar coupling and aligns with it. However, the occurrence of a transparency window is not a requirement, although its proximity in momentum space can widen the wavevector span over which the field is enhanced.

The experimental proof of FW quasi-BIC supercritical coupling is provided using chemically and optically stable upconversion NPs, which enable several microscale addressable sources and lasers. The edge enhancement facilitates directive propagation of self-collimated photons with remarkable control. In contrast to conventional self-collimation, herein beam collimation is not only in the nanostructured slab, because the BIC acts as a filter in the reciprocal space, favouring high directivity and spatial coherence of the outcoupled wave despite its microscale width. Moreover, high collimation extends beyond just the pumping mode, covering the broad spectrum of upconversion because of coherent phased-array emission. The resulting photoluminescence is enhanced by more than eight orders of magnitude, representing one of the highest values achieved with a dielectric resonator[39].

The topological confinement at the BIC expands the model of disorder-immune devices based on light topological phases used for lasing action[45,46]. In the present state, the upconversion emission is broadband and not peaked only at single lasing wavelengths. Thus, by combining several types of NP, the output spectrum can cover a continuum range in the visible with spatial coherent emission. Incorporating enhancement at pump and emission frequencies, this system can offer new capabilities for on-chip microscale light control, providing important possibilities in many nanophotonic processes based on high-$Q$ resonators, such as for light-source technology, energy harvesting, photochemical catalysis, sensing and quantum information.

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

## Methods

### Theory

In the section 'TCMT: critical coupling for an isolated mode', the local field enhancement at critical coupling for an isolated resonant mode is demonstrated. In the section 'Open-resonator TCMT', the non-Hermitian Hamiltonian formalism of TCMT for FW-BIC formation is used. It will be shown that, as the asymptotic condition of BIC cannot be ideally reached, the FW quasi-BIC originates from non-orthogonal modes. This understanding will then be used in the section 'Supercritical coupling' to evaluate the coupling between the dark FW quasi-BIC and the bright leaky partner, demonstrating the analogy to EIT and the equation for supercritical local field enhancement. In the section 'RCWA validation', we validate the TCMT results using RCWA.

**TCMT: critical coupling for an isolated mode.** The basic equation describing the evolution of the mode amplitude $A_1$ (oscillator 1) in a resonating system with a characteristic angular frequency $\omega_1 = 2\pi c/\lambda_1$, is

$$\frac{dA_1}{dt} = j\omega_1 A_1 - \left(\frac{1}{\tau_a} + \frac{1}{\tau_r}\right)A_1, \tag{2}$$

in which energy can be lost through absorption (or other additive non-radiative channels, such as scattering by dielectric fluctuations or in-plane leakage) with decay rate $\gamma_a = 1/\tau_a$, as well as through direct far-field coupling with external radiation in the outer space with a decay rate $\gamma_r = 1/\tau_r$. The amplitude is normalized such that $|A_1|^2$ represents the energy of the mode[15]. When adding the driving field of power $|s_+|^2$ and monochromatic time dependence $\exp(j\omega_{in}t)$, associated with the external excitation and coupled with the resonator with coefficient $\kappa_i$, the equation becomes

$$\frac{dA_1}{dt} = j\omega_1 A_1 - \left(\frac{1}{\tau_a} + \frac{1}{\tau_r}\right)A_1 + \kappa_i s_+. \tag{3}$$

The solution is

$$A_1(\omega_{in}) = \frac{\kappa_i s_+}{j(\omega_{in} - \omega_1) + \left(\frac{1}{\tau_a} + \frac{1}{\tau_r}\right)}. \tag{4}$$

It is possible to demonstrate that the input power coupling must be related to the radiation decay as $\kappa_i = \sqrt{2/\tau_r}$ by invoking energy conservation and time-reversal symmetry of Maxwell's equations. On resonance, that is, when the input frequency $2\pi c/\lambda_{in} = \omega_{in}$ is set at the peak $\omega_1$, it follows that

$$A_1(\omega_1) = \frac{\sqrt{2/\tau_r}\, s_+}{1/\tau_a + 1/\tau_r}. \tag{5}$$

Now let us consider that the quality factor $Q$ of a resonator is defined as the ratio between the stored ($W$) and the lost energy fractions. Indeed, for the absorption-related power loss $P_{abs}$ (or, more generally, all non-radiative losses) and the radiation loss $P_{rad}$, the following holds true

$$\frac{1}{Q} = \frac{1}{Q_a} + \frac{1}{Q_r} = \frac{P_{abs}}{\omega_1 W} + \frac{P_{rad}}{\omega_1 W} = \frac{2}{\omega_1}\left(\frac{1}{\tau_a} + \frac{1}{\tau_r}\right) = \frac{2}{\omega_1}(\gamma_a + \gamma_r). \tag{6}$$

The driving field has amplitude $E_i = s_+/A_c$, in which $A_c$ is a normalized cross-section, which we define as $A_c = 1$, for simplicity. The local field of the resonant mode has amplitude given by $E_{loc} = A_1/\sqrt{V_{eff}}$, with $V_{eff}$ the normalized effective mode volume. Thus, from equation (5), it follows that the local field enhancement $G$ is given by

$$G = \frac{|E_{loc}|^2}{|E_i|^2} \simeq \frac{Q^2}{Q_r V_{eff}} = \frac{Q_a^2 Q_r^2}{Q_r(Q_r + Q_a)^2 V_{eff}}, \tag{7}$$

and depends on the ratio between the total quality factor $Q = 1/(1/Q_a + 1/Q_r) = Q_a Q_r/(Q_a + Q_r)$ and the radiation quality factor $Q_r$. Clearly, if $Q_r \gg Q_a$ as for the ideal BIC with $Q_r \to \infty$, then asymptotically $G \approx Q_a^2/Q_r \to 0$. The maximum enhancement $G_{cr}$ is reached when $Q_r = Q_a$, at the critical coupling condition, for which

$$G_{cr} \approx \frac{Q_a^4}{Q_a^3 V_{eff}} = \frac{Q_a}{V_{eff}}. \tag{8}$$

The above result has been applied to BICs in several papers[13] and its origin dates to the general theory of optical and electrical resonators discussed in textbooks[15]. Supposing a nearly ideal resonator with $Q_r = Q_a$, the maximum field enhancement would reach the physical capacity limit imposed by the unavoidable system losses represented by $Q_a$. In dielectric resonators sustaining quasi-BICs, the critical coupling point can be approached by breaking the in-plane symmetry of the system to tune the radiation quality factor that scales quadratically with the asymmetry parameter[5], which requires precise nanostructure engineering and knowledge of the system losses.

**Open-resonator TCMT.** The theory of FW-BIC formation owing to coupling of two leaky modes has been reviewed in ref. 47. The demonstration based on the non-Hermitian Hamiltonian of temporal coupled modes can be found in recent papers[19]. The formation of FW-BICs has gained attention particularly in the context of photonic-crystal slabs with vertical asymmetry, in which TM-like and TE-like modes couple and interfere[48,49]. However, it is worth noting that the existence of non-radiating modes arising from the interference of vector TE-like and TM-like eigenmodes was first discussed in ref. 17. It was found that, in 2D holey textured slabs, TE and TM modes can couple at virtually any point in the first Brillouin zone, leading to anticrossing of their dispersion and formation of a mode with zero imaginary part of its eigenfrequency, known as an FW-BIC[2]. In this study, a photonic-crystal slab placed over a dielectric waveguide substrate with air cladding was considered, breaking vertical symmetry and favouring the coupling of vector TE-like and TM-like modes. The same system was used in our previous work[10], in which we experimentally observed and applied the FW quasi-BIC, and it is also used in the present study.

To develop what we term the 'supercritical enhancement equation', we start from the non-Hermitian Hamiltonian of coupled waves[25,48]. By generalizing equation (3), the dynamic equations for resonance amplitudes can be written in the following form

$$\frac{d\mathbf{A}}{dt} = (j\hat{\Omega} - \hat{\Gamma}_r - \hat{\Gamma}_a)\mathbf{A} + \hat{K}_i^T \mathbf{s}^+, \tag{9}$$

$$\mathbf{s}^- = \hat{C}\mathbf{s}^+ + \hat{D}\mathbf{A}, \tag{10}$$

in which both $\hat{\Omega}$ and $\hat{\Gamma}_r$ matrices are Hermitian matrices representing the resonance frequencies and the radiation decay, respectively. On the other hand, $\hat{\Gamma}_a$ represents non-radiative losses and is initially set to zero $\hat{\Gamma}_a = 0$ to isolate the radiative rate associated with an ideal BIC. The resonant mode is excited by the incoming far-field waves $\mathbf{s}^+$ coupled to the resonator with coefficients denoted by $\hat{K}_i$. The outgoing waves $\mathbf{s}^-$ depend on the direct scattering channel $\hat{C}$ and the resonant modes $\mathbf{A}$ by means of the decay port coefficients in $\hat{D}$. Energy conservation and time-reversal symmetry imply that $\hat{K}_i = \hat{D}$ and that the coupling with the port is linked with radiation loss, implying that $\hat{D}^\dagger \hat{D} = 2\hat{\Gamma}_r$. These relationships determine the elements of $\hat{K}_i^T$ and imply that $\text{rank}(\hat{\Gamma}_r) = \text{rank}(\hat{D}^\dagger \hat{D}) = \text{rank}(\hat{D})$. Also, $\hat{D} = -\hat{C}\,\hat{D}^*$. Let us consider a system denoted by $\mathbf{A} = (A_1, A_2)^T$, in which $A_1$ and $A_2$ represent the amplitudes of

two modes with frequencies $\omega_1$ and $\omega_2$, respectively. These resonances have radiative lifetimes $\tau_{r1} = 1/\gamma_{r1}$ and $\tau_{r2} = 1/\gamma_{r2}$. Moreover, both resonances may experience absorption loss, characterized by $1/\tau_a = \gamma_a$. It is important to note that, for the specific case of the avoided crossing point, the absorption terms for both modes are the same, as we demonstrate below. Then, in general, $\gamma_{1,2} = \gamma_{r1,r2} + \gamma_a$, but for now, let's turn $\gamma_a = 0$.

Recall that the modes of the resonator are defined as the eigenmodes of the non-Hermitian Hamiltonian operator $\hat{H} = j\hat{\Omega} - \hat{\Gamma}_r$ (neglecting non-radiative loss). Only Hermitian matrices allow for a diagonal representation with orthogonal eigenvectors, whereas non-Hermitian matrices may have linearly dependent or linearly independent but non-orthogonal eigenvectors, or they may have orthogonal eigenvectors depending on specific properties such as parity–time symmetry. The Hamiltonian and its eigenvalues are functions of the in-plane momentum $\mathbf{k} = k_o(\sin\theta\cos\phi, \sin\theta\sin\phi)$. A previous study demonstrated that the eigenvectors of the non-Hermitian Hamiltonian are always non-orthogonal when the total number of independent decay ports is less than the number of optical modes and both modes are coupled to the decay ports[25]. The crucial concept here is that of independent decay ports, which are related to the sharing of the vertical symmetry of the modes. In the case of evolving TE-like and TM-like modes, the inversion of their character at the avoided crossing can occur at any point in energy–momentum space. We know that the eigenmodes of a matrix form an orthogonal basis if and only if $\hat{H}^\dagger \hat{H} = \hat{H}\hat{H}^\dagger$. Because both $\hat{\Omega}$ and $\hat{\Gamma}_r$ are Hermitian, this is equivalent to the relation $\hat{\Omega} \hat{\Gamma}_r = \hat{\Gamma}_r \hat{\Omega}$, which implies that $\hat{\Omega}$ and $\hat{\Gamma}_r$ can be simultaneously diagonalized. When considering two eigenmodes and a single independent radiation channel, in which $\mathrm{rank}(\hat{\Gamma}_r) = 1$, one of the orthogonal eigenmodes of the matrix will have a pure imaginary eigenvalue. This indicates that one of the two modes has an infinite lifetime (BIC) and does not couple to the decay port. As a non-zero coupling with the single decay port exists, the two eigenvectors in the resonator system will always be non-orthogonal[25]. Therefore, the modes are generally non-orthogonal if a single radiation channel is involved. However, they can satisfy the orthogonality condition at a specific point in momentum space. This point is referred to as an ideal FW-BIC point $\mathbf{k}_{BIC}$ when the Hamiltonian ($\hat{H} = j\hat{\Omega} - \hat{\Gamma}_r$, defined below) has a purely imaginary eigenvalue (or, equivalently, $\hat{\Omega} + j\hat{\Gamma}_r$ has a purely real eigenvalue). This allows for the simultaneous diagonalization of the Hermitian matrices $\hat{\Omega}$ and $\hat{\Gamma}_r$.

**FW condition.** The Hamiltonian of a two-waves-two-ports system is represented as:

$$\hat{H} = j\begin{pmatrix} \omega_1 & \kappa \\ \kappa & \omega_2 \end{pmatrix} - \begin{pmatrix} \gamma_{r1} & X \\ X^\star & \gamma_{r2} \end{pmatrix} = j\begin{pmatrix} \omega_1 + j\gamma_{r1} & \kappa + jX \\ \kappa + jX^\star & \omega_2 + j\gamma_{r2} \end{pmatrix} \equiv j\begin{pmatrix} \widetilde{\omega}_1 & \widetilde{\omega}_{12} \\ \widetilde{\omega}_{21} & \widetilde{\omega}_2 \end{pmatrix}, \quad (11)$$

in which $\kappa$ measures the near-field coupling and $X$ represents the coupling mediated by the continuum between the two closed, uncoupled channel resonances of frequencies $\omega_1$ and $\omega_2$. Following the calculation in refs. 19,25, $X$ can be expressed as

$$X = \sqrt{\gamma_{r1}\gamma_{r2}}\, e^{j\psi}, \tag{12}$$

in which the phase angle $\psi$ describes the relative phase of the coupling with the open channel and in general with the two ports (up and down). The eigenvalues of the two diagonal frequency and decay matrices of the Hamiltonian at the BIC point, defined by

$$\hat{H}^r(\mathbf{k}_{BIC}) = \hat{\Omega} + j\hat{\Gamma}_r = \begin{pmatrix} \widetilde{\omega}_+ & 0 \\ 0 & \widetilde{\omega}_- \end{pmatrix} + j\begin{pmatrix} \widetilde{\gamma}_+ & 0 \\ 0 & \widetilde{\gamma}_- \end{pmatrix}, \tag{13}$$

and associated with the collective modes $\widetilde{A}_+$, $\widetilde{A}_-$, are related to the uncoupled mode frequency and decay rates by

$$\widetilde{\omega}_\pm + j\widetilde{\gamma}_\pm = (\omega_1 + \omega_2)/2 + j(\gamma_{r1} + \gamma_{r2})/2 + \tag{14}$$

$$\pm \frac{1}{2}\sqrt{[(\omega_1 - \omega_2) + j(\gamma_{r1} - \gamma_{r2})]^2 + 4(\kappa + j\sqrt{\gamma_{r1}\gamma_{r2}}\, e^{j\psi})^2}. \tag{15}$$

This relation allows us to determine the asymptotic FW condition as a function of the uncoupled mode frequency, the decay rate and the coupling rate among closed channel modes $\kappa$

$$\kappa(\gamma_{r1} - \gamma_{r2}) = \sqrt{\gamma_{r1}\gamma_{r2}}\, e^{j\psi}(\omega_1 - \omega_2), \tag{16}$$

$$\psi = m\pi, \; m \in \mathscr{Z} \tag{17}$$

Substituting $(\gamma_{r1} - \gamma_{r2})$ from equation (16) into equation (15), it is possible to find that the third term with the square root is exactly equal to the second term in equation (14) and cancels, or adds with it, depending on the sign $\pm$. The dark mode acquires ideally zero radiation loss (say, $\widetilde{\omega}_-$ without loss of generality). At this condition, the eigenvalues are

$$\widetilde{\omega}_+ + j\widetilde{\gamma}_+ = \frac{\omega_1 + \omega_2}{2} + \frac{\kappa(\gamma_{r1} + \gamma_{r2})}{2\sqrt{\gamma_{r1}\gamma_{r2}}\, e^{j\psi}} + j(\gamma_{r1} + \gamma_{r2}), \tag{18}$$

$$\widetilde{\omega}_- + j\widetilde{\gamma}_- = \frac{\omega_1 + \omega_2}{2} - \frac{\kappa(\gamma_{r1} + \gamma_{r2})}{2\sqrt{\gamma_{r1}\gamma_{r2}}\, e^{j\psi}}, \quad \text{with } \widetilde{\gamma}_- = 0, \tag{19}$$

in which the wave of amplitude $\widetilde{A}_-$ has no radiative loss and becomes the ideal FW-BIC (ideally dark mode), whereas all radiative loss is transferred to the bright mode $\widetilde{A}_+$. At this point in momentum space ($\mathbf{k} = \mathbf{k}_{BIC}$), $\hat{\Omega}$ and $\hat{\Gamma}_r$ are both diagonal, and because $\mathrm{rank}(\hat{\Gamma}_r) = 1$ (only a single independent decay port exists), the resonant states interfere to annihilate the coupling with the radiation channel of the BIC mode, which guarantees energy conservation, as any coupling among the final orthogonal modes asymptotically vanishes[47].

However, arbitrarily close to the BIC point in the momentum, both modes experience non-zero radiative loss. The modes are coupled with a single independent radiation channel and, thus, are non-orthogonal because their coupling guarantees energy conservation. This behaviour holds true in any real system, particularly with momentum close to ideal FW-BICs, referred to as FW quasi-BICs. It is worth mentioning that, in the presence of non-negligible absorption loss, the modes are always non-orthogonal. If we perturb the ideal FW-BIC condition by moving in momentum space, in the representation in which $\hat{\Omega}$ is diagonal, in general, $\hat{\Gamma}_r$ must have non-zero off-diagonal terms to ensure energy conservation, or similarly, in the representation in which $\hat{\Gamma}_r$ is diagonal, $\hat{\Omega}$ must have non-zero off-diagonal terms, $\kappa_{12,21}$, which represent the near-field coupling. This is a key concept that implies that $\forall \mathbf{k} : \mathbf{k} \simeq \mathbf{k}_{BIC}$, the new perturbed Hamiltonian $\hat{H}^r(\mathbf{k} \simeq \mathbf{k}_{BIC})$ for the final coupled modes, the FW quasi-BIC $A_-(\mathbf{k} \simeq \mathbf{k}_{BIC})$ and bright $A_+(\mathbf{k} \simeq \mathbf{k}_{BIC})$ modes, can be represented with non-zero off-diagonal terms in $\hat{\Omega}(\mathbf{k} \simeq \mathbf{k}_{BIC})$, when $\hat{\Gamma}_r$ is diagonal because of energy conservation, as described below (Extended Data Fig. 1a).

The same non-Hermitian Hamiltonian can also describe the effect of coupled-resonance-induced transparency resulting from the interference of non-orthogonal eigenvectors, that is, at a wavevector different from the ideal FW-BIC condition. Hsu et al. demonstrated that, when several resonances (two or more) are connected to a single independent decay port, a transparency window, known as coupled-resonance-induced transparency, always occurs regardless of the radiation loss values of the resonances because of the off-diagonal terms[21]. Therefore, this coupling, also necessary for any FW quasi-BIC point, can give rise to coupled-resonance-induced transparency in special cases. The condition for EIT can, in principle, also occur with momentum near the ideal FW-BIC point, for example, when $\mathbf{k}_{EIT} = \mathbf{k}_{BIC} + \delta\mathbf{k}$ (Extended Data Fig. 1a). At the EIT point, the slow light condition increases the photon–matter interaction time, enhancing emission properties.

**Supercritical coupling. Coupled-resonance-induced transparency in far-field representation.** We first describe the occurrence of the transparency condition in the far-field representation and its link with the near-field representation. We then consider the perturbation of the Hamiltonian close to the FW-BIC to explicitly demonstrate that the FW quasi-BIC, despite being a quasi-dark mode, can reach the maximum physical limit of the local field enhancement under the supercritical coupling condition, thanks to the near-field coupling with its bright partner. The calculations presented here follow refs. 21,25 for clarity of description, but with harmonic time dependence convention $\exp(j\omega_{in}t)$. Let us first restate the TCMT problem by writing the dynamical equations for the two modes that are non-orthogonal in the representation in which $\hat{\Omega}(\mathbf{k})$ is diagonal, with a single radiation channel. Because the representation is changed with respect to equation (11), we consider different symbols for elements in the matrices and we adopt this representation only because the condition for EIT emergence is rather simple to show:

$$\frac{d}{dt}\begin{pmatrix} A_1 \\ A_2 \end{pmatrix} = \left[ j\begin{pmatrix} \overline{\omega}_1 & 0 \\ 0 & \overline{\omega}_2 \end{pmatrix} - \begin{pmatrix} \overline{\gamma}_{r1} & \gamma_{12} \\ \gamma_{12} & \overline{\gamma}_{r2} \end{pmatrix} - \begin{pmatrix} \gamma_a & 0 \\ 0 & \gamma_a \end{pmatrix} \right]\begin{pmatrix} A_1 \\ A_2 \end{pmatrix} + \begin{pmatrix} d_1 \\ d_2 \end{pmatrix}s^+, \quad (20)$$

$$s^- = c_{21}s^+ + d_1 A_1 + d_2 A_2. \quad (21)$$

In equation (20), the off-diagonal terms $\gamma_{12}$ in the radiative decay matrix must be non-zero for energy conservation if both modes decay in the channel, meaning that the decay matrix and the frequency matrix cannot have diagonal forms simultaneously[21,25]. In equation (21), $s^-$ is the transmitted wave and we have, owing to the presence of the substrate-breaking vertical symmetry, that the direct scattering matrix elements are $c_{11} = -c_{22} = (1-n)/(1+n)$, with $n$ index of the substrate and $c_{12} = c_{21} = 2\sqrt{n}/(1+n)$. Equation (21) simplifies when the system is mirror symmetric because $n = 1$ (ref. 21). Invoking again energy conservation and time-reversal symmetry and using the relations between $\hat{\Gamma}_r$, $\hat{C}$ and $\hat{D}$:

$$d_{1,2} = j\sqrt{2\overline{\gamma}_{r1,r2}/(n+1)}, \quad (22)$$

$$\gamma_{12} = \sqrt{\overline{\gamma}_{r1}\overline{\gamma}_{r2}}. \quad (23)$$

Let us keep using a mirror-symmetric system to determine the condition of induced transparency. The experimental case is then calculated with RCWA, showing that the condition for induced transparency also holds for vertical asymmetry. The complex transmission coefficient at regime is[25]

$$t = c_{21} \mp \frac{(c_{11}\pm c_{12})[j(\omega_{in}-\overline{\omega}_2)+\gamma_a]\overline{\gamma}_{r1} + [j(\omega_{in}-\overline{\omega}_1)+\gamma_a]\overline{\gamma}_{r2}}{[j(\omega_{in}-\overline{\omega}_1)+\gamma_a+\overline{\gamma}_{r1}][j(\omega_{in}-\overline{\omega}_2)+\gamma_a+\overline{\gamma}_{r2}] - \overline{\gamma}_{r1}\overline{\gamma}_{r2}}, \quad (24)$$

in which $|c_{11}+c_{12}| = |c_{22}-c_{12}|$ and we have already established that absorption is the same for both modes and given by $\gamma_a$. The top (bottom) signs are used when both modes are even (odd) with respect to vertical symmetry. In the limit $\gamma_a \ll (\overline{\omega}_1-\overline{\omega}_2)^2/\max(\overline{\gamma}_{r1},\overline{\gamma}_{r2})$, the absorptive decay rate is sufficiently small that the transmission coefficient approaches 1 (EIT condition) when the numerator of the second term becomes zero at the transparency frequency $\omega_t$, given by

$$\omega_{in} = \frac{\overline{\omega}_1\overline{\gamma}_{r2} + \overline{\omega}_2\overline{\gamma}_{r1}}{\overline{\gamma}_{r1} + \overline{\gamma}_{r2}} \doteq \omega_t. \quad (25)$$

This condition is always fulfilled when $\overline{\omega}_1 < \omega_{in} < \overline{\omega}_2$ provided that the resonances are sufficiently close, regardless of their radiative damping. In a real system for $\gamma_a \neq 0$, the approximation to this condition is a

consequence of the optical theorem, for which $t$ cannot reach ideally 1. Nonetheless, the fast dispersion induced at the transparency frequency leads to an enhancement of the local optical field[50–52]. Indeed, when the EIT is approached, light is substantially slowed down, which favours light–matter interactions and enhances the optical-emission process. With this simple demonstration, we have proved that FW-BIC and EIT can be close in principle in the momentum space. Indeed, the induced transparency arises from the coupling of two optical modes to the same radiation channel, which is also the same framework near FW-BIC.

**Near-field representation.** Although the diagonal frequency matrix representation is useful for finding the transparency condition, the next one will provide more insight into the mode coupling. Let us now rewrite the dynamic equations (20) in the representation in which the radiative decay is diagonal. We will indicate the final eigenvector waves at $\mathbf{k} = \mathbf{k}_{EIT}$ with amplitudes $A'_+$ and $A'_-$ (not to be confused with the amplitudes $\widetilde{A}_+$, $\widetilde{A}_-$ at the FW-BIC wavevector $\mathbf{k} = \mathbf{k}_{BIC}$ in equation (13). As mentioned earlier, $\mathrm{rank}(\hat{\Gamma}_r) = \mathrm{rank}(\hat{D}) = 1$. Thus, in its diagonal representation, $\hat{\Gamma}_r$ has only one non-trivial element because the determinant must be zero. It is straightforward to demonstrate that, in this equivalent representation (with $c_{21} = 1$),

$$\frac{d}{dt}\begin{pmatrix} A'_+ \\ A'_- \end{pmatrix} = \left[ j\begin{pmatrix} \omega'_+ & \kappa'_{12} \\ \kappa'_{12} & \omega'_- \end{pmatrix} - \begin{pmatrix} \gamma'_+ & 0 \\ 0 & 0 \end{pmatrix} - \begin{pmatrix} \gamma'_a & \zeta'_{12} \\ \zeta'_{21} & \gamma'_a \end{pmatrix} \right]\begin{pmatrix} A'_+ \\ A'_- \end{pmatrix} + \begin{pmatrix} d'_1 \\ 0 \end{pmatrix}s^+, \quad (26)$$

$$s^- = s^+ + d'_1 A'_+, \quad (27)$$

in which the connection with the previous representation of the diagonal frequency matrix is given by:

$$\omega'_+ = \frac{\overline{\omega}_1\overline{\gamma}_{r1} + \overline{\omega}_2\overline{\gamma}_{r2}}{\overline{\gamma}_{r1} + \overline{\gamma}_{r2}}, \quad (28)$$

$$\omega'_- = \frac{\overline{\omega}_1\overline{\gamma}_{r2} + \overline{\omega}_2\overline{\gamma}_{r1}}{\overline{\gamma}_{r1} + \overline{\gamma}_{r2}}, \quad (29)$$

$$\kappa'_{12} = \frac{(\overline{\omega}_2 - \overline{\omega}_1)\sqrt{\overline{\gamma}_{r1}\overline{\gamma}_{r2}}}{\overline{\gamma}_{r1} + \overline{\gamma}_{r2}}, \quad (30)$$

$$\gamma'_+ = \overline{\gamma}_{r1} + \overline{\gamma}_{r2}, \quad (31)$$

$$\gamma'_- = 0, \quad (32)$$

$$d'_1 = \sqrt{d_1^2 + d_2^2}. \quad (33)$$

The above relations are useful because they directly state that the transparency frequency $\omega_t = \omega'_-$, that is, it corresponds to the final dark mode. This link is important: at the transparency frequency, the fast dispersion slows down the light and enhances the local field, which corresponds to the dark mode. Although in the previous representation we were dealing with non-orthogonal modes in which their coupling was expressed in the far field, in this second representation, we can see that a non-radiative dark mode with $\gamma'_- = 0$ is coupled by means of a non-zero near-field constant $\kappa'_{12}$ to a bright leaky wave with a decay rate $\gamma'_+ = \overline{\gamma}_{r1} + \overline{\gamma}_{r2}$. These identities must not be confused with equations (18) and (19) that express the relations between the diagonal dark and bright modes at the FW-BIC point $\mathbf{k} = \mathbf{k}_{BIC}$ with the original uncoupled modes. Instead, the above equations refer to two different representations of the same modes at fixed and same wavevector $\mathbf{k} = \mathbf{k}_{EIT} \neq \mathbf{k}_{BIC}$. Here, when the drive field is turned off, the dark-mode amplitude decays to zero. In the linear regime, exchange energy occurs

between the modes. We see below that, while the drive field is on, energy flows from the bright mode to the dark mode. As the drive field is turned off, energy flows from the dark mode to the bright mode. Consequently, the dark mode undergoes decay in the far field owing to its nearly zero direct coupling with the radiation channel and its non-zero near-field coupling with the bright mode[53]. In this alternative representation, it is the near-field coupling between a dark mode and the bright mode that gives rise to the transparency condition. This formulation aligns with the general framework used in the subradiant–superradiant model, which illustrates the analogue of EIT in photonic and plasmonic systems[50–52].

**Maximum enhancement at the FW quasi-BIC.** The FW-BIC and classical analogue of EIT formalisms are derived from the same original framework of modes coupled to a single radiation channel: the EIT with non-zero off-diagonal terms, whereas the ideal FW-BIC is a limit of this framework with zero off-diagonal terms. Because the EIT occurs at the avoided crossing, FW-BIC must not be at the avoided crossing, which implies that the radiative decay rates of the closed channel modes in equation (16) differ, $\gamma_{r1} \neq \gamma_{r2}$. Thus, the ideal FW-BIC is not at the avoided crossing ($\omega_1 = \omega_2$) but is shifted in its vicinity. Both conditions can be fulfilled, in principle, for close wavevectors when, for example, $\gamma_{r1} \simeq 5\gamma_{r2}$ (see the simulated linewidths when the modes do not cross each other in Extended Data Fig. 3; orientation angle of the photonic crystal $\phi = 45°$). This also means that the realization of EIT is possible when the involved dark mode is a perturbation of the FW-BIC mode, that is, it exhibits characteristics of an FW quasi-BIC. Although this will be shown using RCWA in our system, let us now explore the consequences for enhancing the local optical field.

As shown in the scheme of Extended Data Fig. 1a, let us write explicitly the dynamical equations (13) and add the perturbation of the diagonal representation (FW-BIC point) of the Hamiltonian as we move away from the ideal BIC wavevector towards the EIT point. Because the radiative $Q$ factor of a BIC scales as $|\mathbf{k} - \mathbf{k}_{BIC}|^{-\alpha}$ with $\alpha \geq 2$, for any wavevector close to the BIC point, $\mathbf{k} = \mathbf{k}_{BIC} + \Delta\mathbf{q} \simeq \mathbf{k}_{BIC}$, it is necessary to admit a finite non-zero decay rate of the dark mode $A_-$, that is, $1/\gamma_- = \tau_{R1}$ with $\gamma_- \to \varepsilon \gtrsim 0$ and, as such, it is necessary to include a non-zero mode coupling $\kappa_{12} \neq 0$ to guarantee energy conservation, as both modes are coupled to a single independent radiation channel. The perturbed Hamiltonian is $\hat{H}^r(\mathbf{k} \simeq \mathbf{k}_{BIC}) = \begin{pmatrix} \omega_+ & \kappa_{12} \\ \kappa_{12} & \omega_- \end{pmatrix} + j\begin{pmatrix} \gamma_+ & 0 \\ 0 & \gamma_- \end{pmatrix}$. It is important to note that the modes are the final coupled modes: their frequencies are considered shifted with respect to the exact frequencies of bright and dark modes of the FW point $\mathbf{k} = \mathbf{k}_{BIC}$ in equation (14). The finite decay rate of the dark mode turns it into a quasi-dark mode (FW quasi-BIC), and this non-zero coupling to the radiation channel ($\sqrt{2\gamma_-} = \sqrt{2/\tau_{R1}}$) will imply non-zero near-field ($\kappa_{12}$) or far-field ($\gamma_{12}$) coupling with the shifted bright partner, depending on the representation used. The bright mode has amplitude $A_+$, with a decay rate $1/\gamma_+ = \tau_{R2} \ll \tau_{R1}$. Generally, the off-diagonal terms can be kept complex to include both near-field and far-field coupling, but we have verified by RCWA that the coupling is real with good approximation in the next section. Here we assume the representation with near-field coupling $\kappa_{12}$. Considering the general dynamical equations with both modes having the same losses included all in $\gamma_a = 1/\tau_a$, it is possible to write, $\forall \mathbf{k} : \mathbf{k} \simeq \mathbf{k}_{BIC}$ that

$$\frac{dA_-}{dt} = j\omega_- A_- - \left(\frac{1}{\tau_a} + \frac{1}{\tau_{R1}}\right)A_- + j\kappa_{12}A_+ + \sqrt{\frac{2}{\tau_{R1}}}\, s_+, \tag{34}$$

$$\frac{dA_+}{dt} = j\omega_+ A_+ - \left(\frac{1}{\tau_a} + \frac{1}{\tau_{R2}}\right)A_+ + j\kappa_{12}A_- + \sqrt{\frac{2}{\tau_{R2}}}\, s_+. \tag{35}$$

This set of equations is valid for any system (for example, plasmonic modes, whispering-gallery modes, guided modes, defect modes).

Considering $\frac{d}{dt} \to j\omega_{in}$ and solving for $A_-$ in equation (34), substituting it in equation (35) and then substituting the resulting $A_+$ again in equation (34), we find, at the steady state, that

$$\frac{A_-(\omega_{in})}{s_+} = \frac{\sqrt{\frac{2}{\tau_{R1}}}}{j(\omega_{in} - \omega_-) + \frac{1}{\tau_a} + \frac{1}{\tau_{R1}}} +$$
$$+ \frac{j/\tau_\kappa \sqrt{\frac{2}{\tau_{R2}}}}{\left[j(\omega_{in} - \omega_-) + \frac{1}{\tau_a} + \frac{1}{\tau_{R1}}\right]\left[j(\omega_{in} - \omega_+) + \frac{1}{\tau_a} + \frac{1}{\tau_{R2}} + \frac{1/\tau_\kappa^2}{j(\omega_{in} - \omega_-) + 1/\tau_a + 1/\tau_{R1}}\right]} +$$
$$- \frac{1/\tau_\kappa^2 \sqrt{\frac{2}{\tau_{R1}}}}{\left[j(\omega_{in} - \omega_-) + \frac{1}{\tau_a} + \frac{1}{\tau_{R1}}\right]^2\left[j(\omega_{in} - \omega_+) + \frac{1}{\tau_a} + \frac{1}{\tau_{R2}} + \frac{1/\tau_\kappa^2}{j(\omega_{in} - \omega_-) + 1/\tau_a + 1/\tau_{R1}}\right]}, \tag{36}$$

$$\frac{A_+(\omega_{in})}{s_+} = \frac{\sqrt{\frac{2}{\tau_{R2}}}}{j(\omega_{in} - \omega_+) + \frac{1}{\tau_a} + \frac{1}{\tau_{R2}} + \frac{1/\tau_\kappa^2}{j(\omega_{in} - \omega_-) + 1/\tau_a + 1/\tau_{R1}}} +$$
$$+ \frac{j/\tau_\kappa \sqrt{\frac{2}{\tau_{R1}}}}{\left[j(\omega_{in} - \omega_-) + \frac{1}{\tau_a} + \frac{1}{\tau_{R1}}\right]\left[j(\omega_{in} - \omega_+) + \frac{1}{\tau_a} + \frac{1}{\tau_{R2}} + \frac{1/\tau_\kappa^2}{j(\omega_{in} - \omega_-) + 1/\tau_a + 1/\tau_{R1}}\right]}. \tag{37}$$

Above, we have explicitly defined the near-field coupling lifetime $\tau_\kappa = \frac{1}{\kappa_{12}}$ and the associated quality factor $\tau_\kappa = 2Q_\kappa/\omega$. We can see that the quasi-dark mode $A_-$ can be excited by means of internal coupling $\kappa_{12}$ more than what is expected from the isolated resonance response of the dark mode, represented by the first term in equation (36) (in Supplementary Information section 1.2 and Supplementary Fig. 4, the mediated drive term is also made explicit in the original quantum model)[2]. In Extended Data Fig. 1b–d, the behaviour for both mode intensities for a specific set of informative values, $Q_{R1} = 5 \times 10^9$, $Q_{R2} = 200$, $Q_a = 5,000$ is plotted to capture the main insight. In Extended Data Fig. 1b, the intensity field enhancement

$$G = \frac{|A_\pm|^2}{|s_+/\sqrt{\omega_{in}}|^2 V_{eff}} \tag{38}$$

is plotted for both modes (solid red line for the dark $A_-$ and blue line for the bright $A_+$), showing that the dark mode on resonance ($\omega_{in} = \omega_-$) reaches the maximum limit of field enhancement possible in a real-world resonator with non-radiative loss, $G_{max} = Q_a/V_{eff}$, even if

$$Q_{R1} \gg Q_a, \tag{39}$$

which would be impossible in case of a single dark resonance, that is, not coupled to another wave (dashed red line). This condition occurs at the supercritical coupling point defined by

$$\overline{\tau}_\kappa = \sqrt{\tau_{R2}\tau_a}, \tag{40}$$

or

$$\overline{Q}_\kappa = \sqrt{Q_{R2}Q_a}. \tag{41}$$

Indeed, assuming $\tau_{R1} \gg \tau_a$, $\tau_{R2}$, $\tau_\kappa$ and $\tau_{R2} < \tau_a$ in equation (36) and considering $\omega_{in} = \omega_-$ (on resonance with the dark mode) and $|\omega_{in} - \omega_+| \simeq 2\kappa_{12} = 2/\tau_\kappa$ (the coupling affects the split in frequencies, thus the pump is shifted from the bright mode when on resonance with the dark one), the relation simplifies as

$$\frac{A_-(\omega_{in} = \omega_-)}{s_+} \to \left[\frac{j/\tau_\kappa \sqrt{\frac{2}{\tau_{R2}}}}{j/(\tau_\kappa \tau_a) + 1/\tau_a^2 + 1/(\tau_{R2}\tau_a) + 1/\tau_\kappa^2}\right]_{\tau_\kappa = \sqrt{\tau_{R2}\tau_a}} \to j\sqrt{\tau_a/2}, \tag{42}$$

in which the first two terms in the denominator were neglected, as they are smaller when $\tau_{R2} < \tau_a$. The above relation proves that the dark-mode intensity enhancement $G = |\frac{A_-(\omega_{in} = \omega_-)}{s_+/\sqrt{\omega_{in}}}|^2 \frac{1}{V_{eff}} = Q_a/V_{eff} = G_{max}$, that is, it can reach the maximum imposed by non-radiative losses even in extreme situations with mismatched quality factors. It is worth mentioning that, when $Q_\kappa \to \infty$ ($\kappa_{12} \to 0$), we again obtain the correct case of uncoupled resonances and the dark-mode field goes to the level it could gain if it were isolated (dashed red line). Indeed, in the plot, we have specified that the near-field coupling rate affects the spectral separation among resonances, as it is proportional to their distance: $\omega_\pm = \omega_o \pm \kappa_{12} = \omega_o[1 \pm 1/(2Q_\kappa)]$ Thus, for $Q_\kappa \to \infty$, the resonant frequencies coincide and cross. Even when out of perfect spectral tuning, the maximum gain achieved by the quasi-dark mode $A_-$ is orders of magnitudes larger than what possible in a single dark mode, as shown in Extended Data Fig. 1c,d. In case $\omega_{in} = \omega_o = 1/2(\omega_+ + \omega_-)$, the optimum shifts to larger $Q^*_\kappa \simeq Q_a$. When $Q_{R2} \to Q_a$ and $\omega_{in} = \omega_-$, the bright mode is critically coupled with the pump, but there is still energy going into the dark mode up to $0.3G_{max}$ at a certain $Q^*_\kappa \lesssim \overline{Q}_\kappa = Q_a$. Furthermore, by inspecting the ratio between the solid red line and the dashed red line, it is possible to appreciate how, even if $Q_\kappa$ does not reach the optimum, the intensity of the coupled dark resonance is orders of magnitude larger than that of the single resonance.

**Further discussion.** The supercritical coupling mechanism guarantees the possibility of achieving the maximum level of local field enhancement when the coupling ($Q_\kappa$) is optimally tuned, and always in the highest $Q$-factor mode, even under the conditions of coupling, for both bright and dark modes, which would be unfavourable in the case of single isolated modes. To give an example, let $Q_{R2} = 10^3 \ll Q_a = 10^6 \ll Q_{R1} = 10^{10}$, thus none of the modes matches $Q_a$; by contrast, they have completely unmatched quality factors. If $Q_\kappa = \sqrt{10^3 \times 10^6} \simeq 3 \times 10^4$ (say, $V_{eff} = 1$ for brevity), the dark mode reaches the maximum intensity enhancement $G_{max} = 10^6$, although the intensity enhancement of the single dark resonance would be only $10^2$, that is, four orders of magnitude less, as shown in Fig. 1e. Also, the supercritical coupling condition is independent of the highest $Q$-factor resonance, unlike the critical coupling condition ($Q_{R1} = Q_a$); the model converges to the critical-coupling result if $Q_{R1} \to Q_a$ and can ensure a higher level of enhancement in the dark mode, with a considerable advantage over the single-dark-resonance case, even when $Q_{R2}$ and $Q_\kappa$ vary over a considerably large range of values. This is shown for fixed $\overline{Q}_\kappa = \sqrt{Q_{R2}Q_a}$ in Extended Data Fig. 1e.

This mechanism holds true for all wavevectors that span the range from an FW quasi-BIC to the EIT point (if this is also present in the system), with correspondingly varied values of the parameters involved (coupled mode frequencies, decay rates and near-field coupling). Far from this momentum region, the mode coupling becomes progressively negligible (as it can be easily calculated numerically) and the isolated single mode response is restored.

Turning to the parallel with coupled-resonance-induced transparency, we understand that, at the dark mode frequency $\omega_t = \omega'_-$ (equations (25) and (29)), in which the transparency window occurs, the fast dispersion leads to slow light and an enhanced field that, with suitable coupling between modes, could reach the maximum field enhancement of the system, as indicated by supercritical coupling. We recall that EIT is not a necessary condition for the FW mechanism, although it may widen, if present, the wavevector span of an enhanced field.

**RCWA validation.** The validity of TCMT is confirmed through numerical simulations using full 3D RCWA. RCWA simulations are performed using the Fourier modal expansion method (Ansys Lumerical, RCWA module). Validation is performed by evaluating the exact transmittance spectra, the 3D-vector-field distribution of the interfering modes, their complex coupling constant, their evolution with momentum, the near-field coupling at EIT, FW quasi-BIC and FW-BIC points in momentum space. The modes belonging to the dispersion curves are a linear combination of tens to hundreds of Fourier plane waves in each

$xy$-periodic, $z$-homogeneous layer satisfying the continuity boundary conditions in each $z$ layer of the structure (with forward and backward propagating factors along the $z$ axis), providing the exact solution of the problem, including material dispersion, matching the experimental transmittance spectrum measured to reconstruct the energy–momentum band diagrams for both $s$-polarized (vector TE-like character) and $p$-polarized (vector TM-like character) excitation. RCWA is indeed used as a benchmark for validating other numerical techniques such as resonant-state expansion, quasi-normal modes and other methods. It provides the 3D vector fields and the exact solution, which can be analytically approximated by the leaky TE-like and TM-like modes of the effective waveguide, or TCMT. Further details are in Supplementary Information with measured refractive index dispersion (Supplementary Fig. 1) and details on fitting, giving imaginary refractive index used for simulations $n_I = 10^{-4}$ over the spectral range 700–1,200 nm.

Extended Data Fig. 2a shows the theoretical TE bands expected for a uniform film of upconversion nanoparticles (UCNPs) with a refractive index of 1.45, matching the experimental absorption band of UCNPs in Extended Data Fig. 2b. Extended Data Fig. 2c shows the mode distribution, whereas Extended Data Fig. 2d evaluates the mode energy fraction superimposed on the nonlinear material as a function of refractive index, for one layer (1L), two layers (2L) and with a cladding of air or silicone oil. The silicone oil promotes vertical symmetry, which means that it increases the field overlap with the UCNPs and helps minimize scattering losses, but it cannot affect the vertical symmetry of the TE-like and TM-like modes, which is determined mainly by the different refractive index of the glass substrate, silicon nitride and UCNPs index. Indeed, the energy fraction with silicone oil only changes from 8% to 9% (Extended Data Fig. 2d). Nonetheless, silicone oil was often useful to better observe the side emission, as the silicone layer acted as a partially opaque screen crossing the outcoupled light (as shown in Fig. 3b). Note that the silicone oil layer was not used in Fig. 4b.

Extended Data Fig. 3 shows the evolution of the transmittance spectra by changing the azimuthal angle of incidence $\phi$. The avoided crossing stops only when the two modes no longer intersect, as shown clearly in Extended Data Fig. 3b at $\phi = 45°$, at which it is also possible to observe that the uncoupled mode 1 has linewidth larger than mode 2, that is, $\gamma_{r1} \gg \gamma_{r2}$. Extended Data Fig. 3c,d shows the details of FW quasi-BIC and avoided crossing.

Extended Data Fig. 4 shows that vector TE-like and TM-like modes evolve and change symmetry along the momentum; they are, in general, non-orthogonal and nearly coincident at the avoided crossing (and approximately even with respect to the $z$-mirror symmetry). Because the modes are nearly coincident, the approximation $\gamma_a = 1/\tau_a$ in the above model, that is, the same for both modes, is correct. Also, because the input intensity is $I_{input} = 1$, the resonance field intensity is much larger than what would be expected on the basis of critical coupling (material absorption loss, $n_I = 10^{-4}$ is included in the simulation), providing an estimate of the field enhancement ($I_1 > 3 \times 10^4 I_{input}$).

Extended Data Fig. 5a shows the spectral coincidence of the coupled-resonance-induced transparency (EIT) frequency (for $\theta = 2.7°$ at the avoided crossing) with the FW quasi-BIC frequency at $\theta = 3.15°$ for the angle mismatch <0.5° (mismatched momentum $\mathbf{k}_{EIT} = \mathbf{k}_{BIC} + \delta\mathbf{k}$). The existence of coupled-resonance-induced transparency can only occur for non-orthogonal modes[25], and the proximity in momentum space to the BIC point proves that FW-BIC is an ideal condition originating from the evolution of non-orthogonal modes. Extended Data Fig. 5b shows the near-field coupling constant normalized to $\omega'_- = 2\pi c/\lambda_{model}$ calculated using the formula in ref. 54 (equation (4.13), page 162, including material distribution), for $\theta$ from 2.7° (EIT) to 3.24° (nearly ideal FW-BIC). The phase mismatch is minimal, thus the two modes also exchange energy along the propagation (Pendellösung effect), as it commonly occurs between two modes of the same waveguide coupled by a periodic modulation[15,54]. The near-field coupling was calculated as

$$\kappa_{12} = \frac{1}{4} \sqrt{\frac{\varepsilon_o}{\mu_o}} \frac{k_o}{\sqrt{N_1 N_2}} \int (\varepsilon - \varepsilon_o) \mathbf{E}_1{}^\star \cdot \mathbf{E}_2 dA,$$

in which $N_{1,2} = \frac{1}{2} \left| \int (\mathbf{E}^*_{1,2} \times \mathbf{H}_{1,2} + \mathbf{c}.\mathbf{c}.) \cdot \hat{z} dA \right|$ are optical power normalizations. The integral is over the unitary cell area $A$. Note that the calculation provides the complex $\kappa_{12}$, in which the imaginary part of $\kappa_{12}$ is to be understood as a representation of $\zeta'_{12}$ in equation (26) above. We estimated that $\zeta'_{12} < 10^{-4} \kappa_{12}$ for all modes in the range $\theta \in (0°, 5°)$, thus $\zeta'_{12} \simeq 0$. Also, we found that $\kappa_{12} \simeq \kappa_{21}$, as expected. The near-field coupling is stronger at the EIT point, whereas it decreases at the ideal FW-BIC, in agreement with the behaviour expected from the temporally coupled mode theory. As the incidence angle varies from the EIT point (2.7°) to the ideal position of the BIC (3.24°), $Q_\kappa = \tau_\kappa \omega/2$ varies accordingly and is characterized by a $Q_\kappa \approx (10^3, 10^4) \approx \sqrt{Q_{R2} Q_a}$ at the FW quasi-BIC mode (dashed black line, 3.15°). As the near-field coupling is modulated, the fulfilment of the supercritical coupling condition can be tuned.

Supplementary Fig. 2 shows the evolution of the interference process as a function of $\kappa_{12}$ and describes how the coupling changes at the edge. The effect of the finite boundary on resonance was investigated using near-field scanning optical microscopy (Witec Alpha RAS 300) and shown in Supplementary Fig. 3.

Supplementary Fig. 4 shows theoretical linewidths calculated with the original FW quantum model[2], revealing that the open-channel wave acts as a drive field in the coupled BIC equation, for representative near-field coupling values.

### Fabrication
Extended Data Fig. 6 shows the energy-level scheme of the produced UCNPs. All materials and synthesis details of NPs, NP characterization, PCNS fabrication and characterization are in Supplementary Information sections 2–4 and Supplementary Figs. 5 and 6.

### Optical characterization
Dispersion-band-diagram measurements, experimental interrogation and detection scheme of upconversion are provided, respectively, in Supplementary Information sections 5 and 6 and Supplementary Figs. 7–10. For upconversion, the pulsed (150-fs) Ti:Sa oscillator, with central wavelength $\lambda_{in} = 810$ nm and full-width at half-maximum of 6 nm, is tuned to the FW quasi-BIC and focused to a 6-μm spot on the PCNS. The power coupled with NPs was 5%, corresponding to 48 kW cm$^{-2}$ at a pulse energy of 6.25 nJ ($10^3$ kW cm$^{-2}$).

### Photoluminescence, enhancement-factor and radiance-enhancement estimation
Enhancement-factor estimation, spectral emission datasets from samples and radiance-enhancement-factor estimation are provided, respectively, in Supplementary Information section 7 and Supplementary Figs. 11 and 12.

### FDTD simulations
The radiation properties of the PCNS were evaluated using the FDTD method in Ansys Lumerical. A single dipole source was used to compute the isofrequency map using the $Z$-transform of the local optical field retrieved within the finite-structure domain with the 3D full-field monitor. The intensity of the $Z$-transform determines the strength of radiation in the momentum space and better represents the radiation properties associated with the PCNS. To validate the results found with this approach, we first simulated a literature case discussed in ref. 44, that is, supercollimation resulting from flat-band dispersion in the momentum space, which is shown in Supplementary Fig. 13. The isofrequency far-field intensity map in momentum space showed, in our case, non-trivial vanishing strips along orthogonal arms (cross of zeros; Fig. 3 and Extended Data Fig. 7). The near-field intensity map showed self-collimation as occurring when flat dispersion is involved.

In Extended Data Fig. 7e, the experimental proof is reported using a rescaled geometry of the PCNS (using the fit in Extended Data Fig. 2e) to move the FW-BIC at 532 nm and make the beam easily visible. At this stage, the radiation properties were examined by placing an array of dipole sources (18 × 18) at the boundary of the finite PCNS with a uniform slab covering an area of several microns squared. The results are shown in Fig. 3c and Extended Data Fig. 8. The sources collectively add up their field and coherently emit radiation in the plane of the slab, as shown in Extended Data Fig. 8a, in which the field propagates along the direction (+1, 0) with intensity enhancement as large as $1.5 \times 10^4$ (normalized to the number of emitters). The emission was always pointing towards the non-textured slab, thus—on the opposite edge—the propagation was along the direction (−1, 0). It was found that, at shorter wavelengths, other preferential directions of propagation were also possible, such as (1, ±1). The divergence was evaluated along 1 mm of propagation from the edge, as shown in Extended Data Fig. 8b, which showed a divergence of 0.02° (Extended Data Fig. 8c), which is even lower than the experimental values. Analysis of the whole visible and near-infrared spectrum revealed that the typical value of the divergence is less than 0.5° (Extended Data Fig. 8d), demonstrating that this regime of narrow radiation is expected to be common in this type of photonic structure. Indeed, as shown in Extended Data Fig. 8e, the full width at half maximum of the beam periodically contracts and expands along the propagation, which is because of a mechanism of self-healing that compensates for diffraction.

### Directivity measurements
Extended Data Fig. 9a shows the microscopy inspection of light propagation near the edge. Extended Data Fig. 9b shows the experimental results on the divergence of the side beam (directed along the outer edge versor), with a polar plot of the edge emission in Extended Data Fig. 9c, in agreement with simulation in Extended Data Fig. 8 in the upconverted emission.

### Data availability
All relevant data that support the findings of this work are available from the authors and are included with the article and its Supplementary information.

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

**Acknowledgements** We thank G. Coppola, M. A. Ferrara and T. Crisci (Institute of Applied Sciences and Intelligent Systems, National Research Council, Italy) for assistance in spin coating and optical set-up; S. Dhuey (Molecular Foundry, Lawrence Berkeley National Laboratory, USA) for assistance in microstructure fabrication; and C.-W. Qiu (National University of Singapore, Singapore) for discussion. G.Z. acknowledges the project PIT-STOP (grant no. PRIN-2017-20173CRP3H, Ministry of University and Research, Italy) and Berkeley Lab, Molecular Foundry, user project no. 8254. Work at Molecular Foundry was supported by the Office of Science, Office of Basic Energy Sciences, of the US Department of Energy, under contract no. DE-AC02-05CH11231. V.M. acknowledges the EU Italian National Recovery and Resilience Plan (NRRP) of NextGenerationEU (PE0000023-NQSTI). I.R. acknowledges NRRP, NextGenerationEU (PE00000001-RESTART). X.L. acknowledges the RIE2025 Manufacturing, Trade and Connectivity (MTC) Programmatic Fund (award no. M21J9b0085) and National Research Foundation, Prime Minister's Office, Singapore under the NRF Investigatorship programme (award no. NRF-NRF105-2019-000).

**Author contributions** G.Z. conceived the idea. C.S., S.R., L.S., L.L., X.L. and G.Z. designed the experimental work. V.L. and F.R. fabricated samples. S.C. and A.S. supervised fabrication. L.L. and J.C. synthesized and characterized upconversion NPs. C.S., S.R. and G.Z. performed dispersion measurements and analysed data. C.S., S.R., L.S. and G.Z. conducted upconversion measurements and analysed data. V.M. and G.Z. performed numerical simulations. All authors discussed the results. X.L. and G.Z. wrote the original manuscript and revised it, with contributions from C.S., S.R., L.S. and L.L. X.L. and G.Z. supervised the research.

**Competing interests** The authors declare no competing interests.

**Additional information**

**Correspondence and requests for materials** should be addressed to Xiaogang Liu or Gianluigi Zito.

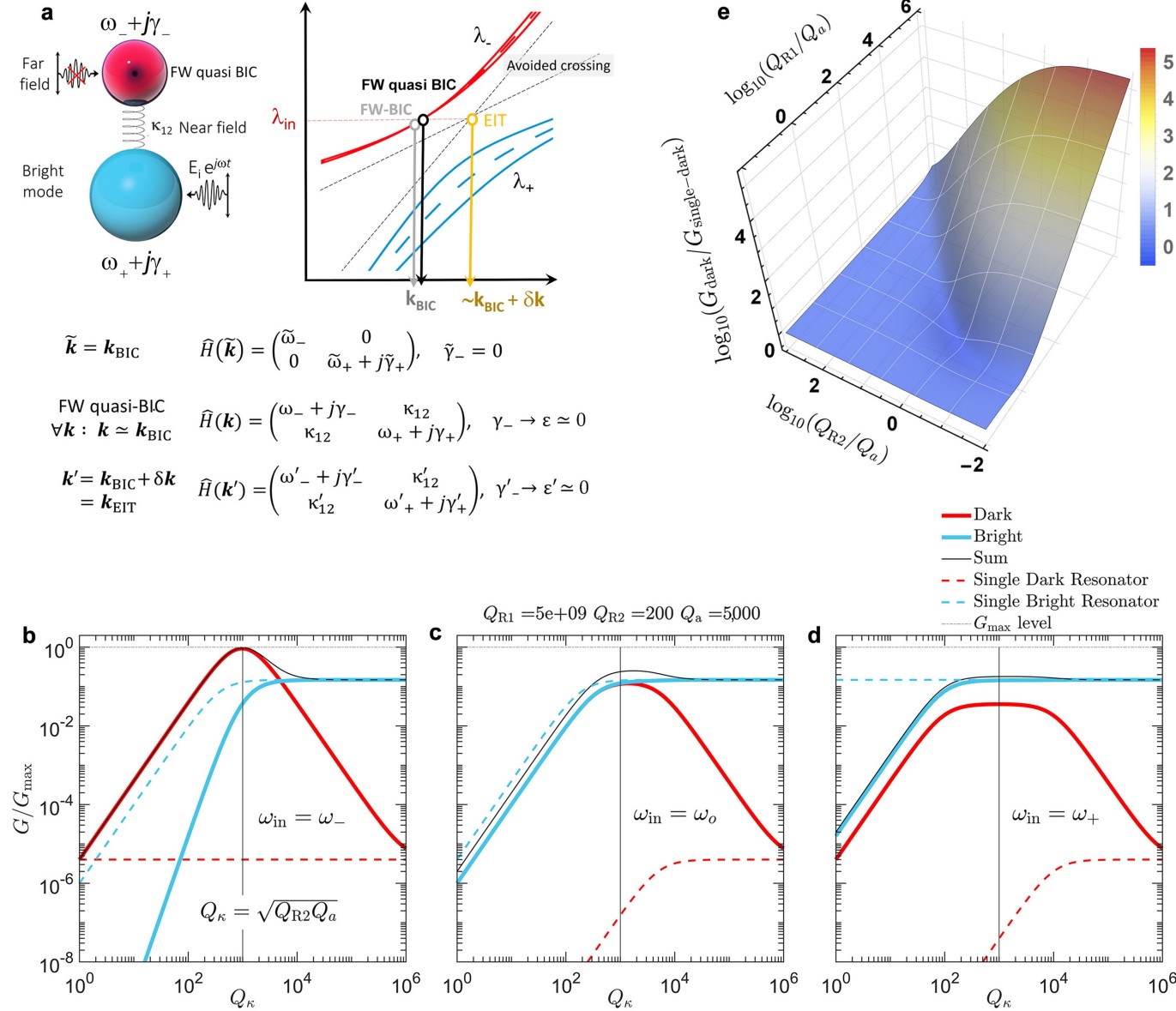

**Extended Data Fig. 1 | Supercritical coupling model. a**, Top, oscillator scheme at the FW quasi-BIC, with depicted dispersion diagram of the FW-BIC formation near the avoided crossing point and with mismatched momentum with respect to the EIT occurring at the avoided crossing (dashed lines, frequencies; solid lines, linewidth). Bottom, the corresponding Hamiltonians (frequency and decay rate) at the ideal FW-BIC ($k_{BIC}$), FW quasi-BIC ($\simeq k_{BIC}$) and EIT ($k_{EIT}$) points. **b**–**d**, Normalized intensity enhancements $G/G_{max}$ for both dark and bright modes (red and blue solid lines, respectively) compared with the corresponding single-resonance intensity enhancements (red and blue dashed lines) as a function of

$Q_\kappa$, for $Q_{R1} = 5 \times 10^9$, $Q_{R2} = 200$, $Q_a = 5,000$, with input frequency tuning with the dark (**b**), middle (**c**) and bright (**d**) frequency. **e**, Intensity-level ratio between the coupled dark mode $G_{dark}$ at supercritical coupling and the single dark mode $G_{single-dark}$ as a function of $Q_{R1}/Q_a$ and $Q_{R2}/Q_a$: when $Q_{R1}/Q_a = 1$, we find the critical coupling condition and the coupled dark mode has the same level of enhancement as the single dark mode. When $Q_{R2} \gg Q_a$, there is no advantage ($G_{dark} \rightarrow G_{single-dark}$) because the input channel is unfavourable. In the remaining region of parameters, $G_{dark} \gg G_{single-dark}$.

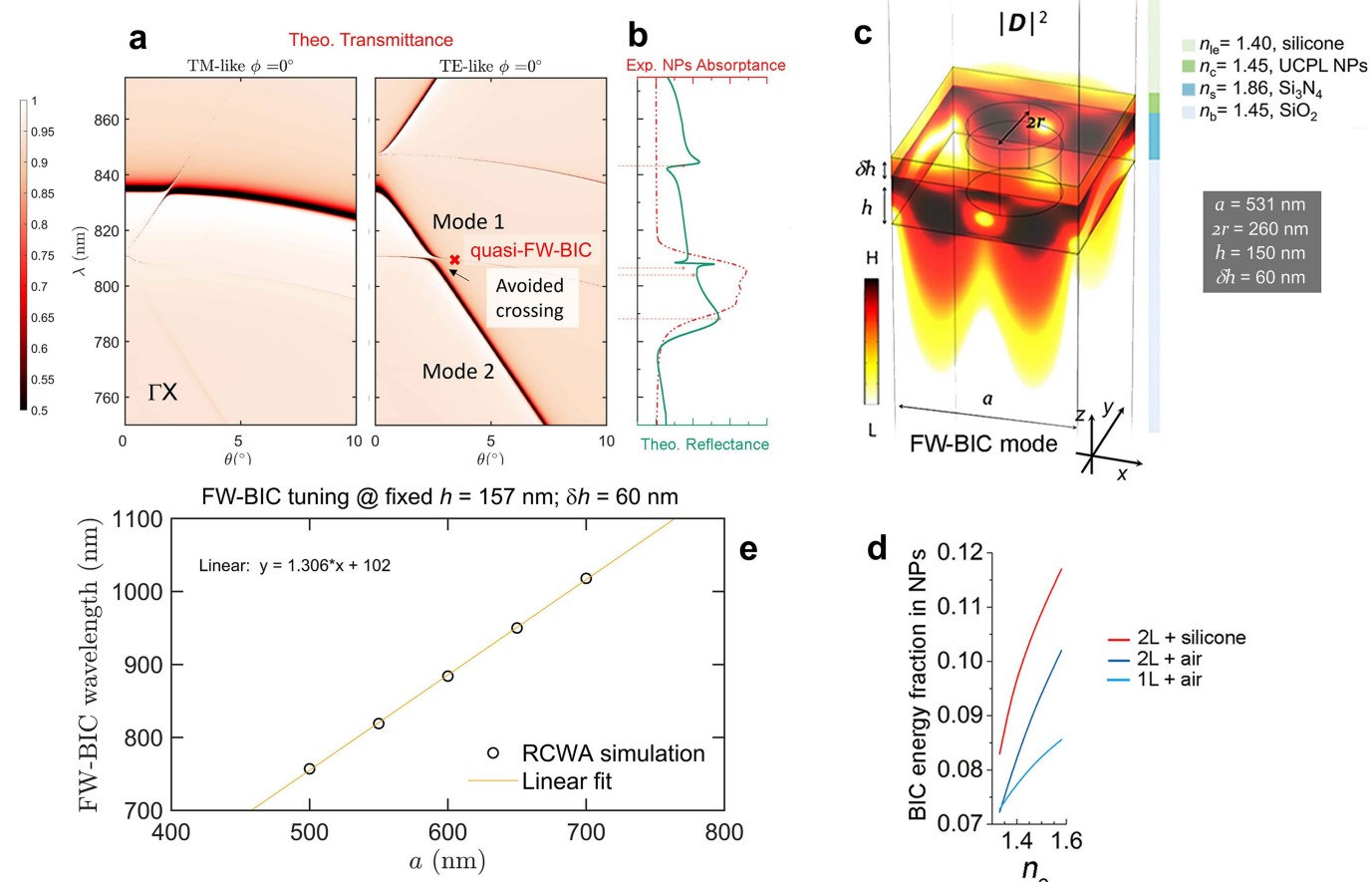

**Extended Data Fig. 2 | RCWA simulations for structure tuning. a**, Numerical band diagrams (RCWA) of the PCNS showing a FW-BIC close to the avoided crossing in momentum space (TE-like modes). **b**, Reflectance spectrum overlapping with the experimental absorption of NPs. **c**, FW-BIC displacement field intensity |**D**|² in the unit cell. **d**, Optical energy fraction in monolayer (1L) and bilayer (2L) NPs with air and silicone superstrates. **e**, RCWA simulation and linear fit for spectral tuning of the FW quasi-BIC position with the lattice constant $a$ for the radius of the circular hole $r = 0.244a$, thus scaling with $a$ (other parameters fixed).

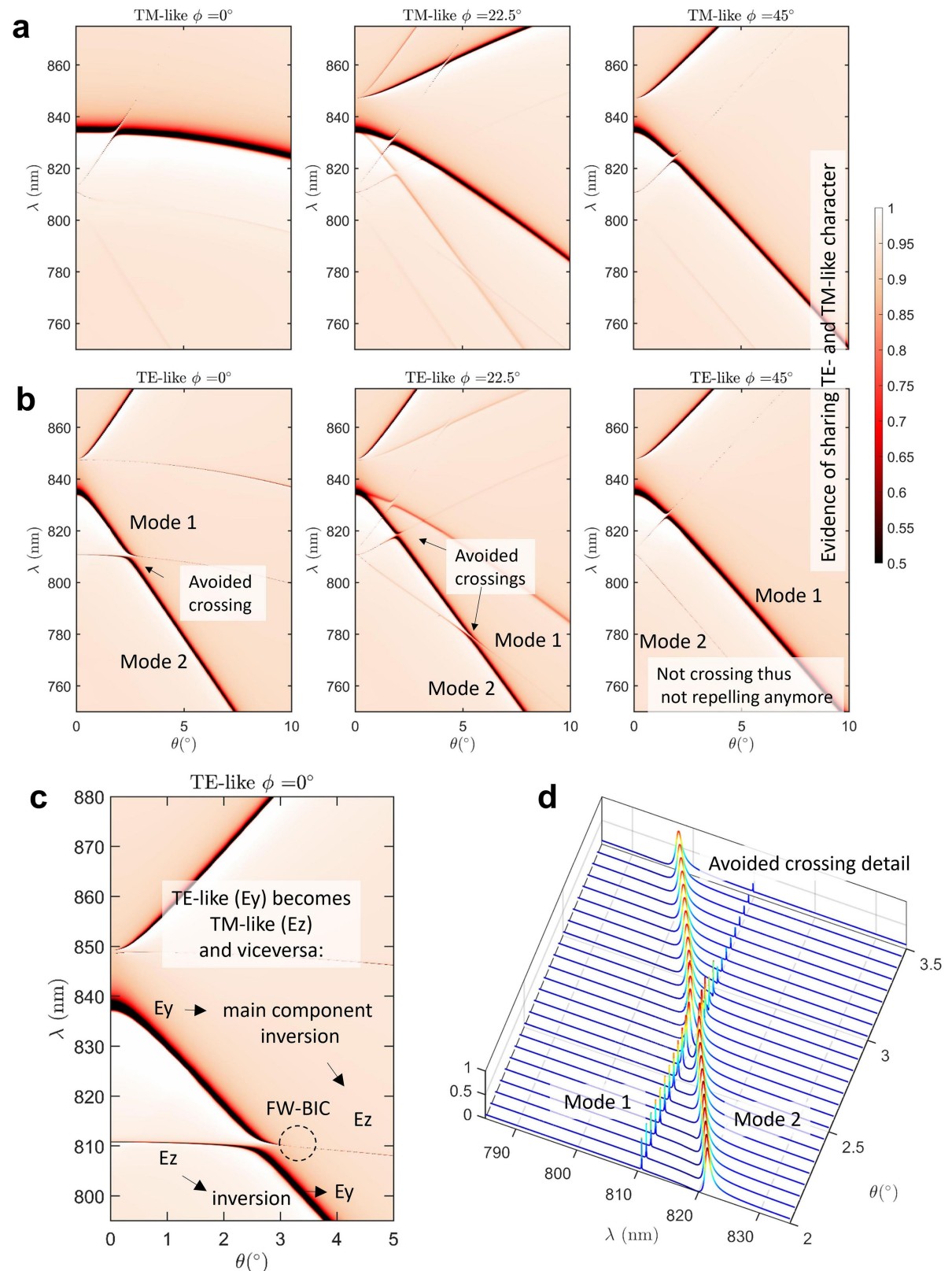

**Extended Data Fig. 3 | Detail of the avoided crossing. a**, *p*-polarized transmittance. **b**, *s*-polarized transmittance. **c**,**d**, Detail of avoided crossing and indication of vector TE-like and TM-like modes evolving into each other, with reference to modes shown in Extended Data Fig. 4.

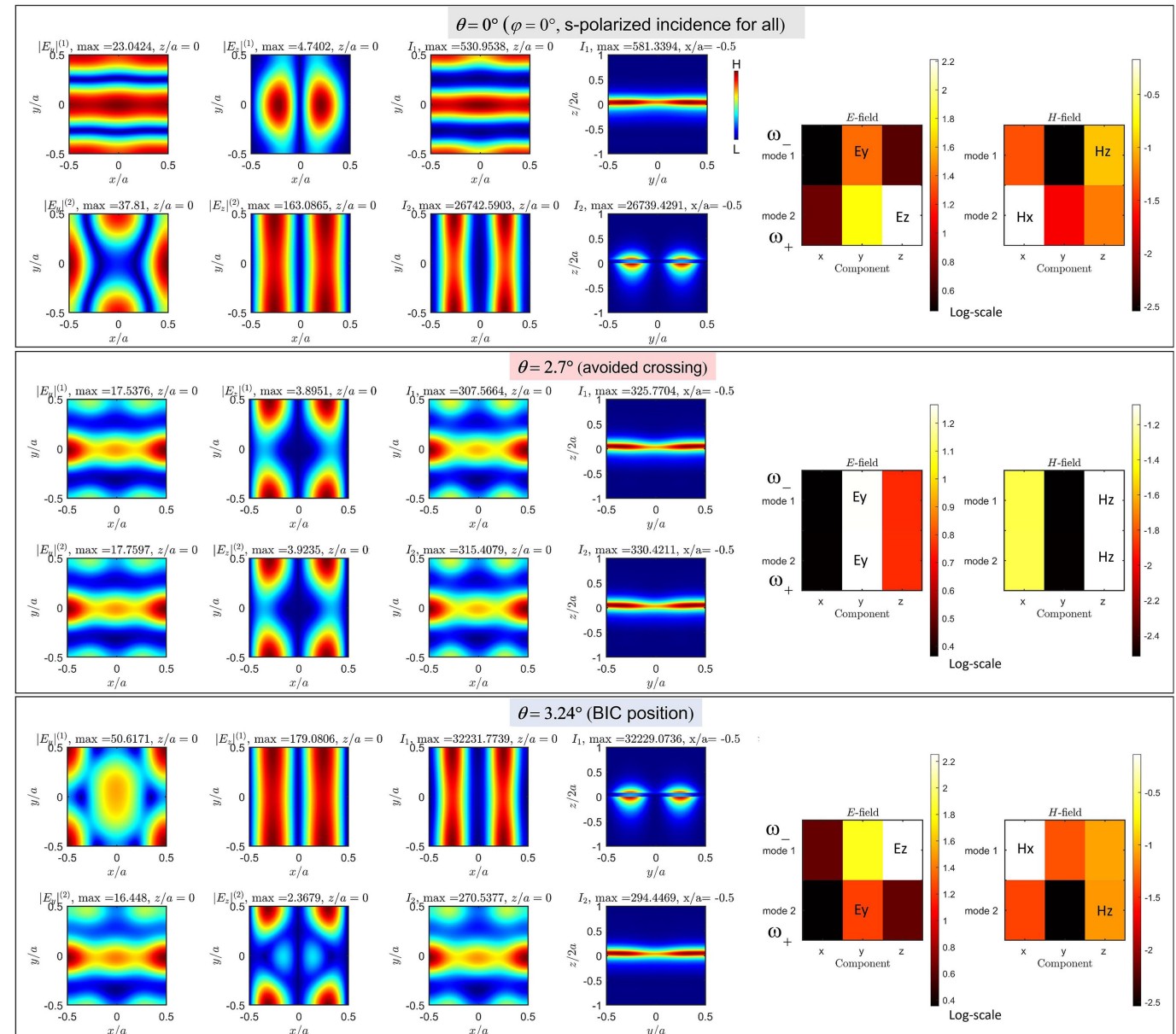

**Extended Data Fig. 4 | Mode field maps near the FW quasi-BIC.** Main components $E_y$, $E_z$ of the vector fields of mode 1 (dark, $\omega_-$) and mode 2 (bright, $\omega_+$) in the $xy$ plane at $z = 0$, together with the intensity map $I_1$, $I_2$ in the $yz$ cross-section for both modes 1 and 2 at the indicated incidence angle $\theta$, as shown in Extended Data Fig. 3c. The colour map of their components (amplitude in modulus, divided by $E_0 = 1\,V\,m^{-1}$, in logarithmic scale), on the right, indicates the change from TE ($E_y$) character to TM ($E_z$) character with varying momentum, $\mathbf{k} = k_o(\sin\theta\cos\phi, \sin\theta\sin\phi)$, at fixed $\phi$.

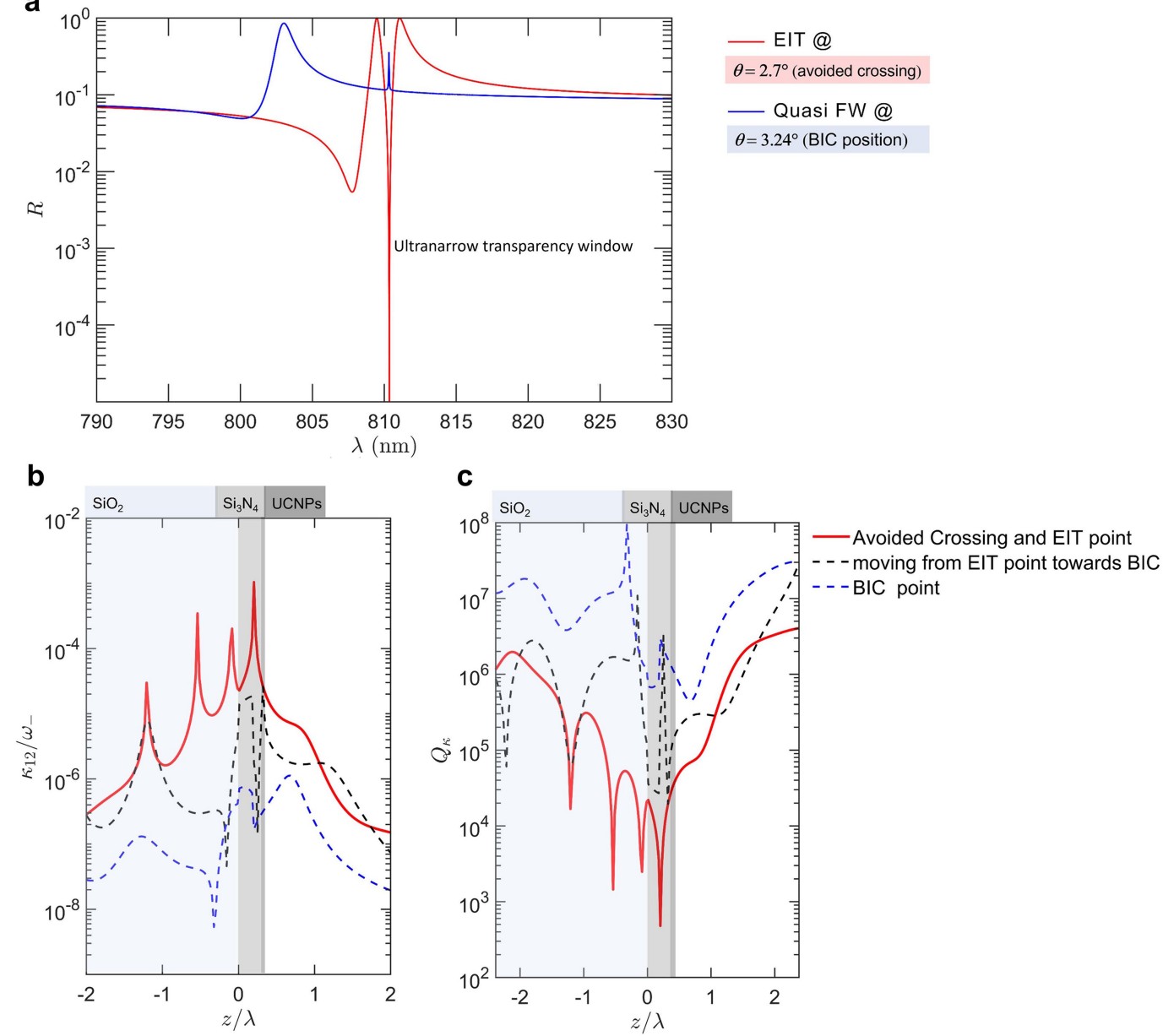

**Extended Data Fig. 5 | Coupled-resonance-induced transparency and near-field coupling dependence. a**, Spectral coincidence of the EIT transparency frequency at the avoided crossing with the FW quasi-BIC frequency for the angle mismatch <0.6°. **b**, Near-field coupling constant (real part) normalized to $\omega_- = 2\pi c/\lambda_{\text{mode-1}}$. **c**, associated quality-factor $Q_\kappa$ (as defined above) as a function of $z$ (normalized to $\lambda_{\text{mode-1}} = 810$ nm) along the $z$ axis and parameterized for $\theta$ from 2.7° to 3.24°.

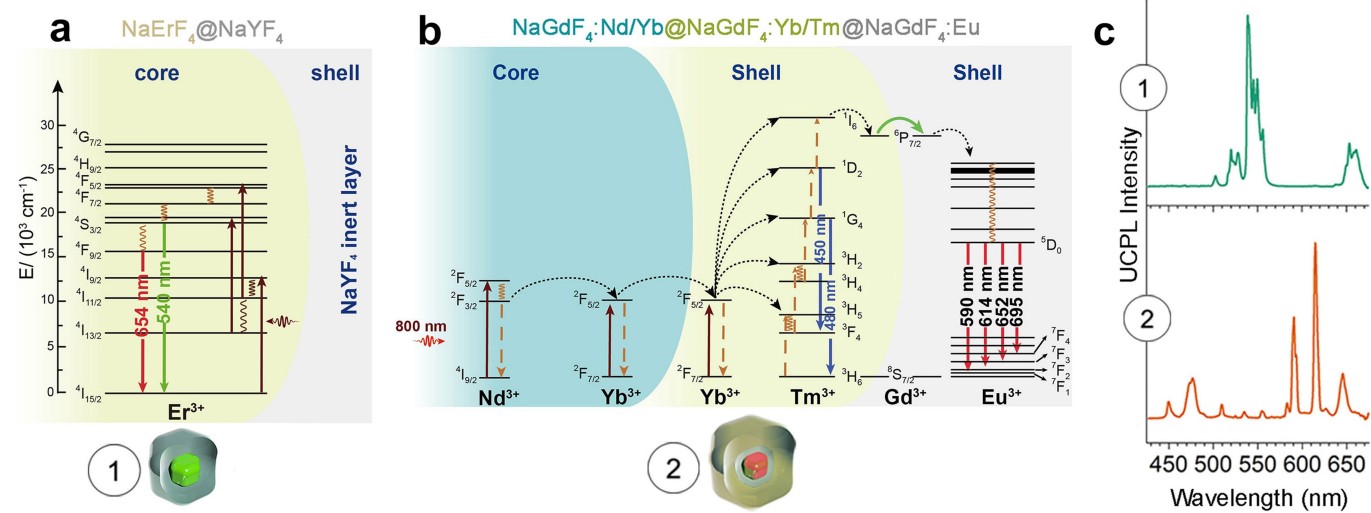

**Extended Data Fig. 6 | Emission lines of the upconversion NPs.** Energy-level scheme of core–shell NPs (**a**) and core–shell–shell NPs (**b**) and their corresponding experimental upconversion photoluminescence (UCPL) spectra (**c**).

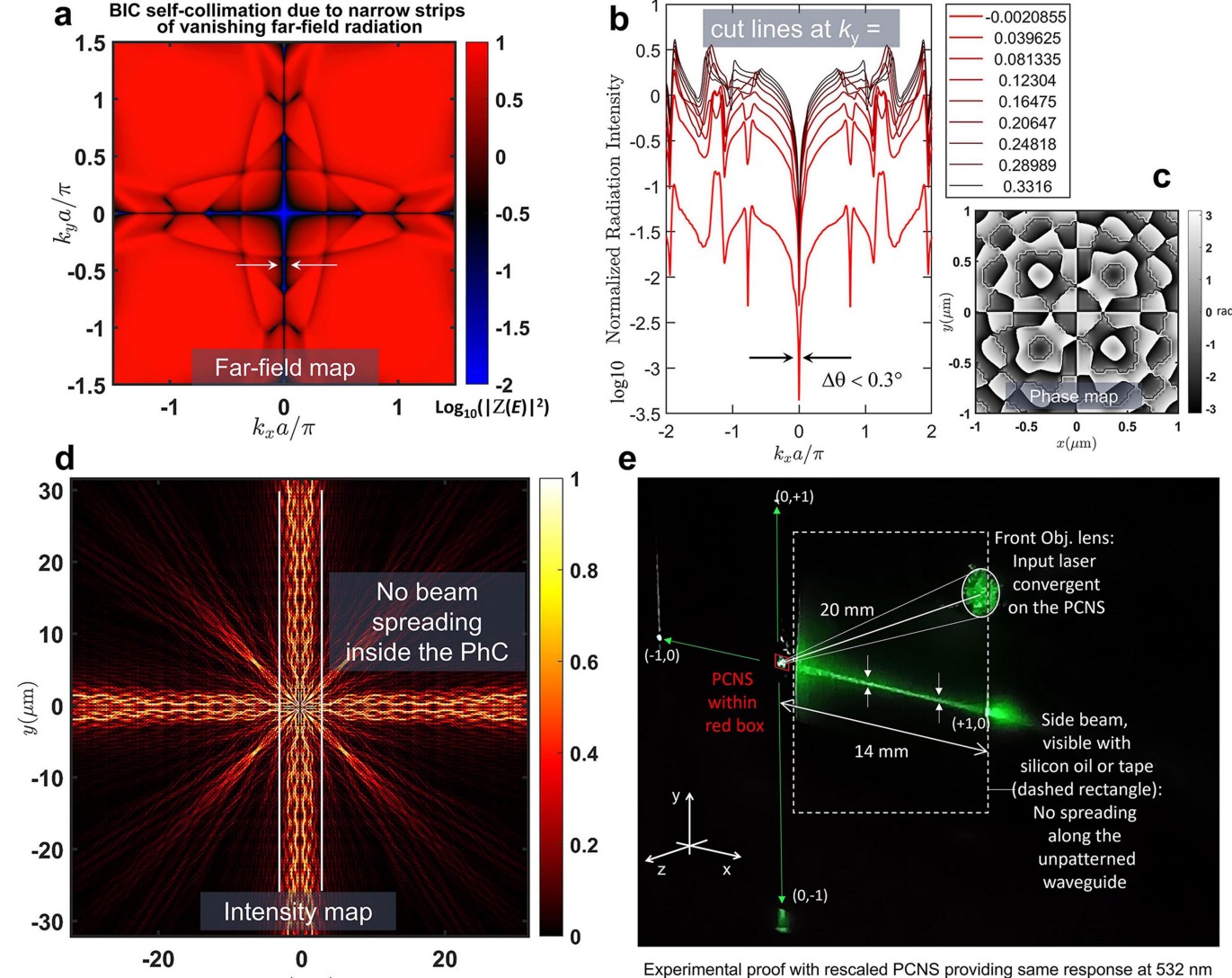

**Extended Data Fig. 7 | Details on self-collimation effect. a**, Isofrequency far-field intensity map in momentum space showing nontrivial vanishing strips. **b**, Cut lines of the *Z*-transform revealing a narrow divergence independent of the incident/outgoing wavevector along the orthogonal direction (cross of zeros). **c**, Phase map in the near field showing a phase vortex singularity. **d**, Corresponding near-field intensity map showing self-collimation. **e**, Experimental proof realized with a rescaled geometry supporting the same band structure at 532 nm: a 20× objective lens focuses the laser (SC, NKT Photonics, filtered at 532 nm) onto the patterned PCNS (red rectangle). The peculiar energy–momentum dispersion induces a resonant field that propagates in plane without spreading along the principal directions (±1, 0) and (0, ±1), in agreement with the simulation in **b**.

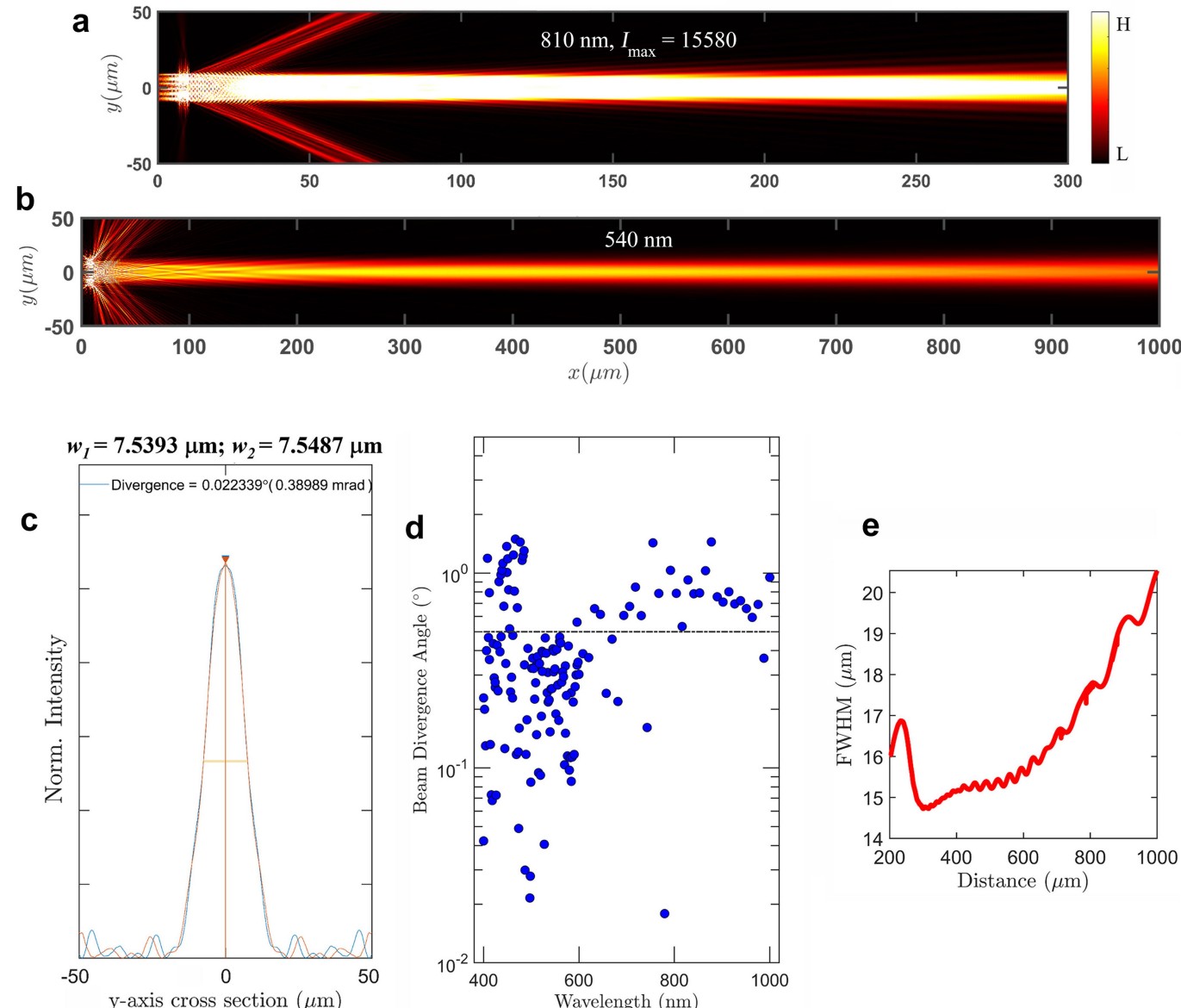

**Extended Data Fig. 8 | FDTD simulation with dipole sources aligned with the main component of the FW quasi-BIC field.** The dipoles cover a circular area of several microns squared (same size as the input spot focused with an objective lens) at the boundary of the PCNS with the uniform slab. Their emission is pumped by the maxima of the local optical field (shown in the experimental near-field map in Supplementary Fig. 3), but their fields also collectively add up coherently emitting radiation in the plane of the slab as triggered by the patterned geometry affecting their spatially correlated emission. **a**, Field propagates along the direction (+1, 0) when the right boundary is excited with intensity enhancement as large as $1.5 \times 10^4$ (normalized to the number of emitters) at the quasi FW-BIC wavelength (810 nm). **b**, At shorter wavelengths (540 nm), such as those produced with the experimental UCNP emitters, the spatially correlated fields produce narrow emission (calculated along 1 mm of propagation from the edge) and with beam divergence even of 0.02° as shown in **c**. **d**, The analysis over the whole visible and near-infrared spectrum revealed that the typical value of divergence was below 0.5°. **e**, The full width at half maximum of the beam shown in **b** changes along the propagation, obeying a mechanism of self-healing, which compensates the diffraction.

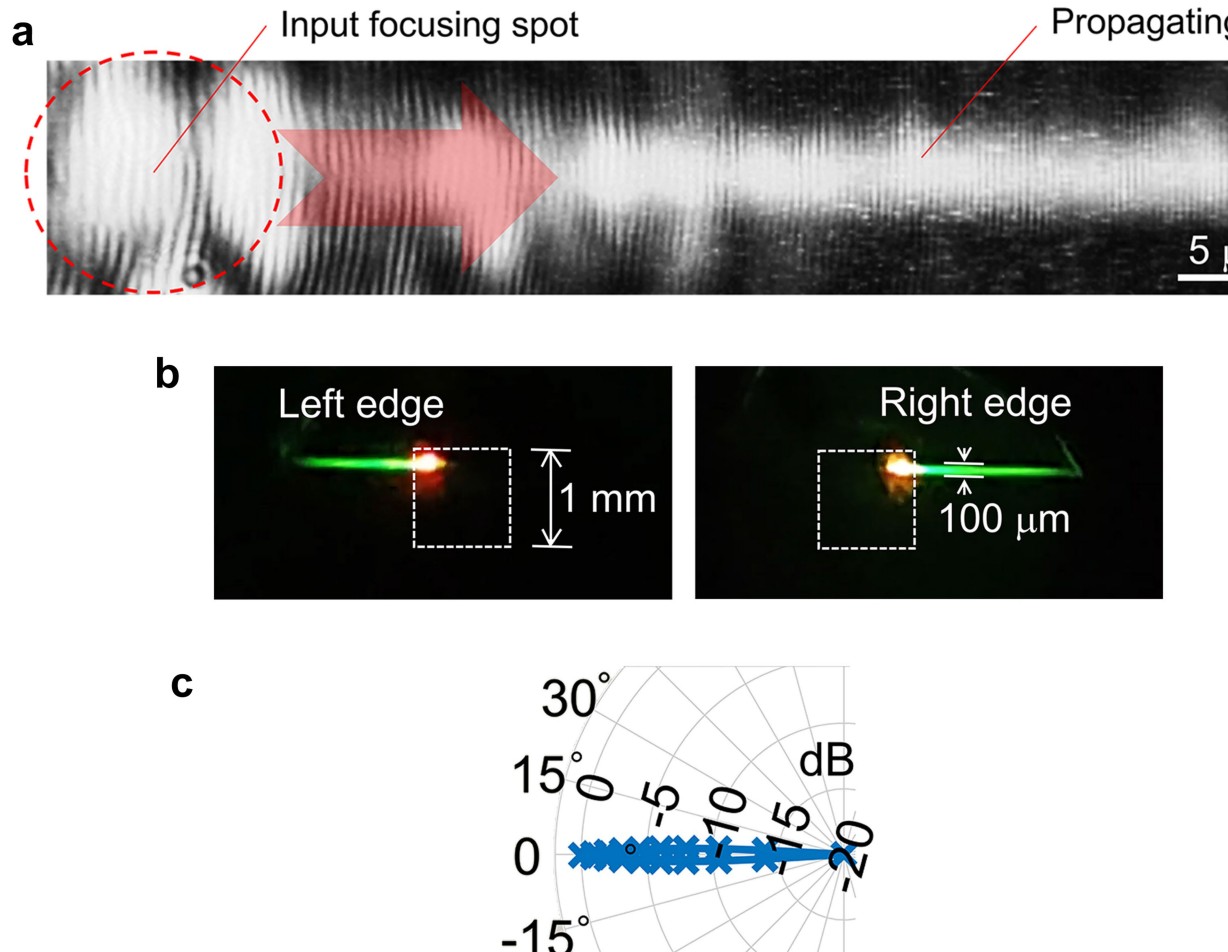

**Extended Data Fig. 9 | Directivity measurements. a**, Microscopy inspection of light propagation near the edge, at which the lateral size of the emitted light remains equal to the input beam waist. Notably, the output collimation is unaffected by the input focusing owing to the influence of the BIC point wavevector. The inset shows the interference pattern resulting from the spatial coherence of the emitted wave. **b**, Experimental side emission directed along the outer edge versor. **c**, Polar plot of the edge emission, in agreement with the simulation in Extended Data Fig. 8.