## [Peer Review File · Nature]

Manuscript Title: Directive Giant Upconversion by Supercritical Bound States in the Continuum

Reviewer Comments & Author Rebuttals

Reviewer Reports on the Initial Version:

Referees' comments:

Referee #1 (Remarks to the Author):

In the manuscript, "Steerable 10^8 -fold Enhanced Upconversion by Supercritical Edge Bound States in the Continuum", Chiara Schiattarella et al. investigate enhancement of coherent upconversion photoluminescence (UCPL) with resonant photonic crystal slabs (PCS) in the visible range. The authors demonstrate experimentally and numerically the enhancement of the UCPL from nanoparticles (NPs) by tuning the PCS mode resonant wavelength to the NPs absorption band. The main approach of enhancement is the engineering of mode coupling constant between a bound state in the continuum (BIC) and bright mode by moving the excitation spot to the structure edge. At the PCS edge the emission propagates in the free space as a highly nondivergent beam with the maximal achieved emission enhancement is 10^8 . The authors demonstrate steering of emission by 90 degrees by changing the input polarization from TE to TM. The authors claim to develop a new concept of supercritical coupling to explain their results theoretically, which is based on pumping of energy to the dark BIC via coupling it to the bright mode. They compare the developed concept with the analogue of electromagnetic induced transparency, explaining how the field enhancement can be maximized for the coupling constant comparable with the dark mode nonradiative linewidth.

Reasons to accept:

The idea of enhancement of UCPL with resonant dielectric structures integrated with active NPs is of high fundamental and practical significance for the field of photonics and optics of nanostructures. The experiment is solid and convincing, the measurements are well-performed. The reported enhancement value is impressive. The effect of directional control and steering is novel.

Reasons to reject:

The broad significance, importance and novelty are questionable, because the basic principle of UCPL enhancement with photonic nanostructures was demonstrated earlier. The developed setup has significant limitations, such as requirement of immersing the sample in a phase matching liquid (e.g., silicone oil). The narrow divergence of the resulting beam is not explained properly. The main claim of the paper related to the "new effect" of supercritical coupling is based on a misleading theoretical model which relies on invalid physical principles - the eigenmodes in open photonic structures do not interact with each other fundamentally, so the proposed energy exchange between a dark and bright mode cannot be realised, at least in the linear regime. Moreover, the model is oversimplified and neglects several key elements, that change the evolution of resonances dramatically.

Overall assessment:

Even though the experimental results (enhancement value and steering effect) are impressive, the paper makes the impression of a "proof-of-the-concept" work rather than a broad impact study of potentially high interest to people from distinct physical communities. Importantly, the experimental data lack any statistical analysis so a limited assessment of the robustness and reliability of the results can be made. The limitations of this work are critical, and I conclude that the manuscript lacks significance, technical quality and statistical tests that would make it acceptable for publication in Nature. Below, I outline a detailed list of comments and criticisms.

Major comments.

(1) The core idea of the manuscript is based on the developed concept of supercritical coupling. In this proposed concept the authors suggest a model of two interacting modes via the near-field coupling with constant κ . One of the modes, called Friedrich-Wintgen BIC (FW-BIC), is dark and has only nonradiative losses. The other mode is bright and has radiative losses as well. In the developed model the pump energy is coupled to the bright mode, then the energy is transferred to the FW-BIC via the near field coupling κ . The major limitation of the developed model is that it is not applicable to the description of the eigenmodes of periodic photonic structures such as a dielectric PCS. More specific, the equations (44,45) in Supporting Information are not applicable to the description of interaction a FW-BIC and a bright mode. The reason is that the true modes of the PCS are orthogonal solutions of the non-Hermitian Hamiltonian and do not interact neither in the near-field, nor in the far-field [see Ref. R1]. This means, that the correct equations for the FW-BIC and bright mode are Eqs.44,45 that do not include the interaction constant κ , so the pumping of the bright mode cannot affect the dark mode because there no energy exchange between them. One of the ways to induce interactions between a bright and a dark mode is the Kerr nonlinearity, this case was studied in detail in Ref. R2, but it requires significant pumping powers close to the damage threshold of the material.

(1a) The Friedrich-Wintgen model can be used for description of FW-BIC formation and explanation of other physical effects in periodic photonic structures, but using a basis of guided mode resonances only. The modes of the PCS can be treated as a superposition of guided resonances (GRs) of the effective waveguide (see Refs. R2-R4) coupled due to the periodic perturbation. In the simplest approximation, the low-frequency fundamental modes of the PCS are a combination of two degenerate fundamental GRs of the effective dielectric slab at the edges of the Brillouin zone propagating in the opposite directions. In this case, the coupling between modes occurs due to the near-field interaction κ due to periodic perturbation and far-field interaction due to their coupling to the open radiation channel via the periodic perturbation. This model is a direct analogue of the Friedrich-Wintgen (FW-model) quantum model discussed by the authors in Supporting Information, but importantly, that it is applied to bound GRs of the effective dielectric slab. By diagonalising the effective Hamiltonian of the FW-model, modes of the open structure can be achieved and under some circumstances a photonic FW-BIC can form (see Refs. R5). The actual physics of formation of FW-BICs is more complicated and a strict non-Hermitian model is required (see Ref. R6). Again, after diagonalisation the modes are not interacting, so the energy transfer between them cannot be achieved.

I also comment of the comparison with the analogue of the electromagnetically induced transparency in dielectric nanostructured arrays. The authors compare their model to Ref. [25], however, it studies a different system: there two photonic lattices are superimposed to form a new lattice. One of the basis lattices contains a dark BIC mode, the other - bright dipolar mode. Then, the FW-model can be applied to them, because the modes of the resulting structure with two elements in the unit cell are not BICs. The lattice modes interact so the true eigenmodes of the final structure will have finite radiation losses. In this manuscript, the authors start with the description of the true modes of the PCS as the basis modes with an interaction which is different.

(1b) Even though the model used by the authors is incorrect, they did not include the far-field coupling of the modes via the continuum to the Eqs. (8-9) of the Supporting Information. This term is crucial for the achievement of BIC regime.

(1c) In addition to the non-radiative losses, the experimental characterization also impacts the observed Q-factors. So, in addition to " Q_a " there should also be a " $Q_{excitation}$ ". The authors describe that the excitation properties affect the Q factor but it is not included in the model. When a small area is excited, highly localized modes will suffer less compared to less localized modes. The observed Q-factor is dependent on the excitation size, and each mode will have a different dependence.

(1d) While conceptually appealing, the authors use one constant " Q_a " to describe non-radiative losses of both modes. The modal profiles of each of these modes are each different. This means the impact of losses varies from mode to mode, and that a target Q for one mode should be higher than the target Q of another. For example, some modes may be more sensitive to sidewall roughness, while others more sensitive to variations from unit-cell to unit-cell at the sharp corners. Roughness at the top versus sides may also differ between quasi-TE and quasi-TM modes.

(1e) At the structure edge, the approximation of a periodic photonic crystal is not applicable, and modes are radiating to the lateral direction. In this sense, the concept of BICs is not clear - the radiation normal to the surface plane can be prohibited by the radiation from the sides of the structure can be strong. Also, the concept of one radiation channel is not applicable so the FW-model cannot be used for description of the effect.

(2) The narrowband spatial divergence of the UCPL is not explained in the paper. The authors mention that it can be related to the narrowband behavior of BICs in the k-space, but it is confusing because the emission is not at the BIC wavelength thus it is not directly related to the spatial distribution of the BIC. Moreover, the BICs are narrowband in the quasi-normal direction of the k-space (5 degrees) while none of that can be said about the lateral direction.

(3) The authors use a phase matching liquid (silicone oil) on top of the sample to achieve the up-down inversion symmetry of the structures which is a requirement for FW-BICs. This geometry seems highly impractical, while without this liquid the FW-BIC cannot form.

(4) More detailed comparison with earlier reports on the enhancement of UCPL with dielectric and plasmonic nanostructures is required to outline the scope of the current results within the existing

literature.

(5) The experimental results lack any statistical analysis. Statistical tests with corresponding error bars and probability values are required to confirm robustness, validity, reliability of the results.

Minor comments.

(6) The authors employ a FW-BIC to realize the effect, however, no explanation is given why symmetry protected BICs are not applicable. For symmetry protected BICs the phase matching liquid could be not necessary.

(7) It is unclear how the angle of incidence required for excitation of the FW-BIC is related to the emission directionality and properties.

References.

- [R1] Sauvan, C., Wu, T., Zarouf, R., Muljarov, E.A. and Lalanne, P., 2022. Normalization, orthogonality, and completeness of quasinormal modes of open systems: the case of electromagnetism. *Optics Express*, 30(5), pp.6846-6885.
- [R2] Fan, S. and Joannopoulos, J.D., 2002. Analysis of guided resonances in photonic crystal slabs. *Physical Review B*, 65(23), p.235112.
- [R3] Tikhodeev, S.G., Yablonskii, A.L., Muljarov, E.A., Gippius, N.A. and Ishihara, T., 2002. Quasiguidded modes and optical properties of photonic crystal slabs. *Physical Review B*, 66(4), p.045102.
- [R4] Andreani, L.C. and Gerace, D., 2006. Photonic-crystal slabs with a triangular lattice of triangular holes investigated using a guided-mode expansion method. *Physical Review B*, 73(23), p.235114.
- [R5] Hsu, C.W., Zhen, B., Stone, A.D., Joannopoulos, J.D. and Soljačić, M., 2016. Bound states in the continuum. *Nature Reviews Materials*, 1(9), pp.1-13.
- [R6] Neale, S. and Muljarov, E.A., 2021. Accidental and symmetry-protected bound states in the continuum in a photonic-crystal slab: a resonant-state expansion study. *Physical Review B*, 103(15), p.155112.

Referee #2 (Remarks to the Author):

The manuscript demonstrates the experimental demonstration of a coherent up-conversion luminescence enhanced by 8 orders of magnitude. The emission is characterized by negligible divergence, microscale width, and direction-steering capability controlled by input focusing and polarization.

BIC-based devices are drawing great interest in recent years and generally appeal to a broad readership. However, I don't believe the work has the impact and significance necessary to be published in Nature. Furthermore, one of the authors published similar work in 2019 in Nature Communications (Up conversion amplification through dielectric superlensing modulation <https://doi.org/10.1038/s41467-019-09345-0>) with an efficiency up-conversion of five orders of

magnitude. A demonstration from 5 orders to 8 orders is not significant enough to be considered for Nature Journal. After addressing my comments, I recommend the authors submit the paper to Journal like Nano letters.

1. Is the term “supercritical coupling” an existing term in literature, or introduced by the authors? References should be cited to introduce the concept and/or the context. It appears that supercritical coupling here actually refers to the regime where κ_{12} is sufficiently small, and it follows that “under this condition the low-Q mode can funnel energy into the high-Q BIC mode via near-field interaction”: why does energy funneling happen when mutual coupling is weak? The text is not clear and lacks backup and/or reference.

2. It will be very interesting to present the state-of-the-art coherent up-conversion luminescence and compare them to this work.

3. I am curious to see how the authors could compare this work to a Valley topological laser where a selective excitation of the corners can be implemented via directional evanescent coupling with a spin-polarized source without high Q :

1. Room-temperature lasing from nanophotonic topological cavities. Light Sci Appl 9, 127 (2020).

<https://doi.org/10.1038/s41377-020-00350-3>

2. Topological Insulator Laser Using Valley-Hall Photonic Crystals,

<https://doi.org/10.1021/acsp Photonics.0c00521>

4. Regarding a key finding of this work, i.e., reduced coupling κ_{12} at the edge, and the resultant high-Q mode, I am not convinced by the corresponding numerical analysis and supporting text (Fig. S3 and Supplementary Section 1.5). Firstly, what happens at the edge should be non-uniform, but the authors seemingly used a uniform periodic simulation to emulate the edge effects, only changing the effective material index. This simulation is oversimplified to me. Secondly, the quantity of interest here is the coupling coefficient κ_{12} (we need to see it decrease at the edge). The authors gave the equation for it but never seemed to have rigorously evaluated it, which requires finding the fields rigorously in the first place (I agree one can qualitatively see the reduced coupling in Fig. S3 but again this is emulated). Lastly, explanation is only qualitative: “Indeed, the FW-BIC is a quadrupole mode” (OK, I have to scroll to Fig. S6c to see that), “but it gains a perturbation-like dipolar character at the boundary because of the broken symmetry of the system at the edge”, (where is the data?). And it follows “At the same time, the bright wave (dipolar mode) gains a quadrupole character at the finite boundary”. (How do you back this up?)

5. The interpretation of the high directionality of the emission is very sloppy. As the paper stated in itself, compared to the actual paradigm of supercollimation (Ref. 48, let us leave aside whether this analogy is valid), here “the high directionality and collimation was observed in visible upconverted radiation, not only in the pumping mode” (Line 237). Now how can the directionality of the emission wave be attributed to k-vector selectivity of the pumping mode? The emitting photonic mode at the up-converted wavelength should be investigated and elaborated as well, at least numerically.

6. The up conversion here is a multi-photon process (Line 109). I am confused by how it exhibits

linear behavior ($s \sim 1$ in Fig. 2d and 3c)? The authors attempted to explain this by stating that with the strong excitation here single-photon absorption readily occurs (Line 143), but how does single-photon absorption contribute to the signal measured at the up-converted wavelength? This should be better explained.

7. The enhancement effect here is localized at the edge. I am wondering how the size of array affects the concept:

a. In the center-illuminated case (Fig. 2), the authors extracted QFW-BIC=5240. Noting this comes from a 1 mm² array, how does this compare with theoretical limit assuming infinite array, as is the case for the periodic unit cell simulation, for which the Q is not explicitly given? Also the paper says this Q is “in excellent superposition with the nanoparticle absorption centered at 800 nm” (Line 134), where is this calculation performed?

b. Now in the edge-illuminated case (Fig. 3), would the size of the array affect, as the mode does not address the full area of the array? Can a similar level of enhancement be achieved with a small pattern? Numerical results may be used to illustrate this, and this discussion may help clarify the advantage of this work (if small pattern works equally well in this scheme).

8. The paragraph associated with equation (3) is very misleading and tedious as the main text of a candidate paper for consideration of publication in Nature. After a lot of redundancy [Line 185 – 200, beginning “Let us now evaluate the expected enhancement...”, while the finding, i.e. equation (3) is not meant to be what one expected], it only arrives at the trivial conclusion that equation (1) underestimates the enhancement factor, and one should use equation (2) [evaluated as equation (4)], which fits the experimental finding better. Yes, I agree that this comparison hints the nature of the resonance, but it can be at best mentioned briefly in one line and details should be given in the Supplementary.

9. The analogy of electromagnetically induced transparency should be a very important aspect as it appears in the abstract and is referred to multiple times, however, the main text does not present anything in that regard but simply redirects to the Supplementary. At least qualitative explanation should be given in the main text to inform the reader of why this analogy is necessary and relevant to the development of this concept presented here.

10. “Beam steering” does not seem to be an accurate word for describing the effect in Fig. 4. The emission comes out in two discrete channels on two edges, and the angle with respect to each edge is 90 degrees fixed.

11. Other unclear figures, text and inappropriate referencing:

a. Fig. 2 b and c are very unclear. The arrows for input laser should extend close to the PCNS to show the reader how in one case the PCNS is illuminated while the other case the bulk substrate is illuminated. The words IN and OUT in both b and c misleadingly imply “input” and “output” intuitively. The three colors in c imply three measurements are performed, but actually only two - green and orange are from the same measurement only indicating different spectral lines.

b. In contrast to b, the main text includes technical details that fit better as methods or Supplementary, such as the fact that the thickness δh used in the simulation (Supplementary Fig. S6) is consistent with the experiment (main text Line 123).

c. Lengthy and confusing abbreviations: UCPL, UCNP, and PCNS.

d. Through Lines 185-220, too many Q's with different subscripts and different numbers were introduced, which is very confusing and does not help with the understanding. All the reader sees are numbers.

e. 18 references on up conversion nanoparticles were cited (Refs. 26-43), out of which many may be unnecessary, e.g. Refs. 26-30 for multi-photon absorption through long-lived intermediate energy states, Refs. 38-40 for long-lived imaging probes, Refs 33-36 for lasing.

Referee #3 (Remarks to the Author):

The manuscript presented by Zito et al is certainly solid and interesting with new knowledge presented and potentially broader impact to the materials, photonics and physics community. The major merits include:

1. Reveal the existence and the key role of supercritical coupling in maximizing energy accumulation.
2. Achieve Friedrich-Wintgen BIC under supercritical coupling with significant local optical field enhancement
3. Get 8 orders of magnitude upconversion enhancement through supercritical edge bound states.
4. Achieve the controlled upconversion emission with negligible divergence, microscale width, and direction-steering capability.

Suggested improvements:

1. Whether the excitation wavelength is tunable by the structure design of the photonic crystal nanoslab (PCNS) in this system? This is the key to the broad usage of this technology.
2. Whether the emission enhancement is wavelength-dependent? It seems all the upconversion emission data is from the NaErF₄@NaYF₄ core-shell nanocrystals and only the green emission was inspected (figure 3a). It is better to collect the data for the red emission, which have a higher intensity than the green emission. Also, there are different kinds of emissions bands for core@shell@shell materials with the energy transfer of Nd→Yb→Tm→Gd→Eu (Figure S5E), which could be an ideal model to study the emission-wavelength-dependent upconversion enhancement. There is detailed information for the NaGdF₄:Nd/Yb(40/5%)@NaGdF₄:Yb/Tm(49/1%)@NaGdF₄:Eu(15%) core-shell-shell nanocrystals. The authors should add their detailed spectra information and the upconversion enhancement at different emission wavelengths in PCNS.

3. Whether the orientation of the UCNPs on the PCNS affects the direction-steering performance?
4. Why were two layers of nanoparticles coated on the PCNS surface? Single-layer nanoparticles are commonly used in plasmonic enhancement (Wu et al. *Nature Nanotechnology*, 1110–1115, 2019) and the fabrication of micro-laser (Shang et al. *Nature Communications* 6156, 2020). Therefore, single-layer nanoparticles may have a better performance here.
5. In Figure S5B, the optical absorption spectrum should be from NaErF₄@NaYF₄ core-shell nanocrystals instead of the core-shell-shell NPs. The authors may double-check the TEM images and other related characterizations.
6. It is better to provide the power density (W/cm²) instead of the power intensity (mW) of the laser.
7. It is better for the authors to add TEM images of the core and core@shell nanoparticles. This information is important for the readers to repeat related experiments.
8. On page 6, the author claimed that “In colloidal samples we measured $s = 1.2$ at an incident power on the order of 500 mW, while we found $s = 1.2$ at 1 mW on the PCNS.” It is better to have related data in the supporting information.

In closing, this is an excellent report with good experimental data. The data support the main point of the paper, in the sense that through supercritical coupling in PCNS it is possible to significantly enhance upconversion luminescence with direction-steering capability. The rational design and experimental validation of the supercritical coupling for emission enhancement in this work represents a sizeable breakthrough that justifies the publication in *Nature*.

<Responses to the reviewers' comments (Nature manuscript 2022-10-16862)

Title: Directive 10^8 -fold Enhanced Upconversion by Supercritical Edge Bound States in the Continuum

Changes in the revised manuscript as a response to the reviewers' comments are highlighted in red color and clarifications regarding the reviewer's comments are provided in blue color.

Referee #1 (Remarks to the Author):

In the manuscript, "Steerable 10^8 -fold Enhanced Upconversion by Supercritical Edge Bound States in the Continuum", Chiara Schiattarella et al. investigate enhancement of coherent upconversion photoluminescence (UCPL) with resonant photonic crystal slabs (PCS) in the visible range. The authors demonstrate experimentally and numerically the enhancement of the UCPL from nanoparticles (NPs) by tuning the PCS mode resonant wavelength to the NPs absorption band. The main approach of enhancement is the engineering of mode coupling constant between a bound state in the continuum (BIC) and bright mode by moving the excitation spot to the structure edge. At the PCS edge the emission propagates in the free space as a highly nondivergent beam with the maximal achieved emission enhancement is 10^8 . The authors demonstrate steering of emission by 90 degrees by changing the input polarization from TE to TM. The authors claim to develop a new concept of supercritical coupling to explain their results theoretically, which is based on pumping of energy to the dark BIC via coupling it to the bright mode. They compare the developed concept with the analogue of electromagnetic induced transparency, explaining how the field enhancement can be maximized for the coupling constant comparable with the dark mode nonradiative linewidth.

Reasons to accept: The idea of enhancement of UCPL with resonant dielectric structures integrated with active NPs is of high fundamental and practical significance for the field of photonics and optics of nanostructures. The experiment is solid and convincing, the measurements are well-performed. The reported enhancement value is impressive. The effect of directional control and steering is novel.

Reasons to reject:

- 1) The broad significance, importance and novelty are questionable, because the basic principle of UCPL enhancement with photonic nanostructures was demonstrated earlier.
- 2) The developed setup has significant limitations, such as requirement of immersing the sample in a phase matching liquid (e.g., silicone oil).
- 3) The narrow divergence of the resulting beam is not explained properly.
- 4) The main claim of the paper related to the "new effect" of supercritical coupling is based on a misleading theoretical model which relies on invalid physical principles - the eigenmodes in open photonic structures do not interact with each other fundamentally, so the proposed energy exchange between a dark and bright mode cannot be realised, at least in the linear regime.
- 5) Moreover, the model is oversimplified and neglects several key elements, that change the evolution of resonances dramatically.

We would like to express our gratitude for the time and effort the reviewer has invested in providing us with a detailed and comprehensive review of our manuscript. We greatly appreciate the reviewer's acknowledgement of the high fundamental significance of our approach for upconversion enhancement and the novelty of the directional control and steering effect. The reviewer's thoughtful

comments are invaluable in improving the quality and significance of our paper in this revised version. In response to the specific points raised by the reviewer, we would like to provide a brief rebuttal below, with more detailed responses provided in the subsequent point-by-point comments:

- 1) Our work demonstrates that a Friedrich-Wintgen BIC can be used to achieve upconversion coherent emission beyond the current state-of-the-art, challenging conventional beliefs about coupling with such modes. We are not aware of any parallel in the literature regarding the coupled-mode enhancement between a bright and a FW quasi-BIC dark mode, nor for achieving such high values of upconversion enhancement and coherent directional emission. We have included a comparison of upconversion enhancement based on previous photonic structures in the revised manuscript [**see response to specific *Comment #4***].
- 2) We would like to clarify that the structure only requires vertical asymmetry, and the use of silicone oil primarily served as a means to make in-plane radiation more visible or to fine-tune the resonance position, depending on the specific sample. [**see response to specific *Comment #3***].
- 3) We have thoroughly discussed the narrow beam divergence in the revised manuscript, providing new simulations and experiments to support our findings [**see response to specific *Comment #2***].
- 4) At the suggestions of the reviewer, we have included a detailed description of the mathematical framework based on temporal-coupled mode theory (TCMT) or the non-Hermitian Hamiltonian formalism. We have also validated our findings through explicit calculations using rigorous coupled wave analysis (RCWA). We discuss the mode coupling that gives rise to the Friedrich-Wintgen BIC and demonstrate the nonorthogonality of the modes near the ideal FW-BIC. The same formalism applies to the coupled-resonance induced transparency (EIT) in the subradiant-superradiant diagonal representation of the decay matrix. This demonstrates that our system is tailored to sustain a FW quasi-BIC in the energy-momentum space, with a small phase mismatch with the EIT point. We have developed a supercritical enhancement equation based on this tailored combination, leading to unprecedented field enhancement. The potential of the new effect of supercritical coupling is highlighted, enabling the superbright upconversion emission observed in our experiments [**see response to specific *Comment #1a-1e***].
- 5) We have addressed all critical issues about the model assumptions and have verified them through RCWA simulations of our specific nanostructure [**see response to specific *Comment #1d-1e***].

Overall assessment: Even though the experimental results (enhancement value and steering effect) are impressive, the paper makes the impression of a "proof-of-the-concept" work rather than a broad impact study of potentially high interest to people from distinct physical communities. Importantly, the experimental data lack any statistical analysis so a limited assessment of the robustness and reliability of the results can be made. The limitations of this work are critical, and I conclude that the manuscript lacks significance, technical quality and statistical tests that would make it acceptable for publication in Nature. Below, I outline a detailed list of comments and criticisms.

We appreciate the reviewer's criticism and would like to take this opportunity to further clarify the merit of our work.

In this study, we introduce a fundamentally new concept called supercritical coupling and experimentally validate its potential to achieve record-breaking enhancement of photon upconversion with precise control over the direction of emission. Our work addresses two key challenges.

The first challenge relates to the fundamental limit to far-field pumping of bound states in the continuum (BICs). BICs have opened up possibilities for topological nanophotonics with ultrahigh radiative quality factors (Q_r up to 10^6), which have implications for quantum physics and information technology. However, as with other optical resonances, the near-field enhancement is limited by critical coupling, and in a dark mode, the enhancement approaches zero.

Previous solutions have relied on finite-by-construction Q_r values, typically on the order of 10^2 , **to satisfy the critical coupling condition** $Q_r = Q_a$ in metasurfaces with large nonradiative losses $Q_a = 10^2$ (see *Phys. Rev. Lett.* 2018, 121, 193903; *Science* 2020, 367, 288). This limitation significantly undermines the potential of high-Q resonators based on BICs.

In our work, we propose and validate a completely new concept: a supercritical coupling condition that can enhance the local optical field in any high-Q resonator. The local optical field enhancements can reach several orders of magnitude higher than those achieved through conventional critical coupling. We have revised the manuscript to emphasize the connection between the formalism of EIT and FW interference, providing a thorough demonstration of the supercritical enhancement equation derived from TCMT (**see response to specific Comment #1a-b**).

The second aspect of this work concerns the enhancement of photon upconversion. Achieving efficient photon upconversion in lanthanide-doped nanoparticles at low irradiance is not only a fundamental challenge but also crucial for numerous applications, such as super-resolution imaging to nanophotonics (see *Nature* 2017, 543, 229; *Nature* 2020, 587, 594; *Nature* 2021, 589, 230). Existing implementations based on surface passivation, surface plasmon coupling, photonic crystal engineering, and broadband sensitization have only achieved limited improvement in upconversion emission (~ 4 to 5 orders of magnitude enhancement). In our study, we experimentally demonstrate that the supercritical edge BIC enables an 8-order-of-magnitude increase in upconversion brightness, surpassing the limit associated with critical coupling by two orders of magnitude. In addition, we demonstrate the unique capability of controlling directive emission, which has broad implications for a vast scenario of microscale photonic chip applications, including light sources, information delivery, photodetection, and sensing (**see response to specific Comment #2**).

Regarding technical quality, we have improved our technical presentation, adhering to rigorous criteria about theoretical aspects and experimental methodologies. In terms of statistics, our previous version involved 17 samples with two completely different types of UCNPs, demonstrating consistent results. We have now expanded our evaluation to include an additional 24 samples, including different sizes, addressing the concerns raised by **reviewer #2**. The statistical analysis has been included in the revised Supplementary Materials, as detailed in **response to specific Comment #5**.

Major comments. Comment #1: The core idea of the manuscript is based on the developed concept of supercritical coupling. In this proposed concept the authors suggest a model of two interacting modes via the near-field coupling with constant κ . One of the modes, called Friedrich-Wintgen BIC (FW-BIC), is dark and has only nonradiative losses. The other mode is bright and has radiative losses as well. In the developed model the pump energy is coupled to the bright mode, then the energy is transferred to the FW-BIC via the near field coupling κ . The major limitation of the developed model is that it is not applicable to the description of the eigenmodes of periodic photonic structures such as a dielectric PCS. More specific, the equations (44,45) in Supporting Materials are not applicable to the description of interaction a FW-BIC and a bright mode. The reason is that the true modes of the PCS are orthogonal solutions of the non-Hermitian Hamiltonian and do not interact neither in the near-field, nor in the far-field [see Ref. R1]. This means, that the correct equations for the FW-BIC and bright mode are Eqs.44,45 that do not include the interaction constant κ , so the pumping of the bright mode cannot affect the dark mode because there no energy exchange between them. One of the ways to induce interactions between a bright and a dark mode is the Kerr nonlinearity, this case

was studied in detail in Ref. R2, but it requires significant pumping powers close to the damage threshold of the material.

(1a) The Friedrich-Wintgen model can be used for description of FW-BIC formation and explanation of other physical effects in periodic photonic structures, but using a basis of guided mode resonances only. The modes of the PCS can be treated as a superposition of guided resonances (GRs) of the effective waveguide (see Refs. R2-R4) coupled due to the periodic perturbation. In the simplest approximation, the low-frequency fundamental modes of the PCS are a combination of two degenerate fundamental GRs of the effective dielectric slab at the edges of the Brillouin zone propagating in the opposite directions. In this case, the coupling between modes occurs due to the near-field interaction κ due to periodic perturbation and far-field interaction due to their coupling to the open radiation channel via the periodic perturbation. This model is a direct analogue of the Friedrich-Wintgen (FW-model) quantum model discussed by the authors in Supporting Materials, but importantly, that it is applied to bound GRs of the effective dielectric slab. By diagonalising the effective Hamiltonian of the FW-model, modes of the open structure can be achieved and under some circumstances a photonic FW-BIC can form (see Refs. R5). The actual physics of formation of FW-BICs is more complicated and a strict non-Hermitian model is required (see Ref. R6). Again, after diagonalisation the modes are not interacting, so the energy transfer between them cannot be achieved.

We thank the reviewer for these insightful comments. In response to the reviewer's concerns, we have provided a comprehensive and thorough explanation, which includes a detailed description of the mathematical framework based on temporal-coupled mode theory (TCMT) and the associated non-Hermitian Hamiltonian formalism. Moreover, we have conducted explicit calculations using rigorous coupled wave analysis (RCWA) to validate our theory.

In our manuscript, we have provided a synthetic version in the **main text** (Section “**Proximity of Friedrich-Wintgen quasi-BIC and electromagnetically induced transparency in the energy-momentum space**”) and a more detailed description is reported in the **Supplementary Materials, Sections 1.2-1.3**. The Supplementary Materials also include numerical validations for the 3D vector hybridized TE- and TM-like modes with RCWA in **Supplementary Materials, Section 1.4**. We would like to emphasize that RCWA is widely recognized as the most reliable numerical methodology for accurately describing the real response of systems like ours. It incorporates the full complex refractive index dispersion and provides 3D vector results. Moreover, RCWA has been shown to match the experimental band diagram in terms of frequency, linewidth, and asymmetric lineshape profile. The results obtained through our RCWA simulations are in excellent agreement with the TCMT model as well as experimental observations.

To address the reviewer's specific comment regarding **the interaction of true modes of the photonic crystal slab (PCS)**, we would like to clarify the following points. In our revised version (**Supplementary Section 1.2, pp. 4-9**), we have considered the case of coupled resonant modes using the non-Hermitian Hamiltonian formalism of two-wave coupled-mode theory of Friedrich-Wintgen BIC formation. It is important to note that orthogonal eigenvectors exist only when the number of radiative channels is larger than or equal to the number of resonances in the non-Hermitian resonator [*W. Suh, Z. Wang, S. Fan, IEEE Journal of Quantum Electronics* **40**, 1511 (2004)]. In addition, the formation of FW-BICs due to interfering nonorthogonal vector TE and TM modes in photonic crystal slabs was already demonstrated in 2000 by *P. Paddon and J. F. Young [Physical Review B* **61**, 2090 (2000)].

In our revised version (**Supplementary Section 1.2, pp. 4-9**), our analysis shows that as the asymptotic condition of FW-BIC can only ideally be reached, in the proximity of the FW-BIC, the FW quasi-BIC with suppressed linewidth is due to interference of modes that are **nonorthogonal**.

It is worth noting that the condition for FW-BIC formation is derived after diagonalization of the full Hamiltonian. However, as far as a single independent radiation channel exists for two eigenmodes (a necessary condition for a FW-BIC), moving away from \mathbf{k}_{BIC} , even **arbitrarily close** to this asymptotic point, **both modes acquire a nonzero radiative decay and are coupled to the same radiation channel**. Therefore, they must be nonorthogonal [*IEEE Journal of Quantum Electronics* **40**, 1511 (2004)]. This near-field coupling can be observed in the diagonal representation of the decay matrix, also known as the subradiant-superradiant model [*Nano Lett.* **14**, 2783–2788 (2014)].

While the ideal FW-BIC point is orthogonal to the final bright mode as required by the simultaneous diagonalization of both energy and radiative loss matrices, it is crucial to acknowledge that this point is an asymptote that is never reached in any real system (**Supplementary Materials Section 1.3**).

“To develop what we term supercritical enhancement equation, we start from the non-Hermitian Hamiltonian of coupled waves [4,10]. By generalizing Eq.(2), the dynamic equations for resonance amplitudes can be written in the following form:

$$\frac{d\mathbf{A}}{dt} = (j\hat{\Omega} - \hat{\Gamma}_r - \hat{\Gamma}_a)\mathbf{A} + \hat{K}_i^T \mathbf{s}^+, \quad (8)$$

$$\mathbf{s}^- = \hat{C}\mathbf{s}^+ + \hat{D}\mathbf{A}, \quad (9)$$

where both $\hat{\Omega}$ and $\hat{\Gamma}_r$ matrices are Hermitian matrices representing the resonance frequencies and the radiation decay, respectively. On the other hand, $\hat{\Gamma}_a$ represents nonradiative losses and is initially set to zero $\hat{\Gamma}_a = 0$ to isolate the radiative rate associated with an ideal BIC. The resonant mode is excited by the incoming far field waves \mathbf{s}^+ coupled to the resonator with coefficients indicated by \hat{K}_i . The outgoing waves \mathbf{s}^- depend on the direct scattering channel \hat{C} and the resonant modes \mathbf{A} via the decay port coefficients in \hat{D} . Energy conservation and time reversal symmetry imply that $\hat{K}_i = \hat{D}$ and that the coupling with the port is linked with the radiation loss so that $\hat{D}^\dagger \hat{D} = 2\hat{\Gamma}_r$, which determine the elements of \hat{K}_i^T and also imply that $\text{rank}(\hat{\Gamma}_r) = \text{rank}(\hat{D}^\dagger \hat{D}) = \text{rank}(\hat{D})$. In addition, $\hat{D} = -\hat{C}\hat{D}^*$. Let be $\mathbf{A} = (A_1, A_2)^T$, i.e. a system of two resonances of frequency ω_1 and ω_2 having radiative lifetime $\tau_{r1} = 1/\gamma_{r1}$ and $\tau_{r2} = 1/\gamma_{r2}$. Moreover, both resonances may experience absorption loss, characterized by $1/\tau_a = 1/\gamma_a$. It is important to note that for the specific case of the avoided crossing point, the absorption for both modes is exactly the same, as we demonstrated below. Then in general $\gamma_{1,2} = \gamma_{r1,r2} + \gamma_a$, but for now let turn $\gamma_a = 0$.

Recall that the modes of the resonator are defined as the eigenmodes of the non-Hermitian Hamiltonian operator $\hat{H} = j\hat{\Omega} - \hat{\Gamma}_r$ (neglecting nonradiative loss). Only Hermitian matrices allow for a diagonal representation with orthogonal eigenvectors, whereas non-Hermitian matrices may have linearly dependent or linearly independent but non-orthogonal eigenvectors, or they may have orthogonal eigenvectors depending on specific properties such as parity-time symmetry. It is important to note that the Hamiltonian and its eigenvalues are functions of the in-plane momentum $\mathbf{k} = k_o(\sin \theta \cos \phi, \sin \theta \sin \phi)$, meaning that eigenvalues and eigenvectors of the non-Hermitian Hamiltonian evolve in momentum space. Suh et al. demonstrated that the eigenvectors of the non-Hermitian Hamiltonian, which represents the modes in the open resonator system, will always be nonorthogonal when the total number of *independent* decay ports (open channels) is less than the number of optical modes, and both modes are coupled to the decay ports [10]. The crucial concept here is the *independent* decay ports, which are related to the sharing of the vertical symmetry of the

modes. In the case of evolving TE and TM-like modes, the inversion of their character at the avoided crossing can occur at any point in the energy-momentum space. We know that the eigenmodes of a matrix form an orthogonal basis if and only if $\widehat{H}^+\widehat{H} = \widehat{H}\widehat{H}^+$. Since both $\widehat{\Omega}$ and $\widehat{\Gamma}_r$ are Hermitian, this is equivalent to the relation $\widehat{\Omega}\widehat{\Gamma}_r = \widehat{\Gamma}_r\widehat{\Omega}$, which implies that $\widehat{\Omega}$ and $\widehat{\Gamma}_r$ can be simultaneously diagonalized. When two eigenmodes and a single independent radiation channel is considered, $rank(\widehat{\Gamma}_r) = 1$, one of the orthogonal eigenmodes of the matrix will have a pure imaginary eigenvalue. This indicates that one of the two modes has an infinite lifetime (BIC) and does not couple to the decay port. As a nonzero coupling with the single decay port exists, the two modes in the resonator system will always be nonorthogonal [10].

This implies that the modes are nonorthogonal in general in momentum space. However, if a single radiation channel is involved, they satisfy the orthogonality condition only at a point in the momentum space where $\widehat{\Omega}\widehat{\Gamma}_r = \widehat{\Gamma}_r\widehat{\Omega}$. This defines the ideal FW-BIC point \mathbf{k}_{BIC} when the Hamiltonian ($\widehat{H} = j\widehat{\Omega} - \widehat{\Gamma}_r$, below defined) has a purely imaginary eigenvalue (or equivalently $\widehat{\Omega} + j\widehat{\Gamma}_r$ a purely real eigenvalue). This allows for simultaneous diagonalization of the Hermitian matrices $\widehat{\Omega}, \widehat{\Gamma}_r$.”

Arbitrarily close to the FW-BIC point in momentum space, or at the exact point where a FW-BIC would exist in a real system with losses, the modes become coupled and nonorthogonal as they decay into a single radiation channel. **In real systems, there is always some radiative loss present, which leads to the FW quasi-BIC being coupled in near-field with the final bright partner.** This coupling persists even after diagonalization since the Hamiltonian is non-Hermitian, resulting in eigenvectors that are generally nonorthogonal, unless $\widehat{\Omega}\widehat{\Gamma}_r = \widehat{\Gamma}_r\widehat{\Omega}$. To corroborate our theoretical speculations, we thoroughly validated all the claims made in the paper by explicitly calculating the modes of our system using RCWA (**Supplementary Figs. 4 and 5**).

Supplementary Fig. 4 | a, p-polarized transmittance. **b**, s-polarized transmittance. **c-d**, Detail of avoided crossing and indication of the evolution of the vector TE-like and TM-like modes evolving into each other, with reference to the modes explicitly reported in **Supplementary Fig. 5**.

Supplementary Fig. 5. Major components E_y , E_z of the vector fields of mode 1 (dark, ω_-) and mode 2 (bright, ω_+), in the xy plane at $z = 0$, together with the intensity map I_1 , I_2 in the yz cross section for both modes 1 and 2 at the indicated incidence angle θ , as shown in **Supplementary Fig. 4c**. The colormap of their components on the right indicate the change from TE (E_y) character to TM (E_z) character with varying momentum (θ).

The same non-Hermitian Hamiltonian describes the effect of **coupled-resonance induced transparency**, *i.e.* the analogue of electromagnetically induced transparency (EIT) [*Phys. Rev. A* **69**, 063804 1–6 (2004); *Nano Lett.* **14**, 2783–2788 (2014); *Opt. Express* **31**, 4347 (2023)]. Hsu *et al.* demonstrated that when multiple resonances (two or more) are connected to a single independent decay port, a transparency window, or known as coupled-resonance-induced transparency, always occurs regardless of the radiation loss of the resonances [*Nano Lett.* **14**, 2783–2788 (2014)].

In the vicinity of the BIC point, the **FW quasi-BIC mode** (dark mode) and its bright partner are nonorthogonal eigenvectors coupled to a single radiation channel. This coupling allows for the occurrence of coupled-resonance-induced transparency. This is a key concept that imply that the FW quasi-BIC (A_-) and bright (A_+) modes can be represented as coupled in near field by $\omega_{12,21} \doteq \kappa_{12}$

when $\widehat{\Gamma}_r$ is diagonal because of energy conservation (**Supplementary Sections 1.2-1.3, pp. 6-8, Eqs. 8-32**).

The EIT frequency point falls between the two collective nonorthogonal modes that have gained radiative losses. However, with a small deviation in momentum, this transparency point can become the FW quasi-BIC if the dispersion is tailored based on the system's geometry. In other words, a FW quasi-BIC can be tailored to serve as the dark mode of the EIT process. **We have calculated this using RCWA in Supplementary Fig. 6a to support our TCMT model. In the same Supplementary Fig. 6, we have also calculated the near field coupling and associated quality factor between the two modes sustained by our specific nanostructure, including considerations such as vertical asymmetry, nanocrystal inclusions and measured refractive index dispersion (Supplementary Fig. 6b and 6c).**

Supplementary Fig. 6. a, Spectral coincidence of EIT transparency frequency at the avoided crossing with the FW quasi-BIC frequency for angle mismatch $< 0.6^\circ$. **b**, Near-field coupling constant (real part) normalized to $\omega_+ = 2\pi c/\lambda_{\text{mode1}}$ and **c**, Associated quality-factor Q_κ (as defined above) as a function of z (normalized to $\lambda_{\text{mode1}} = 810$ nm) along the z -axis and parametrized for θ from 2.7° to 3.24° .

This result is then used to evaluate the consequences of considering the FW quasi-BIC as the dark mode in the EIT model (**Supplementary Section 1.3, Eq. 33-40**). The coupling between the dark FW quasi-BIC and the bright leaky partner leads to **the same supercritical local field enhancement equation** that was discussed in the **previous version of this paper**. It is important to note that the dynamical equations used in both the previous and current versions of the manuscript adhere to the strict non-Hermitian Hamiltonian formalism mentioned by the reviewer.

It is important to note that in the presence of nonnegligible absorption loss, the modes are inherently nonorthogonal [see *Nature* **525**, 354–358 (2015) and the associated supporting materials].

I also comment of the comparison with the analogue of the electromagnetically induced transparency in dielectric nanostructured arrays. The authors compare their model to Ref. [25], however, it studies a different system: there two photonic lattices are superimposed to form a new lattice. One of the basis lattices contains a dark BIC mode, the other - bright dipolar mode. Then, the FW-model can be applied to them, because the modes of the resulting structure with two elements in the unit cell are not BICs. The lattice modes interact so the true eigenmodes of the final structure will have finite radiation losses. In this manuscript, the authors start with the description of the true modes of the PCS as the basis modes with an interaction which is different.

As replied above, the same non-Hermitian Hamiltonian can describe **both** coupled-resonance induced transparency and FW-interference. The Hamiltonian changes in momentum space for different input wavevectors with specific in-plane components. Therefore, both conditions can coexist with minimal phase mismatch if the linewidths of the uncoupled resonances are initially unbalanced. This displacement of the FW-BIC away from the avoided crossing point, where EIT may occur, allows for the coexistence of both phenomena.

The formalism employed in this study is general and applicable to different types of optical resonators. Regarding the reference mentioned in the previous version, reference 25 was cited primarily as a well-known paper describing the EIT in dielectric metasurfaces. It was used to provide the reader with a reference for the dynamical equations used in our paper on EIT, now represented by Eqs. 33 and 34. In ref. 25, there was no explicit evidence of a BIC with an ideally diverging quality factor. However, it did mention the presence of a quadrupolar mode, referred to as a dark mode, with low radiative rate, which could potentially be considered a quasi-BIC.

As mentioned above, while the final modes after EIT acquire finite radiative rates, the linewidth of the original (quasi) dark mode affects the width of the transparency region. This characteristic is then translated to the transparency of the FW quasi-BIC when moving away in momentum space towards the FW quasi BIC point.

Comment #1b: Even though the model used by the authors is incorrect, they did not include the far-field coupling of the modes via the continuum to the Eqs. (8-9) of the Supporting Materials. This term is crucial for the achievement of BIC regime.

We appreciate the reviewer's feedback and perspective on the matter. However, we respectfully disagree with the reviewer's comment. In the previous version, Equations 8 and 9 presented a two-wave description, which is a commonly used approach in the literature [see *K. Koshelev, et al., arXiv:2207.01441 (2022)*; *R. Mermet-Lyaudoz, et al., arXiv:1905.03868 (2019)*; *M. Kang, et al, Physical Review Letters 126, 117402 (2021)*]. These equations incorporated a complex coupling term that accounts for both near-field coupling (real part) and far-field coupling mediated by the continuum (imaginary part). They served as dynamical equations representing the standard transposition of the typical non-Hermitian Hamiltonian matrix, leading to the FW-BIC. In this matrix representation, the

coupling term to the continuum was represented by the off-diagonal elements in the radiative decay matrix, resulting in a system of two equations (2 x 2 elements). For clarity, we have reorganized and reformulated the equations in the revised manuscript using matrix notation (see Eqs. 10, 19, 25, 33-34 in Supplementary Materials, Sections 1.2-1.3).

Comment #1c: In addition to the non-radiative losses, the experimental characterization also impacts the observed Q-factors. So, in addition to “Q_a” there should also be a “Q_{excitation}”. The authors describe that the excitation properties affect the Q factor but it is not included in the model. When a small area is excited, highly localized modes will suffer less compared to less localized modes. The observed Q-factor is dependent on the excitation size, and each mode will have a different dependence.

We thank the reviewer for this accurate description of the experimental phenomenology. We added the paragraph “**Excitation Quality factor**” in the **Supplementary Materials, pp. 40-41**, in which we evaluate its influence.

“The pump had a pulse duration $T = 140$ fs. Therefore, an excitation quality factor at 810 nm of $Q_{ex} = 170$ can be readily determined using the hyperbolic secant approximation for the bandwidth [2]. Q_{ex} should approximately three times the quality factor of the resonance to transfer more efficiently the input energy to the system. However, for high Q resonator and non-chirped source this is not a trivial task. Therefore, a certain amount of uncoupled light is expected in this system, but that can be treated in other schemes. In case of the edge emission, since the relevant resonance is the bright mode that had average $Q_{R2} = 213$, the optimal Q_{ex} should be approximately 600. The estimated value of 170 is approximately only one third of the optimum, but not far from the best value. However, it is worth mentioning that the conclusion on the enhancement factor is not affected by this circumstance since it is compensated by comparing the signal between the structured region and the unstructured film.”

We would like to clarify that the data analysis of the enhancement experiment reported in the main paper and supporting materials was not affected by the excitation quality factor. The same excitation was applied both inside and outside the photonic crystal, and since both modes under investigation are extended modes of the photonic crystal slab (with no localized defect modes considered), they were equally affected by the nonideal excitation. Additionally, the 3D vector distribution at the avoided crossing was found to be nearly indistinguishable based on RCWA simulations (**Supplementary Fig. 5**).

Furthermore, we would like to emphasize that the excitation size has been thoroughly studied in our previous research on BICs, and we have found that the influence of the excitation size is typically negligible compared to other fundamental factors, even when using a large numerical aperture. We have demonstrated this in multiple studies, including examples in [*J. Phys. Chem. C* **122**, 19738–19745 (2018); *ACS Nano* **14**, 15417–15427 (2020)].

Regarding the influence of sample size on the enhancement of the local optical field, which is the main focus of this work, a formal analysis of sample size influence was performed in a study by Kodigala et al. [*Lasing action from photonic bound states in continuum*, *Nature* **541**, 196–199 (2017)] on lasing action from photonic BICs. The authors stated that “the persistence of lasing for all array sizes down to as few as 8-by-8 nanoresonators shows the scalability of our BIC laser, thanks to the large quality factor of the resonance-trapped BIC mode in a wide region of k-space.” This indicates that the enhancement and quality factor of the resonance-trapped BIC mode were not significantly affected by the sample size.

In our evaluation of the enhancement factor, we considered the average quality factor obtained by considering the cone of k vectors covered by the convergent input laser, following the same method described in [*Proc. Natl. Acad. Sci. U.S.A.* **110**, 13711-13716 (2013)]. We believe that these considerations support the validity of our analysis and the reliability of the reported enhancement factor.

Comment #1d: While conceptually appealing, the authors use one constant " Q_a " to describe non-radiative losses of both modes. The modal profiles of each of these modes are each different. This means the impact of losses varies from mode to mode, and that a target Q for one mode should be higher than the target Q of another. For example, some modes may be more sensitive to sidewall roughness, while others more sensitive to variations from unit-cell to unit-cell at the sharp corners. Roughness at the top versus sides may also differ between quasi-TE and quasi-TM modes.

We thank the reviewer for this careful comment. The modal profiles calculated with RCWA are reported in **Supplementary Fig. 5**. The absorption and all possible losses are exactly the same at the avoided crossing point, since the modes are nearly identical there, and the influence over the evolution of the modal profile with momentum was found to be negligible (as it was for Q_k), which is also evident from the fact that the linewidth of the resonances exchanges between the two modes in a symmetrical way. The modal profiles also exchange symmetrically. An uneven loss contribution would be evident in experiment and simulation and affect the two modes in an asymmetrical way, but the experimental linewidths of the two modes do change symmetrically.

The formalism we use for constant loss is the same as that used generally in the literature on BICs (and is not unique to this specific topic). We have thoroughly investigated our system to accurately answer the reviewer's comment. First, we have confirmed that the material absorption of silicon nitride cannot affect the loss (**Supplementary Fig. 3**).

Supplementary Fig. 3 | Experimental complex refractive index of the silicon nitride film (UVISEL Plus, Horiba Jobin Yvon, Spectroscopic Ellipsometer), used for numerical simulations (Lumerical 2022 R1, RCWA module).

The complex refractive index dispersion measured on these samples (included in the RCWA simulations) had an imaginary part of below 10^{-6} and allowed us to estimate a total Q factor at the ideal FW-BIC (Q_r infinite) limited to $Q_a = 10^8$. We can conduct simulations where absorption can be disabled in the first simulation and then enabled to discern its contribution relative to other factors.

Therefore, the contribution to Q_a in the experiment (5×10^3) should come from:

1) scattering losses, 2) side leakage, and 3) absorption losses by UCNPs.

The absorption loss (3) is the same for both modes and independent of angle, as considered in the main paper.

The scattering loss (1) was evaluated as $Q_{\text{scatt}} = 10^9$ according to chapter 10.2.5-*Loss Mechanisms* in [Gagliardi, G., and Loock, H.-P. eds. *Cavity-enhanced spectroscopy and sensing. Vol. 179. Berlin: Springer, 2014*], thereby it was found even less significant than the absorption loss of silicon nitride.

The remaining nonradiative loss (2), which could be different for the two modes, may be only the side leakage. The effect of the finite edge was measured by scanning near-field optical microscopy (SNOM) (**Supplementary Fig. 8**) on many different samples. Since it only affects the last unitary cell, there is no significant influence on the modes spanning many hundreds of cells. This ultimately also confirms the experimentally observed symmetric linewidth exchange in agreement with RCWA simulations.

Supplementary Fig. 8 | SNOM maps of the near field excited at frequency close to the BIC, inside the PCNS (top) and close to the boundary of the finite structure (bottom), with corresponding FFT maps. The resonance is perturbed only at the last unitary cell with an intensity reduced by about 50%.

As a final remark, since it is possible to include a differential absorption contribution, we decided to parametrically evaluate the effect of a-factor-1.5 mismatched loss (so up to 50%) with Mathematica. Although the algebraic representation of the equations may have appeared unnecessarily complex, it is important to note that the deviation from the even-loss model curve was found to be negligible. This indicates that despite the complexity of the equations, the small deviation does not significantly impact the main findings of our study.

Comment #1e: At the structure edge, the approximation of a periodic photonic crystal is not applicable, and modes are radiating to the lateral direction. In this sense, the concept of BICs is not clear - the radiation normal to the surface plane can be prohibited by the radiation from the sides of

the structure can be strong. Also, the concept of one radiation channel is not applicable so the FW-model cannot be used for description of the effect.

We thank the reviewer for these thoughtful comments.

Regarding the approximation at the boundary, we first calculated the finite-size dispersion of the PCNS near the edge using the FDTD method. With the due limitations of this method on the individual modes, which include the wave-guided modes of the slab, the calculation showed that any ideal FW-BIC would turn into a finite high-Q mode or FW quasi-BIC with a Q-factor of ca. 3000 (linewidth < 0.25 nm) (**Supplementary Fig. 7a**), considering the homogenous slab and about 200 unitary cells per side of the PCNS.

Supplementary Fig. 7 | a, RCWA simulations for periodic PCNS (top) and FDTD finite PCNS with uniform slab (bottom). The spectral separation between modes 1 and 2 measuring their near-field coupling κ_{12} reduces at the boundary between the PCNS and the uniform slab. **b**, RCWA TE numerical band diagrams of the PCNS as a function of the effective index N_{eff} of the PCNS material, simulating the perturbation induced at the boundary: the effect of the boundary can be approximated with an effective material index variation with RCWA in better agreement with the experiment (Fig. 3a). **c**, Near Field coupling constant (real part) normalized to $\omega_- = 2\pi c/\lambda_{\text{model}}$ calculated for fixed $\theta = 3.24^\circ$ (highest Q_r) and associated quality-factor Q_k (as defined above) as a function of z (normalized to $\lambda_{\text{model}} = 810$ nm) along the z -axis and parametrized with effective refractive index: as the bands come closer with increasing effective index, the coupling constant decreases.

Surprisingly, the symmetry protected BICs are minimally affected by the finite extent (up to $Q_r = 10^5$), but the two-mode interference is more influenced as it depends on the coupling of the two modes. The experimental band diagrams differed from FDTD simulations in several aspects: they did not show the homogeneous slab leaky modes (likely much more lossy than simulations) and also the band distance varied continuously with the movement of the finite pump spot from inside the PCNS to the edge (**Fig. 3a of the main paper**). That is why we used the effective index model with RCWA

(**Supplementary Fig. 7b**), which shows the tuning of the band coupling and is consistent with what was found in the experiments as the edge was approached (**Fig. 3a**).

In our experiment, when **the pump beam is well confined within the PCNS (Fig. 2b and Fig. 3a)**, there is light that can be observed in the far field with no specific orientation, but it is only a fraction of what observed at the edge, and **there is no light that propagates in the plane with high directionality as seen at the edge, so the radiation normal to the surface plane is not prohibited by radiation from the sides**. Strictly speaking, only a finite (high)-Q mode, not a BIC, can survive if the periodicity is broken at the edge. We always speak of quasi-BIC even within the periodic real PCNS, where the high Q character essentially does not change.

The effects of finite size on the quasi-BICs have been addressed in several papers, and it seems that even with a few dozen of unitary cells, the modal distortion of the quasi-BIC and its experimental influence is minimal [Kodigala, A., et al. "Lasing action from photonic bound states in continuum" *Nature* **541**,7636 (2017): 196-199]. Since the pump beam covers an area of hundreds of unitary cells and SNOM reveals a distortion only at the last unitary cell (see **Supplementary Fig. 8 above**), the concept of **quasi-BIC** is sound for finite structures.

As for the number of radiation channels, it must be said that **the side leakage is always present, even if illumination is in the center of the structure, since the waves forming the BIC (or quasi-BIC) propagate in the plane from the excitation point to the finite boundary of the structure and waveguide in the plane (Supplementary Figs. 20 and 21)**.

Supplementary Fig. 20. a, Isosurface far-field intensity map in momentum space showing nontrivial vanishing strips. **b**, Cut-lines of the Z-transform revealing a narrow divergence independent of the

incident/outgoing wavevector along the orthogonal direction (cross of zeros). **c**, Phase map in near field showing a phase vortex singularity. **d**, Corresponding near-field intensity map showing self-collimation. **e**, Experimental proof realized with a rescaled geometry supporting the same band structure at 532 nm: a 20X objective lens focuses the laser (SC-NKT Photonics, filtered at 532 nm) onto the patterned PCNS (red rectangle). The peculiar energy-momentum dispersion induces a resonant field that propagates in plane without spreading along the principal directions $(\pm 1, 0)$ and $(0, \pm 1)$ in agreement with the simulation in **b**.

This means that the only radiation channel to be considered in the FW-model is the far field and not the in-plane wave-guided energy, which eventually flows out of the system as side leakage. **Otherwise, it would show up in the experimental band diagram, affecting the band separation and broadening the linewidth** to the point where any signature of FW-BIC is destroyed (and eventually producing something that cannot be interpreted with simulations). Instead, the signature of BIC is still present in the experimental band diagrams (**Fig. 2a, Fig. 3a, Supplementary Figs 14 and 15**).

We have a large collection of data on experimental structures, even of 50 μm size and band diagrams, showing that the side leakage **perturbs** the ideal BIC (symmetric, random, or FW-BICs) like any other real structure loss, making it a **quasi-BIC but not a simple leaky wave: it perturbs the ideal confinement characteristics of a BIC, but it does not destroy all its properties, not the polarization singularity as an instance, which is a topological property that cannot be destroyed (and which affects all types of BICs). Thus, side leakage must not be included as a main radiation channel, but as a contribution to the non-radiative loss matrix.**

It is important to clarify that the concept of ideal BICs, which have unavoidable losses and side leakage, does not exist or can be observed in practice. Therefore, we commonly refer to quasi-BICs. In the TCMT model, the side leakage should not be considered as a radiation channel. Instead, it is **the mode itself, created by a finite source** naturally propagating within the waveguide plane. Numerous studies have demonstrated the enhancement of light-matter interaction at quasi-BICs, including our own research. In terms of the influence of finite size on (quasi) BICs, there is no difference between our system and other resonators discussed in the literature, whether they are symmetry-protected or FW-type BICs.

However, as the system approaches the edge, a perturbation of the dispersion occurs, causing the bands to come closer (**Fig. 3a**). **In this sense, only a quasi-FW-BIC can exist and we deliberately utilize this tuning to achieve spectral matching of the two frequencies in our work.**

Comment #2: The narrowband spatial divergence of the UCPL is not explained in the paper. The authors mention that it can be related to the narrowband behavior of BICs in the k -space, but it is confusing because the emission is not at the BIC wavelength thus it is not directly related to the spatial distribution of the BIC. Moreover, the BICs are narrowband in the quasi-normal direction of the k -space (5 degrees) while none of that can be said about the lateral direction.

We thank the reviewer for this very important consideration, which has allowed us to strengthen the relevance of our work.

We agree that the origin of the narrow UCPL emission was only touched upon in the previous version. Recognizing this as an important part of the work, we have now provided, as requested by all reviewers, a comprehensive description of this phenomenon that strengthens the relevance of our experimental findings with an appropriate theoretical framework. Accordingly, we have changed **Figure 4a-c** in the main manuscript, and added **Supplementary Figs. 19-22**.

In the previous version, we claimed that the narrow emission was associated with the reciprocal space filtering effect of the BIC. We have now included in-depth evidence for this claim using FDTD simulations. First, the FDTD method for describing the effect has been validated and reproduces the literature results presented in the paper on the supercollimation effect inside photonic crystal slabs [Rakich, P. T. et al. Achieving centimetre-scale supercollimation in a large-area two-dimensional photonic crystal. *Nat. Mater.* **5**, 93–96 (2006)], in **Supplementary Fig. 19**. This is necessary to better highlight the different principle in our work. We have detailed in the revised manuscript (**Figure 4a-c** and **Supplementary Fig. 20**) equifrequency maps calculated using the far-field Z-transform of the mode at the BIC frequency. They show that the FW quasi-BIC mode produces suppressed radiation along strips crossing the center of reciprocal space, resulting in in-plane and coherent propagation along collimated directions parallel to the principal axes of the photonic lattice.

To investigate the emission characteristics at the edge of the system, we conducted calculations considering over 300 dipole emitters that covered an area corresponding to the input beam spot. The results of these calculations are presented in **Supplementary Figure 21**. Interestingly, our findings revealed that the emission from the edge, directed outward in the plane of the homogeneous slab, can exhibit an extremely narrow divergence angle, reaching as low as 0.02° . This behavior is consistent with our experimental findings and can be attributed to phased-array radiation. Consequently, our study demonstrates that the phenomenon of supercollimation arises from the specific energy-momentum dispersion properties associated with the BIC condition, combined with the coherent dipole-mediated phase emission imposed by the in-plane photonic lattice.

Supplementary Fig. 21 | FDTD simulation with 324 dipole sources aligned with the major component of the FW quasi-BIC field. The dipoles cover a circular area of several microns squared (same size of the input spot focused with an objective lens) at the boundary of the PCNS with the uniform slab. Their emission is pumped by the maxima of local optical field (shown in the experimental near-field map in **Supplementary Fig. 8**), but their fields also collectively add up coherently emitting radiation in the plane of the slab as triggered by the patterned geometry affecting their spatially correlated emission. **a**, The field propagates along the direction $(+1, 0)$ when the right boundary is excited with intensity enhancement as large as 1.5×10^4

(normalized to the number of emitters) at the quasi FW-BIC wavelength (810 nm). **b**, At shorter wavelengths (540 nm), as those produced with the experimental UCNPs emitters, the spatially correlated fields produce narrow emission (calculated along 1 mm of propagation from the edge), and with beam divergence even of 0.02° as shown in **c**. **d**, The analysis over the whole visible and near-IR spectrum revealed that the typical value of divergence was below 0.5° . **e**, The FWHM of the beam shown in **b**, changes along the propagation obeying a mechanism of self-healing which compensates the diffraction.

Comment #3: The authors use a phase matching liquid (silicone oil) on top of the sample to achieve the up-down inversion symmetry of the structures which is a requirement for FW-BICs. This geometry seems highly impractical, while without this liquid the FW-BIC cannot form.

We believe that the reviewer may have misunderstood the role of silicone oil in our experimental setup. The silicone oil used in our experiments was primarily to enhance the visualization of the in-plane propagating coherent upconverted luminescence (UCPL). It acted as a partially opaque screen that allowed for better observation of the side emission. Note that the FW-BIC is favored by the vertical asymmetry produced by the different material index along the z-axis in our system, see [*Phys Rev B* **61**, 2090–2101 (2000) and *R. Mermet-Lyauoz et al. Realization of Bound state In the Continuum induced by vertical symmetry breaking in photonic lattice*, *arXiv:1905.03868v1 [physics.optics]* 2019]. We have better clarified in the revised paper (**Supplementary Materials pp. 15-16, Supplementary Fig. 2**).

“The silicone oil promotes the vertical symmetry, which means that it increases the field overlap with the UCNPs and helps minimize scattering losses, but it cannot affect the vertical symmetry of the TE-like and TM-like modes, which is determined mainly by the different refractive index of the glass substrate, silicon nitride and UCNPs index. Indeed, the energy fraction with silicon oil only changes from 8% to 9% (**Supplementary Fig. 2d**). Nonetheless, silicon oil was often useful to better observe the side emission, as the silicone layer acted as a partially opaque screen crossing the outcoupled light (as shown in Fig. 3b of the main paper). It should be noted that the silicon oil layer was not used in Fig. 4b”

Comment #4: More detailed comparison with earlier reports on the enhancement of UCPL with dielectric and plasmonic nanostructures is required to outline the scope of the current results within the existing literature.

A recent review was cited as ref. 47 [Das, A., Bae, K. & Park, W. Enhancement of upconversion luminescence using photonic nanostructures, *Nanophotonics* **9**, 1359–1371 (2020)] in the **Discussion** section, and this reference seems the most recent and complete when it comes to upconversion enhancement in dielectric nanostructures. Since our main goal was not to compare our system with other resonant systems in terms of upconversion enhancement, our intention was to present a brief discussion on this topic rather than a systematic survey of the state of the art. Nevertheless, we did take note of the values summarized in ref. 46 and found that the enhancement value we reported appears to be the highest among the cases mentioned.

Comment #5: The experimental results lack any statistical analysis. Statistical tests with corresponding error bars and probability values are required to confirm robustness, validity, reliability of the results.

We apologize for the oversight in the previous version of our paper regarding the full details and statistics of our experimental samples. In our study, we used a total of 17 samples to demonstrate the fundamental results. These samples consisted of with two completely different kinds of UCNPs with variations in chemical composition, shape/size and emission lines. The inclusion of different

nanocrystals allowed us to showcase the versatility of our resonator design and its applicability across various systems. In the revised version of the paper, we have further expanded our evaluation by incorporating another 24 samples. These additional samples include variations in size to address the concerns raised by **reviewer #2**. The statistical analysis and results from these samples have been included in **Supplementary Figs 18 and 24**.

Supplementary Fig. 18 | **a**, UCPL emission at 540 nm as a function of the PCNS number (10 points for each sample). **b**, Histogram of the intensity distribution (UCPL peak @540 nm) from core-shell NPs used for EF calculation. **c**, Intensity distribution of the emission achieved scanning the input beam along the full boundary of the PCNS for all major upconversion wavelengths.

Minor comments. Comment #6: The authors employ a FW-BIC to realize the effect, however, no explanation is given why symmetry protected BICs are not applicable. For symmetry protected BICs the phase matching liquid could be not necessary.

In response to this suggestion, we have included a separate section in the **Supplementary Materials (Sec. 1.6)** that provides a more detailed explanation of symmetry-protected BICs and their characteristics, including compliance with critical coupling conditions.

As mentioned earlier, the use of a matching fluid is not necessary in our system. Both symmetry-protected BICs and random BICs can be realized in vertically asymmetric structures simply by employing different refractive indices for the supporting material and the slab with hole patterns in air. This vertical asymmetry, as demonstrated by Paddon and Young in 2000 (reference 23), eliminates the need for additional measures such as partially etched holes in a silicon nitride waveguide to break the vertical symmetry.

Comment #7: It is unclear how the angle of incidence required for excitation of the FW-BIC is related to the emission directionality and properties.

References.

- [R1] Sauvan, C., Wu, T., Zarouf, R., Muljarov, E.A. and Lalanne, P., 2022. Normalization, orthogonality, and completeness of quasinormal modes of open systems: the case of electromagnetism. *Optics Express*, 30(5), pp.6846-6885.
- [R2] Fan, S. and Joannopoulos, J.D., 2002. Analysis of guided resonances in photonic crystal slabs. *Physical Review B*, 65(23), p.235112.
- [R3] Tikhodeev, S.G., Yablonskii, A.L., Muljarov, E.A., Gippius, N.A. and Ishihara, T., 2002. Quasiguidded modes and optical properties of photonic crystal slabs. *Physical Review B*, 66(4), p.045102.
- [R4] Andreani, L.C. and Gerace, D., 2006. Photonic-crystal slabs with a triangular lattice of triangular holes investigated using a guided-mode expansion method. *Physical Review B*, 73(23), p.235114.
- [R5] Hsu, C.W., Zhen, B., Stone, A.D., Joannopoulos, J.D. and Soljačić, M., 2016. Bound states in the continuum. *Nature Reviews Materials*, 1(9), pp.1-13.
- [R6] Neale, S. and Muljarov, E.A., 2021. Accidental and symmetry-protected bound states in the continuum in a photonic-crystal slab: a resonant-state expansion study. *Physical Review B*, 103(15), p.155112.

We thank the reviewer for this important comment, and for the references brought to our attention, several of which very familiar to our work. R5-R6 were included in the revised manuscript (Refs.4,9). In the revised version of our paper, we have conducted a thorough investigation into this phenomenon. This is described in detail in the section titled "**Self-collimation Effect and Radiance Enhancement**" and is supported by **Figure 4** and **Supplementary Figures 19-22**. We demonstrate that this effect can be attributed to the unique energy-momentum dispersion characteristics associated with the BIC condition (**Figure 4a and b**, also added below for convenience):

“Diffraction-free guiding with BICs has been observed in the microwave range⁵⁰ and associated with the phenomenon of self-collimation inside the structured waveguide^{51,52}. In our experiments, however, not only the propagation starts from the PCNS edge and continues in the slab, but the negligible divergence is furthermore observed in the upconversion luminescence at wavelengths very far from the FW quasi-BIC at 810 nm. Therefore, the mechanism of self-collimation at the BIC must be considered only one ingredient of a more complex scenario. To investigate the origin of the narrow upconversion emission, finite-difference-time-domain (FDTD) simulations were performed (Supplementary Sec. 8). In the simulations, a single dipole source was used to compute the iso-frequency map using the Z-transform of the local optical field. Initially, the simulations validated the conventional self-collimation due to flat-band dispersion in the momentum space⁵² (**Supplementary Fig. 19**). In our system, the iso-frequency map shows nontrivial vanishing strips along the symmetry axes of the geometry, intersecting squared flat bands (**Fig. 4a**). The associated real-space intensity map also shows self-collimation characteristics. However, in this case, it is the low coupling in the far field along the strips that leads to negligible divergence (**Fig. 4b**). This was also confirmed experimentally in a PCNS geometry scaled to produce the FW-BIC at 532 nm for making the in-plane beam visible. The field emitted at the BIC frequency confirmed self-collimation over centimeter distance (**Supplementary Fig. 20**).”

Regarding the upconversion frequency, we performed simulations using the FDTD to investigate the emission characteristics of nanocrystals. Our simulations demonstrated that the emitted light follows a mechanism of concurrent coherent phased-array emission, which is dictated by the properties of the photonic lattice (as shown in **Figure 4c** and **Supplementary Figure 21**). Furthermore, we have conducted extensive experimental investigations to provide evidence supporting our theoretical findings. In line 245 of the revised manuscript, we state the following:

“To examine the edge radiation properties, an array of dipole sources covering an area consistent with the experimental pump spot was placed at the boundary between the finite PCNS and the homogenous waveguide slab. FDTD simulations indicate that the dipole point-sources collectively radiate in the plane of the homogenous waveguide with coherent phased-array emission governed by the photonic structure, thereby resulting in minimal divergence, as low as 0.02° , across a large part of the visible spectrum (**Supplementary Fig. 21**). Remarkably, this was experimentally verified and compared with simulations in Fig. 4c, where the emission at representative wavelengths in the visible range was

probed at single input wavelengths. Microscopy analysis of the beam revealed a high degree of spatial coherence, as evidenced by the visibility of the interference pattern (Supplementary Fig. 22).”

Fig. 4 | Directivity and polarization-based switching. **a**, Iso-frequency far-field intensity map in the momentum space, revealing the presence of nontrivial vanishing strips. **b**, Near-field intensity map showing self-collimation along the symmetry axes. **c**, Simulated (top) and experimental (bottom) collimated emission in the visible range at 544, 580, and 650 nm over centimeter scale from a sample of 100 μm of side length. On a glass slide, there is an array of 24 photonic crystal slabs with variable side lengths of 1 mm, 300 μm and 100 μm . The input light is focused to a spot of 10 μm inside a patterned area of 100 μm . Some cross-talk beams are generated by the propagation of the main beams from the illuminated area when they cross surrounding patterned areas. **d**, Snapshots of controlled light propagation near the edge (left inset), demonstrating that both horizontal and vertical side beams can be generated at the corner of the photonic crystal slab: the intensity of the output is determined by the orientation of the input polarization, which is varied with the orientation of the half-wave plate axis (right inset). Scale bar: 1 mm (d).

In closing, we wish to thank the reviewer once again for thoughtful and constructive comments. The reviewer's thorough review has provided valuable guidance as we work to improve.

Referee #2 (Remarks to the Author):

The manuscript demonstrates the experimental demonstration of a coherent up-conversion luminescence enhanced by 8 orders of magnitude. The emission is characterized by negligible divergence, microscale width, and direction-steering capability controlled by input focusing and polarization.

BIC-based devices are drawing great interest in recent years and generally appeal to a broad readership. However, I don't believe the work has the impact and significance necessary to be published in Nature. Furthermore, one of the authors published similar work in 2019 in Nature Communications (Up conversion amplification through dielectric superlensing modulation <https://doi.org/10.1038/s41467-019-09345-0>) with an efficiency up-conversion of five orders of magnitude. A demonstration from 5 orders to 8 orders is not significant enough to be considered for Nature Journal. After addressing my comments, I recommend the authors submit the paper to Journal like Nano letters.

We would like to express our gratitude to the reviewer for the meticulous review and valuable suggestions. We acknowledge and recognize the importance of Nature journal in publishing primary research that is of interest to a broad scientific community. In light of this, we would like to take this opportunity to clarify the significance of our work.

In the paper by Liang et al. [*Nat Commun* **10**, 1391 (2019) <https://doi.org/10.1038/s41467-019-09345-0>], dielectric microbeads were used to enhance photon upconversion in lanthanide-doped nanocrystals through dielectric superlensing effects. This approach resulted in a substantial increase in luminescence, achieving up to 5 orders of magnitude enhancement. The key principle involved the modulation of wavefronts to converge low-power incident light beam into photonic hotspots, thereby increasing light coupling and local energy density at the nanocrystal layer. This technique demonstrated its efficacy in enabling efficient upconversion imaging microscopy with a large field of view.

It is important to note that numerous nanostructures have been developed over the past decades to create platforms that enhance hotspots or resonators, aiming to improve light-matter interactions in various physical phenomena.

In contrast, our work is situated in a distinct context, as it introduces a fundamentally novel concept known as "supercritical coupling" and experimentally verifies its ability to achieve record-breaking enhancement of photon upconversion. This concept, although universal to the physics of light-matter interaction, offers unprecedented control over the emission direction. Our research addresses two fundamental challenges.

The first challenge relates to the fundamental limit to far-field pumping of bound states in the continuum (BICs). BICs have opened up possibilities for topological nanophotonics with ultrahigh radiative quality factors (Q_r up to 10^6), which have implications for quantum physics and information

technology. However, as with other optical resonances, the near-field enhancement is limited by critical coupling, and in a dark mode, the enhancement approaches zero.

Previous solutions have relied on finite-by-construction Q_r values, typically on the order of 10^2 , **to satisfy the critical coupling condition** $Q_r = Q_a$ in metasurfaces with large nonradiative losses $Q_a = 10^2$ (see *Phys. Rev. Lett.* 2018, 121, 193903; *Science* 2020, 367, 288). This limitation significantly undermines the potential of high-Q resonators based on BICs. To address this problem, *virtual critical coupling* was recently theorized for single resonators [*ACS Photonics* 7, 1468–1475 (2020)]. However, this approach relies on exponential-type time modulation of pump intensity, and experimental validation of this principle is still pending.

In our work, we propose and validate a completely new concept: a supercritical coupling condition that can enhance the local optical field in any high-Q resonator. The local optical field enhancements can reach several orders of magnitude higher than those achieved through conventional critical coupling. This was demonstrated by leveraging the proximity of the FW-BIC in energy momentum space to the transparency window produced through the interference of nonorthogonal coupled resonances. This proximity enables the formation of a single radiation channel, leading to significant enhancements in the local optical field. We have further emphasized the connection between the formalism of coupled-resonance-induced transparency, which served as the classical analogue of electromagnetically induced transparency (EIT) and Friedrich-Wintgen interference. Through this analysis, we thoroughly demonstrate the supercritical enhancement equation derived from the coupled-mode theory (TCMT) and validate all our findings through comprehensive numerical simulations, as suggested by the reviewer (**see response to specific *Comment #1***).

The second aspect of this work concerns the enhancement of photon upconversion. Achieving efficient photon upconversion in lanthanide-doped nanoparticles at low irradiance is not only a fundamental challenge but also crucial for numerous applications, such as super-resolution imaging to nanophotonics (see *Nature* 2017, 543, 229; *Nature* 2020, 587, 594; *Nature* 2021, 589, 230). Existing implementations based on surface passivation, surface plasmon coupling, photonic crystal engineering, and broadband sensitization have only achieved limited improvement in upconversion emission (~ 4 to 5 orders of magnitude enhancement). In our study, we experimentally demonstrate that the supercritical edge BIC enables an 8-order-of-magnitude increase in upconversion brightness, surpassing the limit associated with critical coupling by two orders of magnitude. In addition, we demonstrate the unique capability of controlling directive emission (**see response to specific *Comment #5***), which has broad implications for a vast scenario of microscale photonic chip applications, including light sources, information delivery, photodetection, and sensing. In addition to the significant increase in enhancement from 5 to 8 orders of magnitude compared to the work by Liang et al. [*Nat Commun* 10, 1391 (2019)], our approach achieves this enhancement through the precise alignment of two fundamental phenomena, namely EIT and FW quasi-BIC, in the energy-momentum space. This tailored coincidence of EIT and quasi-FW-BIC represents a unique physical principle that has not been explored or documented in existing scientific literature.

Comment #1: Is the term “supercritical coupling” an existing term in literature, or introduced by the authors? References should be cited to introduce the concept and/or the context. It appears that supercritical coupling here actually refers to the regime where κ_{12} is sufficiently small, and it follows that “under this condition the low-Q mode can funnel energy into the high-Q BIC mode via near-field interaction”: why does energy funneling happen when mutual coupling is weak? The text is not clear and lacks backup and/or reference.

Our work is the demonstration that a Friederich-Wintgen BIC can be used to achieve upconversion coherent emission beyond the state of the art, breaking what is conventionally though possible about coupling with such modes by theorizing and demonstrating for the first time the supercritical coupling. There is no parallel in literature for coupled-mode enhancement between bright and dark mode, being the dark one a quasi-FW-BIC; there is no parallel for such values of upconversion enhancement and nor for coherent directional emission (specific Comment #?).

The maximum of the local field enhancement predicted by **Eqs. 38-40** of the revised **Supplementary Materials** is achieved when $Q_\kappa = Q_a$ (or $\tau_\kappa = \tau_a$), which requires that κ_{12} is sufficiently small in order for Q_κ to be large enough to match with Q_a , concurrently having negligible frequency detuning ($\omega_- = \omega_+ - \delta$) so as that the approximation leading to **Eq. 38** in **Supplementary Materials** is valid (or **Eq. (2)** in the main paper).

We revised the text accordingly to clarify this issue from line 169 (main paper):

“The upconversion emission experiences an increase by several orders of magnitude when the input beam crosses the edge of the PCNS. **Figure 3a** shows a continuous transformation causing inner bands to merge into boundary bands with closed gap as the input source crosses the edge, which is correlated with continuously increasing lateral emission. The mode dispersion modification was measured by performing a space-variant band-diagram characterization at the edge of the PCNS using a filtering scheme with microscale resolution (6- μm spot, same as for upconversion, Supplementary Sec. 5). At the edge, an intense beam is generated in the transverse plane at the interface between patterned and unpatterned regions (**Fig. 3a, b** and Supplementary **Media 1**), becoming more visible as it crosses the silicone layer. This occurs when κ_{12} decreases to the point where the bands overlap at -2.9° , exactly where the FW quasi-BIC is located (-3.4° , dashed line) with a momentum mismatch of 17% ($\Delta\theta \approx 0.5^\circ$). This confirms the coupling of dark and bright modes, consistent with numerical calculations (**Supplementary Fig. 7**). According to the supercritical coupling model, the frequency of the FW quasi-BIC is $\omega_- = \omega_+ - \delta$, where δ is proportional to κ_{12} . At the edge point κ_{12} becomes sufficiently small for the resonances to overlap (low detuning), resulting in $\omega_+ \approx \omega_- = \omega_{\text{in}}$. Concurrently, Q_κ increases, thereby allowing the enhancement factor relation to be well approximated by Eq. (2).”

Comment #2: It will be very interesting to present the state-of-the-art coherent up-conversion luminescence and compare them to this work.

A recent review was cited as ref. 47 [Das, A., Bae, K. & Park, W. Enhancement of upconversion luminescence using photonic nanostructures, *Nanophotonics* **9**, 1359–1371 (2020)] in the **Discussion** section, and this reference seems the most recent and complete when it comes to upconversion enhancement in dielectric nanostructures. Since our main goal was not to compare our system with other resonant systems in terms of upconversion enhancement, our intention was to present a brief discussion on this topic rather than a systematic survey of the state of the art. Nevertheless, we did take note of the values summarized in ref. 46 and found that the enhancement value we reported appears to be the highest among the cases mentioned. Moreover, the narrow divergence in our system is a unique property, which has never been discussed before (see also **Comment #5**).

Coherent upconversion discussed in the literature is associated with lasing, which imparts spatial and temporal coherence to the emission depending on the choice of cavity mode momentum, due to the large density of the states at the specific emission wavelength produced by the nanostructure. Our work does not deal with lasing, so there is no control on time coherence because the local field is enhanced at the pump wavelength and not at the emission wavelengths (a second BIC at emission would also lead to lasing). The spatial coherence is imposed by two factors:

- i) The peculiar nature of the BIC phase across the field distribution (**Supplementary Fig. 20c**), which is associated with the reciprocal space filtering arisen from the prohibited radiation along strips in momentum space (shown in revised **Figure 4a**): emitters driven by a coherent pump field resonantly enhanced by field phase coherence inherit this spatial coherence [*Popmintchev, T., et al. "Bright coherent ultrahigh harmonics in the keV x-ray regime from mid-infrared femtosecond lasers" Science 336, 1287-1291(2012)*].
- ii) In addition, our simulations with dipole emitters at the upconversion frequencies have revealed coherent phased-array emission: the emitters radiate in phase with each other obeying the field and phase distribution sustained by the photonic crystal slab in which they are incorporated. Summation over their phased-array field produces narrow divergence radiation, as in Bragg scattering. This was observed in the upconversion emission and also independently verified by exciting the nanostructure in the visible range with single wavelengths from the supercontinuum laser (**Figure 4c**). To the best of our knowledge, this is the first time such coherent phased array emission at upconversion wavelengths has been reported, due to the phase coherence of the PCNS mode at emission frequencies (see also specific *Comment #5*).

Comment #3: I am curious to see how the authors could compare this work to a Valley topological laser where a selective excitation of the corners can be implemented via directional evanescent coupling with a spin-polarized source without high Q:

1. Room-temperature lasing from nanophotonic topological cavities. *Light Sci Appl* 9, 127 (2020). <https://doi.org/10.1038/s41377-020-00350-3>
2. Topological Insulator Laser Using Valley-Hall Photonic Crystals, <https://doi.org/10.1021/acsp Photonics.0c00521>

We thank the reviewer's suggestion to consider the mentioned papers. However, there are several key differences between our work and the paper referenced.

First, in our study, we did not observe lasing, which is a distinct characteristic of the topological corner states described in those papers. Our focus was on achieving supercritical coupling and enhancing photon upconversion, rather than demonstrating lasing phenomena.

Secondly, the topological nature of the modes in the referenced papers arises from specific nanostructures, such as triangular lattices with tailored geometries, leading to band inversion and the formation of nontrivial Chern numbers. These structures confine light in the transverse plane, with additional out-of-plane confinement achieved through total internal reflection. In contrast, our system relies on the confinement of the field in the vertical direction due to the topological properties of FW quasi-BIC in the reciprocal space. The mode in our system is extended in the direct space, and the directive emission observed is a result of super-collimation and phased-array emission facilitated by the photonic lattice, rather than lasing.

Lastly, while the papers mentioned by the reviewer highlight the importance of achieving high-Q modes for enhancing light-matter interactions, our work addresses this requirement as well. In dielectric structures like ours, high-Q is crucial for increasing the interaction time and enhancing any light-matter process, as emphasized in the reference provided [*Khurgin, J.B. Nat. Photon. (2023)*]. Our system achieves a high intrinsic quality factor (Q) of 5.2×10^3 , comparable to the range reported in one of the referenced papers for the corner states. However, it should be noted that our work differs significantly in terms of the overall system design and the physical mechanisms driving the observed enhancements in photon upconversion.

In summary, while the referenced papers offer valuable insights into topological cavities and lasing phenomena, our work explores a distinct research direction focusing on supercritical coupling and the enhancement of photon upconversion through tailored coincidence of electromagnetically induced transparency and FW quasi-BIC. In our system, a notable aspect is that the emission spectrum of the upconversion is not modified by the final density of states. Unlike systems where emission is peaked at a specific wavelength, our approach allows a broader output spectrum. This is advantageous when using a combination of nanoparticles with emissions covering a continuum spectrum in the visible range. In essence, our system enables the production of a white spectrum of emitted light, while still maintaining the collimation and spatial coherence characteristics of a laser beam in the plane. The combination of broadband emitted light and collimation enhances the versatility and potential applications of our system.

We have discussed this comparison in the discussion section of the main paper, citing the mentioned references as Refs. 53 and 54, starting from line 293:

“The resulting ultrabright upconversion luminescence is enhanced by over 8 orders of magnitude, representing one of the highest values achieved with a dielectric resonator⁴⁷. This opens up possibilities for future developments in upconversion lasing. The topological field confinement associated with the BIC, which has already been used for lasing action, expands the paradigm of disorder-immune compact photonic devices based on light topological phases^{53,54}. By incorporating concurrent upconversion enhancement at pump and emission frequencies, this system offers promising capability for on-chip microscale light control, sensing, and quantum information. In the present state though, a point of strength of our system is that the emission spectrum of the upconversion is not peaked at a specific wavelength since the field enhancement is at the pump wavelength. In other words, the output spectrum can be produced by the combination of several type of nanoparticles with spatial coherent emission covering a continuum range in the visible. This unprecedented control over emission provides significant new possibilities for all-dielectric on-chip nanophotonics.”

Comment #4: Regarding a key finding of this work, i.e., reduced coupling κ_{12} at the edge, and the resultant high-Q mode, I am not convinced by the corresponding numerical analysis and supporting text (Fig. S3 and Supplementary Section 1.5). Firstly, what happens at the edge should be non-uniform, but the authors seemingly used a uniform periodic simulation to emulate the edge effects, only changing the effective material index. This simulation is oversimplified to me. Secondly, the quantity of interest here is the coupling coefficient κ_{12} (we need to see it decrease at the edge). The authors gave the equation for it but never seemed to have rigorously evaluated it, which requires finding the fields rigorously in the first place (I agree one can qualitatively see the reduced coupling in Fig. S3 but again this is emulated). Lastly, explanation is only qualitative: “Indeed, the FW-BIC is a quadrupole mode” (OK, I have to scroll to Fig. S6c to see that), “but it gains a perturbation-like dipolar character at the boundary because of the broken symmetry of the system at the edge”, (where is the data?). And it follows “At the same time, the bright wave (dipolar mode) gains a quadrupole character at the finite boundary”. (How do you back this up?)

We are grateful to the reviewer for this comment as it prompted us to further explore and refine our theoretical framework and experimental investigations. In response, we have made significant revisions to the manuscript and Supplementary Materials to address all the criticisms raised. Specifically, we have added 12 Supplementary Figures that provide detailed simulations and experimental data to support our theory. To address the reviewer's concern about mode profiles, we have included explicit 3D vector simulations of the mode profiles in **Supplementary Fig. 5**. Additionally, we have calculated the complex coupling constant as a function of the modes in **Supplementary Fig. 6** and approximated the boundary conditions in **Supplementary Fig. 7**.

These supplementary figures, along with the revised text, provide a detailed account of the theoretical model and numerical simulations. We have carefully evaluated all the quantities discussed in the TCMT model and conducted numerical calculations to validate our approach. Regarding the character of the modes, we have checked the references cited in the previous version and incorporated full calculations and details as suggested by the reviewer.

Comment #5: The interpretation of the high directionality of the emission is very sloppy. As the paper stated in itself, compared to the actual paradigm of supercollimation (Ref. 48, let us leave aside whether this analogy is valid), here “the high directionality and collimation was observed in visible upconverted radiation, not only in the pumping mode” (Line 237). Now how can the directionality of the emission wave be attributed to k-vector selectivity of the pumping mode? The emitting photonic mode at the up-converted wavelength should be investigated and elaborated as well, at least numerically.

In our previous version, we did not fully discuss the narrow divergence of outgoing UCPL radiation.

We agree with the reviewer that a systematic interpretation of this phenomenon was necessary. To address this, we have added a section titled "**Self-collimation Effect and Radiance Enhancement**" starting from line 224. This section provides a thorough theoretical and experimental analysis of the self-collimation effect and the resulting enhancement in radiance. We have included a new **Figure 4** and **Supplementary Fig. 20 and 21** to present relevant data and support our findings.

Fig. 4 | Directivity and polarization-based switching. **a**, Iso-frequency far-field intensity map in the momentum space, revealing the presence of nontrivial vanishing strips. **b**, Near-field intensity map showing self-collimation along the symmetry axes. **c**, Simulated (top) and experimental (bottom) collimated emission

in the visible range at 544, 580, and 650 nm over centimeter scale from a sample of 100 μm of side length. On a glass slide, there is an array of 24 photonic crystal slabs with variable side lengths of 1 mm, 300 μm and 100 μm . The input light is focused to a spot of 10 μm inside a patterned area of 100 μm . Some cross-talk beams are generated by the propagation of the main beams from the illuminated area when they cross surrounding patterned areas. **d**, Snapshots of controlled light propagation near the edge (left inset), demonstrating that both horizontal and vertical side beams can be generated at the corner of the photonic crystal slab: the intensity of the output is determined by the orientation of the input polarization, which is varied with the orientation of the half-wave plate axis (right inset). Scale bar: 1 mm (**d**).

Supplementary Fig. 20 | **a**, Isofrequency far-field intensity map in momentum space showing nontrivial vanishing strips. **b**, Cut-lines of the Z-transform revealing a narrow divergence independent of the incident/outgoing wavevector along the orthogonal direction (cross of zeros). **c**, Phase map in near field showing a phase vortex singularity. **d**, Corresponding near-field intensity map showing self-collimation. **e**, Experimental proof realized with a rescaled geometry supporting the same band structure at 532 nm: a 20X objective lens focuses the laser (SC-NKT Photonics, filtered at 532 nm) onto the patterned PCNS (red rectangle). The peculiar energy-momentum dispersion induces a resonant field that propagates in plane without spreading along the principal directions $(\pm 1, 0)$ and $(0, \pm 1)$ in agreement with the simulation in **b**.

Supplementary Fig. 21 | FDTD simulation with 324 dipole sources aligned with the major component of the FWquasi BIC field. The dipoles cover a circular area of several microns squared (same size of the input spot focused with an objective lens) at the boundary of the PCNS with the uniform slab. Their emission is pumped by the maxima of local optical field (shown in the experimental near-field map in **Supplementary Fig. 8**), but their fields also collectively add up coherently emitting radiation in the plane of the slab as triggered by the patterned geometry affecting their spatially correlated emission. **a**, The field propagates along the direction $(+1, 0)$ when the right boundary is excited with intensity enhancement as large as 1.5×10^4 (normalized to the number of emitters) at the quasi FW-BIC wavelength (810 nm). **b**, At shorter wavelengths (540 nm), as those produced with the experimental UCNP emitters, the spatially correlated fields produce narrow emission (calculated along 1 mm of propagation from the edge), and with beam divergence even of 0.02° as shown in **c**. **d**, The analysis over the whole visible and near-IR spectrum revealed that the typical value of divergence was below 0.5° . **e**, The FWHM of the beam shown in **b**, changes along the propagation obeying a mechanism of self-healing which compensates the diffraction.

From line 228 in the main manuscript, we explain that the origin of self-collimation can be attributed to the specific nature of the energy-momentum dispersion associated with the BIC condition. Additionally, we discuss how the photonic lattice, in conjunction with dipole sources, imposes concurrent coherent phased-array emission at the upconversion frequencies. This combination of factors leads to the observed self-collimation effect.

“Diffraction-free guiding with BICs has been observed in the microwave range⁵⁰ and associated with the phenomenon of self-collimation inside the structured waveguide^{51,52}. In our experiments, however, not only the propagation starts from the PCNS edge and continues in the slab, but the negligible divergence is furthermore observed in the upconversion luminescence at wavelengths very far from the FW quasi-BIC at 810 nm. Therefore, the mechanism of self-collimation at the BIC must

be considered only one ingredient of a more complex scenario. To investigate the origin of the narrow upconversion emission, finite-difference-time-domain (FDTD) simulations were performed (Supplementary Sec. 8). In the simulations, a single dipole source was used to compute the iso-frequency map using the Z-transform of the local optical field. Initially, the simulations validated the conventional self-collimation due to flat-band dispersion in the momentum space⁵² (**Supplementary Fig. 19**). In our system, the iso-frequency map shows nontrivial vanishing strips along the symmetry axes of the geometry, intersecting squared flat bands (**Fig. 4a**). The associated real-space intensity map also shows self-collimation characteristics. However, in this case, it is the low coupling in the far field along the strips that leads to negligible divergence (**Fig. 4b**). This was also confirmed experimentally in a PCNS geometry scaled to produce the FW-BIC at 532 nm for making the in-plane beam visible. The field emitted at the BIC frequency confirmed self-collimation over centimeter distance (**Supplementary Fig. 20**).”

To investigate the upconversion frequency of the light radiated by the nanocrystals, we simulated light emission using FDTD simulations. The upconverted light follows a mechanism of concurrent coherent phased-array emission, which is imposed by the presence of the photonic lattice (**Figure 4c**, **Supplementary Fig. 21**). To support our theoretical findings, we have conducted additional experimental measurements. The new experimental data further validate the predicted mechanism of concurrent coherent phased-array emission. At line 246 of the revised manuscript, we state the following:

“To examine the edge radiation properties, an array of dipole sources covering an area consistent with the experimental pump spot was placed at the boundary between the finite PCNS and the homogenous waveguide slab. FDTD simulations indicate that the dipole point-sources collectively radiate in the plane of the homogenous waveguide with coherent phased-array emission governed by the photonic structure, thereby resulting in minimal divergence, as low as 0.02° , across a large part of the visible spectrum (**Supplementary Fig. 21**). Remarkably, this was experimentally verified and compared with simulations in **Fig. 4c**, where the emission at representative wavelengths in the visible range was probed at single input wavelengths. Microscopy analysis of the beam revealed a high degree of spatial coherence, as evidenced by the visibility of the interference pattern (**Supplementary Fig. 22**).”

Comment #6: The up conversion here is a multi-photon process (Line 109). I am confused by how it exhibits linear behavior ($s \sim 1$ in Fig. 2d and 3c)? The authors attempted to explain this by stating that with the strong excitation here single-photon absorption readily occurs (Line 143), but how does single-photon absorption contribute to the signal measured at the up-converted wavelength? This should be better explained.

In a simplified view, the upconversion process occurs due to the sequential absorption of temporally close single photons from intermediate excited energy states to higher energy states. This process is facilitated by the relatively long lifetimes of these metastable levels, which can extend to microseconds. When the intermediate levels are strongly populated, as is the case with our system where nanocrystals experience a large enhancement of the local optical field, the probability of absorbing a single and promoting an electron to the highest excited state becomes dominant. This phenomenon is described by equations (1-3) in reference 46 [*Das, A., et al. Nanophotonics* **9**, 1359–1371 (2020)]. Note that a similar principle of overcoming intensity saturation in nonlinear multiple-quantum-well metasurfaces for high-efficiency frequency upconversion has been demonstrated [Nefedkin, N. et al. *Advanced Materials* 2106902 (2021)].

To provide clarity on this aspect, we have included a specific section in the revised manuscript (lines 158-162):

“The scaling depends on the input power, as n -photon scaling occurs only for small absorption cross sections. The local optical field, which is proportional to the absorption cross section, influences the upconversion scaling. The nearly unitary exponent s observed indicates strong excitation due to the enhanced local optical field. As a result, the scaling reaches a saturation level at which single-photon promotion to excited states readily occurs^{47,48}.”

Comment #7: The enhancement effect here is localized at the edge. I am wondering how the size of array affects the concept:

- a. In the center-illuminated case (Fig. 2), the authors extracted QFW-BIC=5240. Noting this comes from a 1 mm² array, how does this compare with theoretical limit assuming infinite array, as is the case for the periodic unit cell simulation, for which the Q is not explicitly given? Also the paper says this Q is “in excellent superposition with the nanoparticle absorption centered at 800 nm” (Line 134), where is this calculation performed?
- b. Now in the edge-illuminated case (Fig. 3), would the size of the array affect, as the mode does not address the full area of the array? Can a similar level of enhancement be achieved with a small pattern? Numerical results may be used to illustrate this, and this discussion may help clarify the advantage of this work (if small pattern works equally well in this scheme).

(a) Any real-structure resonator can only sustain so called *quasi*-BICs (symmetry protected or accidental ones), not any ideal BICs with infinite quality factors. Regarding the radiative quality factor (Q_r) of our system, without loss, numerical simulations show an asymptotic approach to infinity (on the order of 10^{10}) as we refine the spectral resolution. However, when we include the imaginary refractive index component of silicon nitride to account for material absorption, the intrinsic Q factor is dominated by the absorption quality factor (Q_a), limiting the radiative Q factor to around 10^8 . We have clarified this aspect in the revised manuscript.

We have now used the term “quasi-BIC” in the revised manuscript to avoid any misleading. It is indeed important to note that in practical applications, including simulations, ideal BICs with infinite quality factors are not achievable and the observed resonances are quasi-BICs.

Furthermore, we apologize for the unclear writing in line 134. We meant to convey that the wavelength of the BIC is in spectral superposition with the absorption peak. For clarity, we have revised the main text and **Supplementary Fig. 2a**.

(b) We agree with the reviewer that a discussion of footprint scalability may be helpful to the discussion: additional results from smaller patterns of 300 and 100 μm have been included in the revised version (**Figure 3c**) and a study of the dependence of enhancement on pattern size (**Supplementary Fig. 24**, also below).

Main paper line 265 “Moreover, at the corner of the PCNS, by appropriately positioning the focused input beam and rotating the input polarization, it is possible to switch between vertically and horizontally emitted beams (**Fig. 4d** and **Supplementary Media 2**). The switch in emission direction is determined by aligning the input polarization perpendicular to the x -wave (or y -wave) depicted in **Fig. 4b**, thereby selectively exciting only one of them (beam 1 or 2) or both of them when at 45° of inclination. To investigate emission properties at a microscale level, the structure was reduced in size by a factor 3 and 10. This scaling down resulted in a manageable logarithmic slow variation of the output signal (**Supplementary Fig. 24**).”

Supplementary Materials, Section 9 “For each sample, we performed 30 acquisitions by moving the input beam along the side edge (**Supplementary Fig. 24**). It is important to note that the structure has a side length of 0.1 mm, consisting of 190 unit cells per side. The decrease in the signal is primarily attributed to the reduced

quality factor of the resonator. However, slight modification in the band dispersion due to the size of the structure can also contribute to signal variations.”

Supplementary Fig. 24. Scaling of the intensity with the size of the pattern.

The input is always the same size in the edge-illuminated case as in the non-edge case. The extent of the structure and its diffractive collective contribution to in-plane mode does not change as the input spot converges to the edge, while the effective index affecting mode coupling is the mechanism ascribed to the bands closing at the edge (see revised **Supplementary Fig. 7**).

We have also included the SNOM of the edge as **Supplementary Fig. 8**. Modifying the edge coupling increases Q_k by matching it with Q_a , which increases the achievable local field enhancement. The pattern size definitely affects the losses of the system, in particular the side leakage, which eventually affects the Q factor and also the spectral position of the BIC. However, this can be tuned with appropriate modifications during fabrication to compensate for the spectral shift).

Supplementary Fig. 8 | SNOM maps of the near field excited at frequency close to the BIC, inside the PCNS (top) and close to the boundary of the finite structure (bottom), with corresponding FFT maps. The resonance is perturbed only at the last unitary cell with an intensity reduced by about 50%.

Comment #8: The paragraph associated with equation (3) is very misleading and tedious as the main text of a candidate paper for consideration of publication in Nature. After a lot of redundancy [Line 185 – 200, beginning “Let us now evaluate the expected enhancement...”, while the finding, i.e. equation (3) is not meant to be what one expected], it only arrives at the trivial conclusion that equation (1) underestimates the enhancement factor, and one should use equation (2) [evaluated as equation (4)], which fits the experimental finding better. Yes, I agree that this comparison hints the nature of the resonance, but it can be at best mentioned briefly in one line and details should be given in the Supplementary.

We thank the reviewer for this suggestion, and we have moved the relevant details to the revised **Supplementary Materials**, specifically at pag. 44 in the paragraph titled **“Extracted Enhancement Factor parameters from Q factors.”**

Comment #9: The analogy of electromagnetically induced transparency should be a very important aspect as it appears in the abstract and is referred to multiple times, however, the main text does not present anything in that regard but simply redirects to the Supplementary. At least qualitative explanation should be given in the main text to inform the reader of why this analogy is necessary and relevant to the development of this concept presented here.

We thank the reviewer for these insightful suggestions. We included in detail in the revised manuscript the connection between EIT and FW-BIC.

Line 68 “In this work, we show that a Friedrich-Wintgen^{2,24–26} (FW) quasi-BIC can be achieved through supercritical coupling. We demonstrate that this phenomenon occurs in the presence of coupled-resonance-induced transparency^{27–29}.”

Line 80 “**Proximity of Friedrich-Wintgen quasi-BIC and electromagnetically induced transparency in the energy-momentum space**

In a typical experiment, we designed a dielectric nanostructure consisting of a transparent holey photonic crystal nanoslab (PCNS) covered with a conformal layer of upconversion nanoparticles^{30,31} (NPs) (**Fig. 1a**). The behavior of the system is described by the eigenvectors of the non-Hermitian Hamiltonian $\hat{H}_{\mathbf{k}} = j\hat{\Omega}(\mathbf{k}) - \hat{\Gamma}_r(\mathbf{k})$, which models the FW interference between photonic resonances. These eigenvectors correspond to vector TE-like and TM-like modes coupled to a single independent radiation channel, thus originally nonorthogonal, with a complex coupling coefficient κ_{12} measuring near- and far-field interference (ref. ^{24,32}). In the energy-momentum space, these modes evolve and eventually approach a FW-BIC point at a specific wavevector $\mathbf{k} = \mathbf{k}_{BIC}$ if the FW condition that diagonalizes $\hat{H}_{\mathbf{k}_{BIC}}$ can be satisfied by the system. Interestingly, even when initial leaky modes 1 and 2 have largely uneven radiative loss rates (γ_{r1} and γ_{r2}), it is possible to achieve a FW-BIC. When $\kappa_{12} > 2\gamma_r$, ($\gamma_r = \gamma_{r1} + \gamma_{r2}$), the coupled final modes of wavelengths λ_- and λ_+ split apart by Λ_g at the avoided crossing point and are not spectrally overlapped (**Fig. 1b, c**). One of these waves becomes a perfect dark mode (ideal FW-BIC) with a zero linewidth $\gamma_- = 0$ and a diverging lifetime $\tau_- = 1/\gamma_- = 2Q_{R1}/\omega_1 = \infty$. On the contrary, the low- Q_{R2} bright mode acquires all radiative losses with $\gamma_+ = \gamma_{r1} + \gamma_{r2}$. At the FW condition, $\hat{\Omega}(\mathbf{k})$ and $\hat{\Gamma}_r(\mathbf{k})$ are simultaneously diagonalized, resulting in asymptotically orthogonal modes with the dark mode decoupled from the radiation channel. However, in the vicinity of the FW-BIC, the modes are nonorthogonal and can be represented using the diagonal representation of $\hat{\Gamma}_r(\mathbf{k})$ with nonzero off-diagonal terms in $\hat{\Omega}(\mathbf{k})$ (the near field coupling) because of energy conservation^{28,33} (subradiant-superradiant model, Supplementary Sec. 1.2).

The same non-Hermitian Hamiltonian formalism of two modes coupled with single radiation channel leads to the coupled-resonance-induced transparency, which is the classical analogue of electromagnetically induced transparency (EIT) in photonic/plasmonic systems. This phenomenon provides exceptionally slow light and enhanced local optical field³⁴. If the PCNS geometry and dispersion are suitably engineered, EIT may occur close to the FW quasi-BIC, minimizing phase mismatch (**Fig. 1d**). Essentially, the FW quasi-BIC serves as the dark mode in the EIT process.

The TCMT model was discussed in Supplementary Sec. 1.3 and **Supplementary Fig. 1**. This two-wave-one-decay-port model was validated by thorough RCWA-based simulations of our specific nanostructure (Supplementary Sec. 1.4), thereby providing the evolution of the energy-momentum dispersion and the exact full 3D vector modes. The simulations also demonstrated the symmetry inversion of vector TE-like and TM-like modes at the avoided crossing and FW-BIC formation (**Supplementary Fig. 2-5**), showcased the onset of the EIT condition in proximity of the FW quasi-BIC in our tailored system, with small spectral detuning and an incidence angle separation of less than 0.5° (**Supplementary Fig. 6**).”

We have expanded the model and incorporated the role of EIT, as well as the proximity of EIT to the (quasi) FW-BIC. **Figure 1d** has been revised, and new **Supplementary Fig. 4-6** have been added to **Sec. 1.2-1.4** in the Supplementary Materials.

Fig. 1 | Principle of supercritical coupling and directive upconversion emission. **a**, Layout of the PCNS unit cell geometry and collimated upconversion radiation generated through supercritical edge BIC coupling. **b**, Schematic depicting internally coupled interfering resonances and their external coupling to the far-field. **c**, Scenario in non-edge pumping, where the coupled waves λ_- and λ_+ (dashed lines) and the linewidths (solid lines) shift apart by Λ_g due to intercoupling κ_{12} : λ_- becomes a FW quasi-BIC, with the laser wavelength λ_{in} being set. **d**, Tailoring the energy-momentum dispersion of the vertical asymmetric system to achieve simultaneously at the upconversion pump wavelength a FW quasi-BIC at $\sim k_{BIC}$ and EIT at $k_{BIC} + \delta k$, with minimal phase mismatch δk . **e**, Scenario in edge-pumping, where the decrease of κ_{12} at the PCNS edge causes spectral overlap between λ_- and λ_+ with the pump wavelength λ_{in} , reaching the condition of supercritical coupling. This diverts the far-field power more efficiently to the near-field drive, which can excite the high- Q_{R1} FW quasi-BIC beyond the critical coupling threshold. **f**, Intensity enhancement of the local optical field achievable in a single resonator, represented by the black line as a function of its radiative quality factor Q_R when the system dissipation is $Q_a = 5,000$. The red line represents the enhancement reached by the high- Q_{R1} FW quasi-BIC mode as a function of the coupling term, $Q_k \sim 1/\kappa_{12}$ with the bright mode. For supercritical coupling, the enhancement becomes $G^{sc} = 4(Q_a/Q_{R2}) \times G_{cr}$ (for $Q_{R2}/Q_a = 0.04$, $G^{sc} = 100G_{cr}$).

Comment #10: “Beam steering” does not seem to be an accurate word for describing the effect in Fig. 4. The emission comes out in two discrete channels on two edges, and the angle with respect to each edge is 90 degrees fixed.

We accept the criticism, and the word steering has been substituted with “directive” in the title: “Directive 10^8 -fold Enhanced Upconversion by Supercritical Edge Bound States in the Continuum.”

Comment #11: Other unclear figures, text and inappropriate referencing:

- Fig. 2 b and c are very unclear. The arrows for input laser should extend close to the PCNS to show the reader how in one case the PCNS is illuminated while the other case the bulk substrate is illuminated. The words IN and OUT in both b and c misleadingly imply “input” and “output” intuitively. The three colors in c imply three measurements are performed, but actually only two - green and orange are from the same measurement only indicating different spectral lines.
- In contrast to b, the main text includes technical details that fit better as methods or Supplementary, such as the fact that the thickness δh used in the simulation (Supplementary Fig. S6) is consistent with the experiment (main text Line 123).

- c. Lengthy and confusing abbreviations: UCPL, UCNP, and PCNS.
- d. Through Lines 185-220, too many Q's with different subscripts and different numbers were introduced, which is very confusing and does not help with the understanding. All the reader sees are numbers.
- e. 18 references on up conversion nanoparticles were cited (Refs. 26-43), out of which many may be unnecessary, e.g. Refs. 26-30 for multi-photon absorption through long-lived intermediate energy states, Refs. 38-40 for long-lived imaging probes, Refs. 33-36 for lasing.

a. We thank the reviewer for pointing out this inaccuracy. We have revised **Figure 2b** by adding the indication of the pump spot and changing the caption to clarify that only two measurements were carried out, inside and outside the PCNS, and that the colors green and red correspond to the associated spectral range (at 540 nm-green and 650 nm-red).

Fig. 2 | Experimental characterization of forward upconversion radiation. **a**, Experimental TE band diagram of the PCNS resonator, highlighting the high- Q FW-BIC. **b**, Experimental upconversion emission, with the left image focusing inside the PCNS area and the right image focusing outside of it. The red circle indicates the pump spot, and the inset provides a magnified view of the PCNS region. **c**, Corresponding upconversion spectrum along the forward direction inside (green and red curves) and outside the PCNS area (grey curve). **d**, Upconversion intensity excited inside the PCNS versus input power.

- b. We keep this technical detail since it is important for responding to reviewer #1 and reviewer #3.
- c. As suggested by the reviewer, we have deleted UCPL and UCNP from the main paper and left these acronyms only in the Supplementary Materials.
- d. The text has been revised for clarity and details have been moved to the Supplementary Materials.
- e. We have revised the references as suggested by the reviewer.

In closing, we have taken great care to thoroughly explain the underlying mechanisms and provide experimental validation for the phenomena reported in this work. We believe that the revised manuscript now offers a detailed account of the self-collimation effect and radiance enhancement in our system. We are grateful to the reviewer for bringing this aspect to our attention and helping us strengthen our work.

Referee #3 (Remarks to the Author):

The manuscript presented by Zito et al is certainly solid and interesting with new knowledge presented and potentially broader impact to the materials, photonics and physics community. The major merits include:

1. Reveal the existence and the key role of supercritical coupling in maximizing energy accumulation.
2. Achieve Friedrich-Wintgen BIC under supercritical coupling with significant local optical field enhancement
3. Get 8 orders of magnitude upconversion enhancement through supercritical edge bound states.
4. Achieve the controlled upconversion emission with negligible divergence, microscale width, and direction-steering capability.

We sincerely appreciate the time and effort the reviewer dedicated to reviewing our manuscript. The reviewer's enthusiastic and encouraging feedback is greatly valued and deeply appreciated.

Suggested improvements:

1. Whether the excitation wavelength is tunable by the structure design of the photonic crystal nanoslab (PCNS) in this system? This is the key to the broad usage of this technology.

We appreciate the reviewer's comment. We have addressed the tunability aspect in our revised manuscript. Specifically, we included **Supplementary Fig. 2d**, which demonstrates through linear fitting in simulations how the BIC frequency can be scaled, enabling its movement from the visible range to the infrared range and beyond. We have also provided experimental evidence of collimated radiation at the pump wavelength of 532 nm in **Supplementary Fig. S20e**.

Supplementary Fig. 2 | a, Numerical band diagrams (RCWA) of the PCNS showing a FW-BIC close to the avoided crossing in the momentum space (TE-like modes). **b**, Reflectance spectrum overlapped with the experimental absorption of NPs. **c**, FW-BIC displacement field intensity $|D|^2$ in the unit cell (left). **d**, Optical energy fraction in monolayer (1L) and bilayer (2L) NPs with air and silicone superstrates. **e**, RCWA simulation

and linear fit for spectral tuning of the quasi-FW-BIC position with the lattice constant a for radius of the circular hole $r=0.244a$, thus scaling with a (other parameters fixed).

2. Whether the emission enhancement is wavelength-dependent? It seems all the upconversion emission data is from the NaErF₄@NaYF₄ core-shell nanocrystals and only the green emission was inspected (figure 3a). It is better to collect the data for the red emission, which have a higher intensity than the green emission. Also, there are different kinds of emissions bands for core@shell@shell materials with the energy transfer of Nd→Yb→Tm→Gd→Eu (Figure S5E), which could be an ideal model to study the emission-wavelength-dependent upconversion enhancement. There is detailed Materials for the NaGdF₄:Nd/Yb(40/5%)@NaGdF₄:Yb/Tm(49/1%)@NaGdF₄:Eu(15%) core-shell-shell nanocrystals. The authors should add their detailed spectra Materials and the upconversion enhancement at different emission wavelengths in PCNS

We thank the reviewer for this suggestion. We examined the enhancement only for core-shell nanoparticles in the previous version. We have now considered the emission from all wavelengths as suggested. In the revised manuscript, we have included **Supplementary Fig. 18c**, which presents the results of the emission from core-shell-shell nanoparticles (blue + red) in the polarization switching experiment (revised **Figure 4d**).

Supplementary Fig. 18 | a, UCPL emission at 540 nm as a function of the PCNS number (10 points for each sample). **b**, Histogram of the intensity distribution (UCPL peak @540 nm) from core-shell NPs used for EF calculation. **c**, Intensity distribution of the emission achieved scanning the input beam along the full boundary of the PCNS for all major upconversion wavelengths.

3. Whether the orientation of the UCNPs on the PCNS affects the direction-steering performance?

We thank the reviewer for this suggestion. In our current study, the position of nanoparticles appeared completely random, and we did not have control over their orientation. Therefore, the direction of emission was solely controlled by the positioning of the pump beam and the input polarization.

4. Why were two layers of nanoparticles coated on the PCNS surface? Single-layer nanoparticles are commonly used in plasmonic enhancement (Wu et al. Nature Nanotechnology, 1110–1115, 2019)

and the fabrication of micro-laser (Shang et al. Nature Communications 6156, 2020). Therefore, single-layer nanoparticles may have a better performance here.

We thank the reviewer for this thoughtful suggestion. The main reason we focused on two layers is to achieve a uniform coverage, which ensures reproducibility and better agreement with numerical simulations that can be performed. In ref. Wu et al. [*Nature Nanotechnology* **14**, 1110–1115 (2019)], the cavity was formed by the sandwich with the nanocube filled with only a few NPs. In our case, the coverage must be uniform over a millimeter scale. It is possible to see below what happens with a single layer deposition. In most cases, the single layer leaves empty spots and a nonuniform filling of holes. Since the local field distribution overlapping with nanoparticles is relevant, the random coverage could affect performance. For this reason, perovskite lasers with BICs have often been fabricated from bulk materials directly patterned as photonic crystal slabs. Nonetheless, we considered several conditions for nanoparticle deposition, and the one reported in this paper worked better.

5. In Figure S5B, the optical absorption spectrum should be from NaErF4@NaYF4 core-shell nanocrystals instead of the core-shell-shell NPs. The authors may double-check the TEM images and other related characterizations.

We apologize for the error, and have corrected the caption.

6. It is better to provide the power density (W/cm²) instead of the power intensity (mW) of the laser.

As suggested by the reviewer, we have made changes accordingly in both Figures 2 and 3.

7. It is better for the authors to add TEM images of the core and core@shell nanoparticles. This information is important for the readers to repeat related experiments.

The required TEM image has been included.

8. On page 6, the author claimed that “In colloidal samples we measured $s = 1.2$ at an incident power on the order of 500 mW, while we found $s = 1.2$ at 1 mW on the PCNS.” It is better to have related data in the supporting Materials.

We have revised the text mentioning both energy and power density in the main paper. The behavior mentioned is the same as in **Supplementary Fig. 11e-f**.

In closing, this is an excellent report with good experimental data. The data support the main point of the paper, in the sense that through supercritical coupling in PCNS it is possible to significantly

enhance upconversion luminescence with direction-steering capability. The rational design and experimental validation of the supercritical coupling for emission enhancement in this work represents a sizeable breakthrough that justifies the publication in Nature.

Once again, we are grateful to the reviewer for the careful review and suggestions, for the positive feedback but also for the criticism that certainly allowed us to improve the technical presentation of this revised manuscript.

Reviewer Reports on the First Revision:

Referees' comments:

Referee #1 (Remarks to the Author):

After carefully checking the response to all reviewers' comments and criticisms, I can conclude that the authors did a comprehensive revision of the manuscript addressing the requested changes in response to the most of technical questions. The positive improvement of the revised text and supporting information includes the detailed description of the mode non-orthogonality condition, RCWA simulations of transmittance for an infinite and finite-size structure, analysis of different loss mechanisms contributing to quasi-BIC losses, explanation of the narrowband spatial divergence of the UCNP emission and statistical analysis. I believe that the authors did a great job with improving the mentioned sections, which contributed substantially to the quality of the manuscript.

Even so, the additional explanations regarding the validity of supercritical coupling model and its derivation via the TCMT equations are highly inconsistent and, moreover, contradict the results and well-known models used in the literature. From the extended derivation of the final TCMT model for the description of supercritical coupling presented in Supplementary Materials [Eqs. (33-34)], I can see that the authors literally copy-pasted the derivation of 3 different equivalent TCMT models (specifically, copy-pasted from Refs. [5, 10, 11] in SM) and produced their own new set of equations with an unphysical combination of quantities from all three models. More specific, the authors start with the coupled Hamiltonian of two initial non-orthogonal modes [Eq.(10)] and fully diagonalize it to find the true eigenmodes (resonant states) with complex frequencies, one of which can be tuned to become a FW-BIC. Then, they re-write the initial Hamiltonian from scratch in a different (second) representation [Eqs.(19-20)], where the initial modes are coupled via the far-field only. Finally, using the second Hamiltonian, they re-write it again in the third representation, with two initial modes interacting via the near-field coupling only. Then, the authors use the eigensolutions from the diagonalization of the first Hamiltonian [the one in Eq.(10)] as the initial non-yet-diagonalized coupled modes of the third Hamiltonian and arrive at their own completely new model [Eqs. (33-34)]. The authors assign incorrect physical meaning to the quantities of their model, forgetting about the relation to the initial physical meaning of these quantities imposed by the first three Hamiltonians. Thus, they assume unphysical relations between parameters of their model and arrive at invalid conclusions leading to supercritical coupling regime. I also note that the time convention changes from model to model, which is strange, e.g. in Eqs.[8-10] the convention is $\exp(j\omega t)$, and later in Eqs. [19, 25, 33,34] it is $\exp(-j\omega t)$.

To check the author's results, I reproduced the model described by Eqs. (33-34) of the SM starting from non-diagonalized Hamiltonian describing modes coupled in the near and far field with well-defined physical meaning of parameters. The resulting equations formally coincide with authors' Eqs. (33-34), while the physical meaning of parameters is critically different. I tried to reproduce the supercritical coupling regime with the accurate model, but it does not appear in the derivation. My detailed calculation is attached as an additional file for the sake of clarity and completeness.

Moreover, I was unable to reproduce the Eq.(2) of the main text [Eq.(38) of Supp.Mat.] by directly following the author steps, without thinking about model physical validity. Starting from Eqs. (33-34), I was able to reproduce Eqs. (35, 36), but not Eq. (37). More specific, the authors have a term $\tau_{\text{kappa}}^2 / (\tau_a (\tau_{R2} + \tau_a))$ in the denominator, while I get the term $\tau_{\text{kappa}}^2 / (\tau_a \tau_{R2})$. I do not see a reason for appearance of an additional τ_a assuming that $\omega_{\text{in}} = \omega_{\text{-}}$ and $\delta = 0$ following the authors' assumption. Without this term, the final Eq.(38) [Eq.(2) of the main text] changes dramatically and the supercritical coupling regime condition is no longer present.

Given that (i) the developed supercritical coupling regime cannot be analytically reproduced from the given model, (ii) the developed model is based on certain unphysical assumptions, I cannot recommend the manuscript for publication in Nature in any form. My overall impression that the supercritical coupling regime is an invalid concept in general, as it contradicts the general physical limitation of linear systems with harmonic excitation to be excited more efficiently than in the standard critical coupling regime (see Ref.[23] of the main text). Despite that, the experimental results of the manuscript are of very high technical quality, the demonstration of strong resonant directive upconversion emission from the sample edge is of high practical importance and the presented extended numerical data analysis supports the data. Thus, I would possibly recommend the manuscript for a more technical journal, like Nature Communications, in case the authors do a major revision by completely removing the supercritical coupling story from the text and focusing on the experimental results fitting them within a more standard theory.

As an alternative option, if the authors manage to carefully explain their results with a revised accurate theory based on correct assumptions, and indeed uncover that the UCNP emission enhancement is driven by a new physical concept, this paper might be still a candidate for publication in Nature. For this, they would need to show that their concept works in at least two representations of the Hamiltonian, as they all are equivalent.

Below, I list other more minor technical comments. I also attach an additional file providing a detailed analysis of different Hamiltonian representations, explaining, why the physical meaning of the model parameters in the authors' model is invalid and showing that the supercritical coupling regime does not follow from the TCMT equations.

Other comments:

(1) The authors discuss the importance of vertical asymmetry for realization of FW-BIC and strong coupling between TE and TM modes, citing Refs. [R. Mermet-Lyauoz, et al., arXiv:1905.03868 (2019); Phys Rev B 61, 2090–2101 (2000)]. I disagree with authors and note that a true FW-BIC with zero radiative losses requires complete vertical symmetry as was established in pioneering papers [Hsu, C.W. et al. Nature, 499(7457), pp.188-191 (2013); Zhen, B. et al. Physical review letters, 113(25), p.257401 (2014)]. In Ref. [Phys Rev B 61, 2090–2101 (2000)] the authors did not show that the losses of the dark FW mode reach complete zero at the avoided resonance crossing, as they did numerical simulations only. Ref. [R. Mermet-Lyauoz, et al., arXiv:1905.03868 (2019)] raised an extended discussion in the community back to 2019 and was concluded to be misleading, thus it cannot be a reliable source of information (please note that it was not published in a peer-reviewed journal since 2019). I also agree that from the point of view of experimental applications, a true FW-

BIC with zero radiation losses and a quasi-FW-BIC with near-zero radiation losses are indistinguishable. Still, the authors emphasise the role of asymmetry here "the vertical asymmetry, which promotes the interference of vector TE-like and TM-like modes and the formation of FW quasi-BIC, depends on the supporting substrate and cladding layer" which is a drawback rather than a benefit for FW-BIC Q factor value. I would suggest the authors to revise this statement. On the other hand, I agree that the strength of TE-TM coupling can be increased with the increase of the vertical asymmetry.

(2) In response to my comment, the authors added a comparison to symmetry-protected BICs (sp-BICs) to Supp.Mat. I disagree with the comparison, as they state that sp-BICs are not formed due to interference compared to FW-BICs. Actually, sp-BICs can be also understood as a complete destructive interference between two non-orthogonal coupled modes and the same Eq.(15) from SM can be applied to them. In this case, however, $\gamma_1 = \gamma_2$ due to symmetry considerations, thus Eq.(15) is satisfied for $\omega_1 = \omega_2$. Therefore, I would suggest the authors to clarify more why the observed effect of enhanced directive UCNP emission cannot be realized for sp-quasi-BICs for nearly-normal incidence.

(3) The authors provide a detailed explanation of the relation of the observed effects and EIT regime. In general, I agree with their explanations, however, I would note that the EIT is a more general effect that can be observed without a BIC for any two coupled modes with close frequencies and lifetimes. The authors' description makes an impression that the EIT is always driven by the BIC which is not true.

(4) The authors say that the excitation Q factor is usually very large even for large numerical aperture of the excitation objective, as follows from their own observations in previous works. I would not agree that it is a general case. In general, the excitation Q factor can be quite low for quasi-BICs as due to the presence of beam components with high k-vector the effective excited mode is an average in k-space in the vicinity of the quasi-BIC k-vector. If the quasi-BIC Q factor decreases quickly in its vicinity, the averaged value would be sufficiently smaller, reaching ~ 100 for objectives with $NA > 0.1$.

1 Temporal coupled-mode theory in different representations

Let us describe the resonant response of the photonic structure with two resonances and one port using the conventional temporal coupled-mode theory (TCMT)

$$\begin{aligned} \frac{d}{dt} \begin{pmatrix} A_1 \\ A_2 \end{pmatrix} &= -i \left(\widehat{\Omega} - i\widehat{\Gamma}_r - i\widehat{\Gamma}_a \right) \begin{pmatrix} A_1 \\ A_2 \end{pmatrix} + \begin{pmatrix} d_1 \\ d_2 \end{pmatrix} s_+; \\ s_- &= cs_+ + \begin{pmatrix} d_1 & d_2 \end{pmatrix} \begin{pmatrix} A_1 \\ A_2 \end{pmatrix}. \end{aligned} \quad (1)$$

Here, $A_{1,2}$ are the resonant amplitudes, $d_{1,2}$ are the coupling coefficients between the modes and incident field, $s_{+,-}$ are the amplitudes of the incident and transmitted fields, respectively, and c is the direct scattering matrix. The (2×2) matrices $\widehat{\Omega}, \widehat{\Gamma}_r, \widehat{\Gamma}_a$ describe the resonant frequencies, radiative and absorptive losses, respectively.

1.1 Representation with near- and far-field coupling

We start with the most standard representation of the TCMT equations Eq. (1) in the basis of closed channel modes coupled via the near field and far field. For this representation we use superscript (1) for all quantities.

$$\begin{aligned} \frac{d}{dt} \begin{pmatrix} A_1^{(1)} \\ A_2^{(1)} \end{pmatrix} &= -i \left(\widehat{\Omega}^{(1)} - i\widehat{\Gamma}_r^{(1)} - i\widehat{\Gamma}_a^{(1)} \right) \begin{pmatrix} A_1^{(1)} \\ A_2^{(1)} \end{pmatrix} + \begin{pmatrix} d_1^{(1)} \\ d_2^{(1)} \end{pmatrix} s_+; \\ s_- &= cs_+ + \begin{pmatrix} d_1^{(1)} & d_2^{(1)} \end{pmatrix} \begin{pmatrix} A_1^{(1)} \\ A_2^{(1)} \end{pmatrix}. \end{aligned} \quad (2)$$

In this basis, the matrices are defined as

$$\begin{aligned} \widehat{\Omega}^{(1)} &= \begin{pmatrix} \omega_1^{(1)} & \kappa^{(1)} \\ \kappa^{(1)} & \omega_2^{(1)} \end{pmatrix}; \\ \widehat{\Gamma}_r^{(1)} &= \begin{pmatrix} \gamma_1^{(1)} & \sqrt{\gamma_1^{(1)}\gamma_2^{(1)}} \\ \sqrt{\gamma_1^{(1)}\gamma_2^{(1)}} & \gamma_2^{(1)} \end{pmatrix}; \\ \widehat{\Gamma}_a^{(1)} &= \begin{pmatrix} \gamma_a & 0 \\ 0 & \gamma_a \end{pmatrix}, \\ \begin{pmatrix} d_1^{(1)} \\ d_2^{(1)} \end{pmatrix} &= \begin{pmatrix} \sqrt{\gamma_1^{(1)}} \\ \sqrt{\gamma_2^{(1)}} \end{pmatrix}. \end{aligned} \quad (3)$$

The resonant structure studied by the authors in the manuscript can be described by this set of equations. More specifically, the closed channel coupled

modes are the TE and TM modes of the effective waveguide slab with frequencies $\omega_1^{(1)}$ and $\omega_2^{(1)}$. The coupling coefficient $\kappa^{(1)}$ and radiative loss rates $\gamma_1^{(1)}$ and $\gamma_2^{(1)}$ can be derived from the grating waveguide coupled-mode theory [?,?]. The characteristic dependencies are

$$\begin{aligned}\kappa^{(1)} &\propto \int dz (\varepsilon - 1) \mathbf{E}_1^* \cdot \mathbf{E}_2, \\ \gamma_i^{(1)} &\propto \int dz \mathbf{E}_i^* \cdot \widehat{\mathbf{G}}_{\text{wg}} \cdot \mathbf{E}_1, \quad \text{for } i = 1, 2.\end{aligned}\tag{4}$$

Here, $\widehat{\mathbf{G}}_{\text{wg}}$ is the waveguide Green's function. The first equation qualitatively coincides with the equation for κ_{12} on page 19 of SM. We note that $\kappa^{(1)}$ describes the coupling strength between modes and defines the visible Rabi splitting in the transmittance spectra.

We note that used the energy conservation theorem to relate coupling vector elements $d_{1,2}^{(1)}$ to the radiative loss rates, assuming low absorption losses. In this representation, the energy conservation theorem is

$$|s_-|^2 = -\frac{d}{dt} \begin{pmatrix} (A_1^{(1)})^* & (A_2^{(1)})^* \end{pmatrix} \begin{pmatrix} A_1^{(1)} \\ A_2^{(1)} \end{pmatrix}.\tag{5}$$

1.2 Representation with far-field coupling only

We can transform Eq. (2) to the basis, where there is only the far-field coupling, reproducing Eqs. (19-20) of the SM of the authors' manuscript. For this, we diagonalize the real frequency matrix $\widehat{\Omega}^{(1)}$, construct a unitary matrix $\widehat{U}^{(12)}$ from the normalized eigenvectors of $\widehat{\Omega}^{(1)}$ and apply the unitary transformation to the Hamiltonian $\widehat{H}^{(1)} = \widehat{\Omega}^{(1)} - i\widehat{\Gamma}_r^{(1)} - i\widehat{\Gamma}_a^{(1)}$. For this representation we use superscript (2) for all quantities.

$$\begin{aligned}\frac{d}{dt} \begin{pmatrix} A_1^{(2)} \\ A_2^{(2)} \end{pmatrix} &= -i \left(\widehat{\Omega}^{(2)} - i\widehat{\Gamma}_r^{(2)} - i\widehat{\Gamma}_a^{(2)} \right) \begin{pmatrix} A_1^{(2)} \\ A_2^{(2)} \end{pmatrix} + \begin{pmatrix} d_1^{(2)} \\ d_2^{(2)} \end{pmatrix} s_+; \\ s_- &= cs_+ + \begin{pmatrix} d_1^{(2)} & d_2^{(2)} \end{pmatrix} \begin{pmatrix} A_1^{(2)} \\ A_2^{(2)} \end{pmatrix}.\end{aligned}\tag{6}$$

In the vicinity of the mode strong coupling regime ($|\omega_1^{(1)} - \omega_2^{(1)}| \ll \kappa^{(1)}$), the

matrices in the new basis can be written as

$$\begin{aligned}
\widehat{\Omega}^{(2)} &= (\widehat{U}^{(12)})^{-1} \widehat{\Omega}^{(1)} \widehat{U}^{(12)} = \begin{pmatrix} \omega_1^{(2)} & 0 \\ 0 & \omega_2^{(2)} \end{pmatrix}; \\
\widehat{\Gamma}_r^{(2)} &= (\widehat{U}^{(12)})^{-1} \widehat{\Gamma}_r^{(1)} \widehat{U}^{(12)} = \begin{pmatrix} \gamma_1^{(2)} & \sqrt{\gamma_1^{(2)} \gamma_2^{(2)}} \\ \sqrt{\gamma_1^{(2)} \gamma_2^{(2)}} & \gamma_2^{(2)} \end{pmatrix}; \\
\widehat{\Gamma}_a^{(2)} &= (\widehat{U}^{(12)})^{-1} \widehat{\Gamma}_a^{(1)} \widehat{U}^{(12)} = \begin{pmatrix} \gamma_a & 0 \\ 0 & \gamma_a \end{pmatrix}, \\
\begin{pmatrix} d_1^{(2)} \\ d_2^{(2)} \end{pmatrix} &= (\widehat{U}^{(12)})^{-1} \begin{pmatrix} d_1^{(1)} \\ d_2^{(1)} \end{pmatrix} = \begin{pmatrix} \sqrt{\gamma_1^{(2)}} \\ \sqrt{\gamma_2^{(2)}} \end{pmatrix}, \\
\begin{pmatrix} A_1^{(2)} \\ A_2^{(2)} \end{pmatrix} &= (\widehat{U}^{(12)})^{-1} \begin{pmatrix} A_1^{(1)} \\ A_2^{(1)} \end{pmatrix}.
\end{aligned} \tag{7}$$

Here, the transformation matrix $\widehat{U}^{(12)}$ for $(|\omega_1^{(1)} - \omega_2^{(1)}| \ll \kappa^{(1)})$ is given by

$$\widehat{U}^{(12)} = \begin{pmatrix} -\frac{1}{\sqrt{2}} & \frac{1}{\sqrt{2}} \\ \frac{1}{\sqrt{2}} & \frac{1}{\sqrt{2}} \end{pmatrix}. \tag{8}$$

Applying the transformation matrix, we get

$$\begin{aligned}
\omega_1^{(2)} &= \frac{\omega_1^{(1)} + \omega_2^{(1)}}{2} - \kappa^{(1)}, \\
\omega_2^{(2)} &= \frac{\omega_1^{(1)} + \omega_2^{(1)}}{2} + \kappa^{(1)}, \\
\gamma_1^{(2)} &= \frac{(\sqrt{\gamma_2^{(1)}} - \sqrt{\gamma_1^{(1)}})^2}{2}, \\
\gamma_2^{(2)} &= \frac{(\sqrt{\gamma_2^{(1)}} + \sqrt{\gamma_1^{(1)}})^2}{2}.
\end{aligned} \tag{9}$$

We note that the meaning of $\omega_{1,2}^{(2)}$ and $\gamma_{1,2}^{(2)}$ in this representation is different to the original frequencies and loss rates of coupled TM and TE modes of the effective waveguide slab.

We note that since $\widehat{U}^{(12)}$ is unitary, $d_{1,2}^{(2)}$ are still related to radiative loss rates, assuming low absorption losses. In this representation, the energy conservation theorem has the same form

$$|s_-|^2 = -\frac{d}{dt} \begin{pmatrix} (A_1^{(2)})^* & (A_2^{(2)})^* \end{pmatrix} \begin{pmatrix} A_1^{(2)} \\ A_2^{(2)} \end{pmatrix}. \tag{10}$$

1.3 Representation with near-field coupling only

We can also transform Eq. (2) to the other basis, where there is only the near-field coupling, reproducing Eqs. (25-26) of the SM of the authors' manuscript. For this, we diagonalize the radiative damping rate matrix $\widehat{\Gamma}_r^{(1)}$, construct a unitary matrix $\widehat{U}^{(13)}$ from the normalized eigenvectors of $\widehat{\Gamma}_r^{(1)}$ and apply the unitary transformation to the Hamiltonian $\widehat{H}^{(1)} = \widehat{\Omega}^{(1)} - i\widehat{\Gamma}_r^{(1)} - i\widehat{\Gamma}_a^{(1)}$. For this representation we use superscript (3) for all quantities.

$$\begin{aligned} \frac{d}{dt} \begin{pmatrix} A_1^{(3)} \\ A_2^{(3)} \end{pmatrix} &= -i \left(\widehat{\Omega}^{(3)} - i\widehat{\Gamma}_r^{(3)} - i\widehat{\Gamma}_a^{(3)} \right) \begin{pmatrix} A_1^{(3)} \\ A_2^{(3)} \end{pmatrix} + \begin{pmatrix} d_1^{(3)} \\ d_2^{(3)} \end{pmatrix} s_+; \\ s_- &= cs_+ + \begin{pmatrix} d_1^{(3)} & d_2^{(3)} \end{pmatrix} \begin{pmatrix} A_1^{(3)} \\ A_2^{(3)} \end{pmatrix}. \end{aligned} \quad (11)$$

In the vicinity of the mode strong coupling regime ($|\omega_1^{(1)} - \omega_2^{(1)}| \ll \kappa^{(1)}$), the matrices in this third basis can be written as

$$\begin{aligned} \widehat{\Omega}^{(3)} &= (\widehat{U}^{(13)})^{-1} \widehat{\Omega}^{(1)} \widehat{U}^{(13)} = \begin{pmatrix} \omega_1^{(3)} & \kappa^{(3)} \\ \kappa^{(3)} & \omega_2^{(3)} \end{pmatrix}; \\ \widehat{\Gamma}_r^{(3)} &= (\widehat{U}^{(13)})^{-1} \widehat{\Gamma}_r^{(1)} \widehat{U}^{(13)} = \begin{pmatrix} 0 & 0 \\ 0 & \gamma_2^{(3)} \end{pmatrix}; \\ \widehat{\Gamma}_a^{(3)} &= (\widehat{U}^{(13)})^{-1} \widehat{\Gamma}_a^{(1)} \widehat{U}^{(13)} = \begin{pmatrix} \gamma_a & 0 \\ 0 & \gamma_a \end{pmatrix}, \\ \begin{pmatrix} d_1^{(3)} \\ d_2^{(3)} \end{pmatrix} &= (\widehat{U}^{(13)})^{-1} \begin{pmatrix} d_1^{(1)} \\ d_2^{(1)} \end{pmatrix} = \begin{pmatrix} 0 \\ \sqrt{\gamma_2^{(3)}} \end{pmatrix}, \\ \begin{pmatrix} A_1^{(3)} \\ A_2^{(3)} \end{pmatrix} &= (\widehat{U}^{(13)})^{-1} \begin{pmatrix} A_1^{(1)} \\ A_2^{(1)} \end{pmatrix}. \end{aligned} \quad (12)$$

Here, the transformation matrix $\widehat{U}^{(13)}$ is given by

$$\widehat{U}^{(13)} = \begin{pmatrix} -\sqrt{\frac{\gamma_2^{(1)}}{\gamma_1^{(1)} + \gamma_2^{(1)}}} & \sqrt{\frac{\gamma_1^{(1)}}{\gamma_1^{(1)} + \gamma_2^{(1)}}} \\ \sqrt{\frac{\gamma_1^{(1)}}{\gamma_1^{(1)} + \gamma_2^{(1)}}} & \sqrt{\frac{\gamma_2^{(1)}}{\gamma_1^{(1)} + \gamma_2^{(1)}}} \end{pmatrix}. \quad (13)$$

Applying the transformation matrix, we get

$$\begin{aligned}
\omega_1^{(3)} &= \frac{\omega_1^{(1)} + \omega_2^{(1)}}{2} - \kappa^{(1)} \frac{2\sqrt{\gamma_1^{(1)}\gamma_2^{(1)}}}{\gamma_1^{(1)} + \gamma_2^{(1)}}, \\
\omega_2^{(3)} &= \frac{\omega_1^{(1)} + \omega_2^{(1)}}{2} + \kappa^{(1)} \frac{2\sqrt{\gamma_1^{(1)}\gamma_2^{(1)}}}{\gamma_1^{(1)} + \gamma_2^{(1)}}, \\
\kappa^{(3)} &= \kappa^{(1)} \frac{\gamma_1^{(1)} - \gamma_2^{(1)}}{\gamma_1^{(1)} + \gamma_2^{(1)}}, \\
\gamma_2^{(3)} &= \gamma_1^{(1)} + \gamma_2^{(1)}.
\end{aligned} \tag{14}$$

We note that the meaning of $\omega_{1,2}^{(3)}$, $\gamma_2^{(3)}$ and $\kappa^{(3)}$ in this representation is also different to the original frequencies and loss rates of coupled TM and TE modes of the effective waveguide slab.

We note that since $\widehat{U}^{(13)}$ is unitary, $d_{1,2}^{(3)}$ are still related to radiative loss rates, assuming low absorption losses. In this representation, the energy conservation theorem has the same form

$$|s_-|^2 = -\frac{d}{dt} \begin{pmatrix} (A_1^{(3)})^* & (A_2^{(3)})^* \end{pmatrix} \begin{pmatrix} A_1^{(3)} \\ A_2^{(3)} \end{pmatrix}. \tag{15}$$

1.4 Representation with fully diagonal Hamiltonian

Finally, we can fully diagonalize $\widehat{H}^{(1)}$ in Eq. (2), reproducing Eqs. (10-16) of the SM of the authors' manuscript. For this, we diagonalize $\widehat{H}^{(1)}$, construct a matrix $\widehat{U}^{(14)}$ from the normalized eigenvectors of $\widehat{H}^{(1)}$ and apply the unitary transformation to all functions in Eq. (2). For this representation we use superscript (4) for all quantities. We note that in this case the transformation is no longer unitary, as the Hamiltonian $\widehat{H}^{(1)}$ is non-Hermitian.

$$\begin{aligned}
\frac{d}{dt} \begin{pmatrix} A_1^{(4)} \\ A_2^{(4)} \end{pmatrix} &= -i\widehat{H}^{(4)} \begin{pmatrix} A_1^{(4)} \\ A_2^{(4)} \end{pmatrix} + \begin{pmatrix} K_1^{(4)} \\ K_2^{(4)} \end{pmatrix} s_+; \\
s_- &= cs_+ + \begin{pmatrix} d_1^{(4)} & d_2^{(4)} \end{pmatrix} \begin{pmatrix} A_1^{(4)} \\ A_2^{(4)} \end{pmatrix}.
\end{aligned} \tag{16}$$

The matrices in this fourth basis can be written as

$$\begin{aligned}
\widehat{H}^{(4)} &= (\widehat{U}^{(14)})^{-1} \widehat{H}^{(1)} \widehat{U}^{(14)} = \begin{pmatrix} \omega_1^{(4)} - i\gamma_1^{(4)} & 0 \\ 0 & \omega_2^{(4)} - i\gamma_2^{(4)} \end{pmatrix}, \\
\widehat{\Gamma}_a^{(4)} &= (\widehat{U}^{(14)})^{-1} \widehat{\Gamma}_a^{(1)} \widehat{U}^{(14)} = \begin{pmatrix} \gamma_a & 0 \\ 0 & \gamma_a \end{pmatrix}, \\
\begin{pmatrix} k_1^{(4)} \\ k_2^{(4)} \end{pmatrix} &= (\widehat{U}^{(14)})^{-1} \begin{pmatrix} d_1^{(1)} \\ d_2^{(1)} \end{pmatrix}, \\
\begin{pmatrix} d_1^{(4)} & d_2^{(4)} \end{pmatrix} &= \begin{pmatrix} d_1^{(1)} & d_2^{(1)} \end{pmatrix} \widehat{U}^{(14)}, \\
\begin{pmatrix} A_1^{(4)} \\ A_2^{(4)} \end{pmatrix} &= (\widehat{U}^{(14)})^{-1} \begin{pmatrix} A_1^{(1)} \\ A_2^{(1)} \end{pmatrix}.
\end{aligned} \tag{17}$$

Applying the transformation matrix, we get

$$\begin{aligned}
\omega_{1,2}^{(4)} - i\gamma_{1,2}^{(4)} &= \frac{\omega_1^{(1)} + \omega_2^{(1)}}{2} - i \frac{\gamma_1^{(1)} + \gamma_2^{(1)}}{2} \mp \\
&\mp \frac{1}{2} \sqrt{\left[(\omega_1^{(1)} - \omega_2^{(1)}) - i(\gamma_1^{(1)} - \gamma_2^{(1)}) \right]^2 + 4 \left(\kappa^{(1)} - i\sqrt{\gamma_1^{(1)} \gamma_2^{(1)}} \right)^2}.
\end{aligned} \tag{18}$$

The quasi-BIC condition can be achieved in the regime

$$\kappa^{(1)} \left(\gamma_1^{(1)} - \gamma_2^{(1)} \right) = \sqrt{\gamma_1^{(1)} \gamma_2^{(1)}} \left(\omega_1^{(1)} - \omega_2^{(1)} \right). \tag{19}$$

Then, the diagonalized mode frequencies will be

$$\begin{aligned}
\omega_1^{(4)} &= \frac{\omega_1^{(1)} + \omega_2^{(1)}}{2} + \kappa^{(1)} \frac{\gamma_1^{(1)} + \gamma_2^{(1)}}{2\sqrt{\gamma_1^{(1)} \gamma_2^{(1)}}}, \\
\omega_2^{(4)} &= \frac{\omega_1^{(1)} + \omega_2^{(1)}}{2} + \kappa^{(1)} \frac{\gamma_1^{(1)} + \gamma_2^{(1)}}{2\sqrt{\gamma_1^{(1)} \gamma_2^{(1)}}}, \\
\gamma_1^{(4)} &= 0, \\
\gamma_2^{(4)} &= \gamma_1^{(1)} + \gamma_2^{(1)}.
\end{aligned} \tag{20}$$

In this case, the amplitude $A_1^{(4)}$ describes the BIC, and $A_2^{(4)}$ describes its bright mode partner.

We note that since $\widehat{U}^{(14)}$ is not unitary, $(\widehat{U}^{(14)})^\dagger \neq (\widehat{U}^{(14)})^{-1}$ and the energy conservation theorem changes to

$$|s_-|^2 = -\frac{d}{dt} \begin{pmatrix} (A_1^{(4)})^* & (A_2^{(4)})^* \end{pmatrix} (\widehat{U}^{(14)})^\dagger \widehat{U}^{(14)} \begin{pmatrix} A_1^{(4)} \\ A_2^{(4)} \end{pmatrix}. \tag{21}$$

1.5 Comparison between representations and analysis of authors' model

First, the authors used $A_{1,2}^{(4)}$ that describe the BIC and bright mode and obey Eqs. (16) in their Eqs. (33-34) of SM equivalent to our Eqs. (11). In other words, the authors assumed $A_{1,2}^{(3)} = A_{1,2}^{(4)}$ and $\omega_{1,2}^{(3)} = \omega_{1,2}^{(4)}$, $\gamma_{1,2}^{(3)} = \gamma_{1,2}^{(4)}$.

Second, the authors expressed the parameters in Eqs. (33-34) via the parameters of initial coupled TE and TM modes $\omega_{1,2}^{(1)}$, $\gamma_{1,2}^{(1)}$, $\kappa^{(1)}$ in an incorrect way, as we can see from comparison to our Eqs. (23). In particular, the frequency detuning $\delta^{(3)} = \omega_1^{(3)} - \omega_2^{(3)} = 2\kappa^{(1)} \frac{2\sqrt{\gamma_1^{(1)}\gamma_2^{(1)}}}{\gamma_1^{(1)} + \gamma_2^{(1)}}$ and cannot be assumed as zero which was done in Eq. (37) of SM. This is because representation (3) is only a specific choice of basis, allowing to convey analysis of coupling via the near-field coefficients, but the actual frequencies and coupling constants do not exactly equal to the ones of initial coupled waveguide modes.

Finally, we can try to achieve the supercritical coupling regime from the correct model solving Eqs. 11 for incident frequency ω_{in} for the dark mode amplitude $A_1^{(3)}$

$$A_1^{(3)} = \left(\frac{-i\kappa^{(3)}}{-i(\omega_{\text{in}} - \omega_1^{(3)}) + \gamma_a} \right) \frac{\sqrt{\gamma_2^{(3)}} s_+}{-i(\omega_{\text{in}} - \omega_2^{(3)}) + \gamma_2^{(3)} + \gamma_a + \frac{(\kappa^{(3)})^2}{-i(\omega_{\text{in}} - \omega_1^{(3)}) + \gamma_a}}. \quad (22)$$

For further analysis, we assume that we are near the BIC in the parameter space (k-space), so the condition Eq. (19) is almost fulfilled. In the vicinity of the avoided resonance crossing ($|\omega_1^{(1)} - \omega_2^{(1)}| \ll \kappa^{(1)}$) the BIC condition can be fulfilled only if $\delta\gamma^{(1)} = \gamma_1^{(1)} - \gamma_2^{(1)} \ll \sqrt{\gamma_1^{(1)}\gamma_2^{(1)}}$. Thus, we write $\gamma_{1,2}^{(1)} = \gamma_0^{(1)} \pm \delta\gamma^{(1)}/2$. We also define $\omega_{1,2}^{(1)} = \omega_0^{(1)} \pm \delta\omega^{(1)}/2$. Then, the parameters of Eqs. 11 are simplified as

$$\begin{aligned} \omega_1^{(3)} &= \omega_0^{(1)} - \kappa^{(1)}, \\ \omega_2^{(3)} &= \omega_0^{(1)} + \kappa^{(1)}, \\ \kappa^{(3)} &= \kappa^{(1)} \frac{\delta\gamma^{(1)}}{2\gamma_0^{(1)}}, \\ \gamma_2^{(3)} &= 2\gamma_0^{(1)}. \end{aligned} \quad (23)$$

Following the authors, we assume the resonant case $\omega_{\text{in}} = (\omega_1^{(3)} + \omega_2^{(3)})/2 = \omega_0^{(1)}$. Then, we can write for the field enhancement

$$\left| \frac{A_1^{(3)}}{s_+} \right|^2 = \frac{2\gamma_0^{(1)}(\kappa^{(3)})^2}{[(\kappa^{(1)})^2 + \gamma_a^2 + 2\gamma_a\gamma_0^{(1)}]^2 + 4(\kappa^{(1)})^2(\gamma_0^{(1)})^2}. \quad (24)$$

Finally, following the authors, we assume that $\kappa^{(1)} \simeq \gamma_a$ and $\gamma_a \ll \gamma_0^{(1)}$, thus

$\kappa^{(1)} \ll \gamma_0^{(1)}$. Thus, we get

$$\left| \frac{A_1^{(3)}}{s_+} \right|^2 = \frac{(\delta\gamma^{(1)})^2}{(2\gamma_0^{(1)})^3} \frac{(\kappa^{(1)})^2}{[(\kappa^{(1)})^2 + \gamma_a^2]}. \quad (25)$$

One can see, that there is no optimal condition for this quantity.

Referee #2 (Remarks to the Author):

The authors have provided comprehensive responses to all the questions raised. In my opinion, considering the quality of their answers, the paper is suitable for acceptance in the journal Nature. I also have some additional comments, which are as follows:

Error bars should be defined in all figures, and the authors should include a sentence describing how it measures it. For example, Fig 2(d), Fig (2c), fig 4(d)

Additional comments

Figure 2 lambda-in is unclear to me and should be better illustrated.

Referee #3 (Remarks to the Author):

In the revised iteration, the authors have diligently addressed all the suggestions provided, leading to a marked enhancement in the overall quality of the manuscript. I am impressed by the incorporation of novel data in Figure 4, encompassing Iso-frequency far-field and near-field intensity mappings, alongside the demonstration of collimated emissions spanning diverse visible ranges on a centimeter scale. Following a comprehensive review of the revised manuscript, I agree with its publication in Nature.

<Responses to the reviewers' comments (Nature manuscript 2022-10-16862A-Z)

(2-nd Round)

Title: Directive 10^8 -fold Enhanced Upconversion by Supercritical Edge Bound States in the Continuum

Changes in the revised manuscript as a response to the reviewers' comments are highlighted in red color and clarifications regarding the reviewer's comments are provided in blue color.

Referee #1 (Remarks to the Author):

After carefully checking the response to all reviewers' comments and criticisms, I can conclude that the authors did a comprehensive revision of the manuscript addressing the requested changes in response to the most of technical questions. The positive improvement of the revised text and supporting information includes the detailed description of the mode non-orthogonality condition, RCWA simulations of transmittance for an infinite and finite-size structure, analysis of different loss mechanisms contributing to quasi-BIC losses, explanation of the narrowband spatial divergence of the UCNP emission and statistical analysis. I believe that the authors did a great job with improving the mentioned sections, which contributed substantially to the quality of the manuscript.

We are grateful to the reviewer for the time and effort dedicated to our manuscript, for the detailed calculations in the additional file provided, and for the insightful suggestions.

Comment #1. Even so, the additional explanations regarding the validity of supercritical coupling model and its derivation via the TCMT equations are highly inconsistent and, moreover, contradict the results and well-known models used in the literature. From the extended derivation of the final TCMT model for the description of supercritical coupling presented in Supplementary Materials [Eqs. (33-34)], I can see that the authors literally copy-pasted the derivation of 3 different equivalent TCMT models (specifically, copy-pasted from Refs. [5, 10, 11] in SM) and produced their own new set of equations with an unphysical combination of quantities from all three models. More specific, the authors start with the coupled Hamiltonian of two initial non-orthogonal modes [Eq.(10)] and fully diagonalize it to find the true eigenmodes (resonant states) with complex frequencies, one of which can be tuned to become a FW-BIC. Then, they re-write the initial Hamiltonian from scratch in a different (second) representation [Eqs.(19-20)], where the initial modes are coupled via the far-field only. Finally, using the second Hamiltonian, they re-write it again in the third representation, with two initial modes interacting via the near-field coupling only. Then, the authors use the eigensolutions from the diagonalization of the first Hamiltonian [the one in Eq.(10)] as the initial non-yet-diagonalized coupled modes of the third Hamiltonian and arrive at their own completely new model [Eqs. (33-34)]. The authors assign incorrect physical meaning to the quantities of their model, forgetting about the relation to the initial physical meaning of these quantities imposed by the first three Hamiltonians. Thus, they assume unphysical relations between parameters of their model and arrive at invalid conclusions leading to supercritical coupling regime. I also note that the time convention changes from model to model, which is strange, e.g. in Eqs.[8-10] the convention is $\exp(j\omega t)$, and later in Eqs. [19, 25, 33,34] it is $\exp(-j\omega t)$.

The reviewer's comments, in particular those regarding the representation change and the coupling limit (below discussed), have helped to improve the quality of our presentation and the general significance of our paper in this revised version. In response to the specific points raised by the reviewer, we would like to stress that in the revised version we have removed all inconsistencies due to misused symbols in various representations. We have improved the problem statement and its

presentation by clearly declaring all variables and parameters in various representations and their meaning, with the corresponding Hamiltonians, which are also included in new **Supplementary Figure 1**. In particular, we note that Supplementary Eqs. 33,34 involved the final coupled modes, not the initial uncoupled modes, as now clarified. Other symbolic representations of physical quantities and the time convention $\exp(j\omega t)$ have been fully revised. We have explicitly acknowledged the references given in our paper to invite the reader to look for more information in those well-known papers. It was not our intention to copy the derivation of 3 different equivalent TCMT models. Our goal was to establish a proper connection between the perturbed FW quasi-BIC Hamiltonian and the EIT model by pointing out that this perturbed Hamiltonian near a FW-BIC requires nonzero off-diagonal terms as in the EIT model and by recognizing the dark mode of the EIT transparency frequency as a mode of an unexpected large local field that must be somehow externally pumped despite being a dark mode. We apologize for the misrepresentation of our model, which we have now corrected.

- 1) The diagonalized Hamiltonian occurring under the Friedrich-Wintgen condition is presented in terms of close channel parameters (initial uncoupled modes). It states that this is possible only because the final dark mode is totally decoupled from the radiation channel, otherwise it would not obey energy conservation. The modes must be internally coupled (by near or far field) when both are connected via nonzero radiative losses to the input radiation channel to ensure energy conservation. However, since the ideal FW-BIC has exactly zero radiative loss, energy conservation is preserved and the modes can become orthogonal [W. Suh, Z. Wang, S. Fan. Temporal coupled-mode theory and the presence of non-orthogonal modes in lossless multimode cavities. IEEE Journal of Quantum Electronics 40, 1511 (2004)].
- 2) After that, we explicitly reconstruct the occurrence of EIT in the diagonal frequency representation, for new amplitudes and new corresponding parameters, to find the transparency frequency (which is easier in this representation). We skipped the link with the previous parameters because it is unnecessary. Then we changed the representation to the diagonal decay to recognize that the **dark mode is at the transparency frequency** (here, the connection with the diagonal decay parameters is given). This was the point we wanted to highlight in our previous version: EIT means an enhanced field at the transparency frequency because of slow light due to fast dispersion. That means that the dark mode, despite being dark and poorly coupled to the radiation channel, can gain an enhanced local field! How is this possible? Supercritical coupling will explain this.
- 3) We now turn to the **diagonalized** Hamiltonian (**thus final modes**) of the ideal FW-BIC and **consider a small perturbation by slightly moving the wavevector away from the BIC wavevector**, thus changing the frequencies, the radiative terms, and the mode coupling to the wavevector (the Hamiltonian is a function of momentum). Away from the BIC wavevector, the mode is a FW quasi-BIC **and takes on nonzero coupling radiative loss. Since it is coupled to the radiation channel, it must necessarily also be coupled internally to the bright partner by nonzero off-diagonal terms**, for example in the near-field representation (without loss of generality). **This is valid for the final modes** and leads to the **Supplementary Eqs. 33-34** also presented in our previous version.

This clarifies the meaning of all physical parameters involved and hopefully removes the confusion of our earlier presentation. We have also added a sketch in **Fig. 1c** and **Supplementary Fig. 1a** to clarify that FW-BIC is displaced on a wavevector close to, but not coincident with the EIT wavevector, which is at the avoided crossing, because orthogonal modes (ideal FW-BIC) cannot be simultaneously nonorthogonal (EIT or FW quasi-BIC). It is useful to point out a significant improvement we have provided to the model. In the introduction of the main manuscript, we say, at line 50

“When the radiative loss becomes negligible (diverging Q_r), the storable energy becomes limited only by unavoidable nonradiative losses.”

This is a universal limit of any resonator that cannot be overcome, which of course obeys energy conservation. The maximum achievable physical enhancement in a resonator is dictated by the lowest Q factor - the larger loss. Thus, the ultimate limit is given by unavoidable nonradiative losses, due to material absorption, as instance, when the light-matter interaction is the aim. In our current model, energy conservation and the balance of energy exchange between modes are preserved. The limit for the behavior of the uncoupled modes $Q_\kappa \rightarrow \infty$ ($\kappa_{12} \rightarrow 0$) is retrieved with a poor pumping of the dark mode as predicted by the standard model of single resonances. As suggested by the reviewer, we have now clarified that it is extremely important and innovative to excite the dark mode (even in the case of a nearly-divergent radiative quality factor) to the maximum physical enhancement possible in the resonant system imposed by the nonradiative losses. This occurs under the supercritical coupling condition, which is now correctly determined as $\bar{Q}_\kappa = \sqrt{Q_{R2}Q_a}$ (with the exact parameter range in which it is valid) and used in the revised experimental section of the paper to interpret the results about the extraordinary enhancement of the dark mode. This is a breakthrough: the near-field coupling mechanism with the bright partner in a coupled dark-bright mode system is capable of restoring the enhancement to the maximum level, always in the dark mode, even in cases of mismatched quality factors, impossible for a single uncoupled dark mode, which could gain only a fraction, i.e., an order-of-magnitude attenuated enhancement with respect to the maximum possible level in the resonator. In addition, the results of standard single mode coupling and critical coupling are included in the model. Having clarified the presentation, we can now discuss below the major point raised by the reviewer regarding our calculation.

Comment #2. To check the author's results, I reproduced the model described by Eqs. (33-34) of the SM starting from nondiagonalized Hamiltonian describing modes coupled in the near and far field with well-defined physical meaning of parameters. The resulting equations formally coincide with authors' Eqs. (33-34), while the physical meaning of parameters is critically different. I tried to reproduce the supercritical coupling regime with the accurate model, but it does not appear in the derivation. My detailed calculation is attached as an additional file for the sake of clarity and completeness.

Moreover, I was unable to reproduce the Eq.(2) of the main text [Eq.(38) of Supp.Mat.] by directly following the author steps, without thinking about model physical validity. Starting from Eqs. (33-34), I was able to reproduce Eqs. (35, 36), but not Eq. (37). More specific, the authors have a term $\tau_\kappa^2 / (\tau_a * (\tau_{R2} + \tau_a))$ in the denominator, while I get the term $\tau_\kappa^2 / (\tau_a * \tau_{R2})$. I do not see a reason for appearance of an additional τ_a assuming that $\omega_{in} = \omega_{-}$ and $\delta = 0$ following the authors' assumption. Without this term, the final Eq.(38) [Eq.(2) of the main text] changes dramatically and the supercritical coupling regime condition is no longer present.

We are extremely thankful to the reviewer's thorough evaluation of our results, which has inspired us to formulate a general model without any approximation. This has resulted in the derivation of analytically exact solutions, as presented in new **Supplementary Eqs. 35,36**, describing the dark and bright modes, respectively. The reviewer rightly pointed out an incongruence related to the limit ($\kappa_{12} \rightarrow 0$), which could imply a lack of coupling among the modes. This raised the question of how the coupling could facilitate the transfer of the drive field to the dark mode. In our work, this approximation was employed solely to simplify the analytical expression, where we assumed that the frequency terms ($\omega_{in} - \omega_{\pm}$) could be considered negligible due to small detuning, as is common in

other papers. Notably, the term κ_{12} was not removed elsewhere. But this approximation introduced an inconsistency that could have limited the validity of former Eq. (2), which we had not initially recognized. Additionally, an error occurred during the conversion from LaTeX to Word, leading to an incorrect presentation of supplementary Eq. (37), although supplementary Eq. (38) was correctly written, albeit with an inappropriate limit of validity ($Q_{R2} \gg Q_a$). **In our revised version, all of these issues have been addressed.**

Indeed, we understood that the analytical simplification with approximation was not necessary and could have been misleading. Consequently, **we have now included every term correctly in our new solutions.** In the reviewer's calculations, in section 1.5 "**Comparison between representations and analysis of authors' model**" of the reviewer's additional file [reviewer-Eq. 22-25], there was an attempt to substitute the **ideal** FW-BIC condition, which relates the initial uncoupled parameters with superscripts (1), able to diagonalize the Hamiltonian and making it Hermitian despite a single radiation channel (therefore providing **orthogonal final modes**, possible only because the dark mode becomes completely decoupled from radiation), in the representation of **nonorthogonal final modes** with nonzero off-diagonal frequency terms κ_{12} , which was related to a different wavevector from the FW-wavevector in our model. However, this situation is not permissible, and we have taken the necessary steps to indicate the Hamiltonians, different wavevectors, and the representation that must be considered. In addition, the reviewer considered as our calculation's optimum coupling condition " $\kappa^{(1)} \simeq \gamma_a$ and $\gamma_a \ll \gamma_0^{(1)}$ " to arrive at reviewer-Eq. 25, which was not our position, since we were referring to the final-mode near-field coupling κ_{12} with $Q_a \ll Q_{R1}$ (the final radiative quality factor R1 of the coupled **quasi** dark mode) that would have superscripts (3) in reviewer's notation. Please note indeed that considering the EIT [near-field representation (3)] and FW condition for initial modes (1) simultaneous would imply that the FW condition occurs at the avoided crossing, possible only if $\gamma_1^{(1)} - \gamma_2^{(1)} = 0$, thus producing $\kappa^{(3)} = 0$ (reviewer-Eq. 23), that means zero near-field coupling of the final modes (since orthogonal).

We have explicitly stated that **the Hamiltonian for supercritical coupling is a perturbation of the FW-BIC diagonal Hamiltonian**, as mentioned in **line 73** of the main manuscript, and that the parameters in it describe the **final modes of the system**. When we move away from the ideal FW wavevector, the **Hamiltonian of the final modes must have nonzero κ_{12}** : there is no representation with simultaneous frequency and decay matrices that can be diagonal, because the modes are both radiatively coupled to a single radiation channel, indeed the final mode with higher radiative Q-factor has **finite Q_{R1} factor** and is coupled to the radiation channel through it (finite linewidth of the final mode as clearly shown in the simulations). The occurrence of **nonzero κ_{12}** also applies for the Hamiltonian described in our **Supplementary Eq. (25)** describing the near-field representation of the EIT process, which is given for the final modes of the system, not initial uncoupled modes. This is summarized in our revised Supplementary Materials, providing an overview of all the calculations conducted.

SM Sec.1.2-1.3 page 6 "As a nonzero coupling with the single decay port exists, the two eigenvectors in the resonator system will always be nonorthogonal [10].

This implies that the modes are generally nonorthogonal in momentum space. However, if a single radiation channel is involved, they can satisfy the orthogonality condition at a specific point in momentum space where $\hat{\Omega}\hat{\Gamma}_r = \hat{\Gamma}_r\hat{\Omega}$. This point is referred to as the ideal FW-BIC point \mathbf{k}_{BIC} when the Hamiltonian ($\hat{H} = j\hat{\Omega} - \hat{\Gamma}_r$, below defined) has a purely imaginary eigenvalue (or equivalently $\hat{\Omega} + j\hat{\Gamma}_r$ a purely real eigenvalue). This allows for the simultaneous diagonalization of the Hermitian matrices $\hat{\Omega}, \hat{\Gamma}_r$.

The Hamiltonian of a two-waves-two-ports system is represented as:

$$\hat{H} = j \begin{pmatrix} \omega_1 & \kappa \\ \kappa & \omega_2 \end{pmatrix} - \begin{pmatrix} \gamma_{r1} & X \\ X^* & \gamma_{r2} \end{pmatrix} = j \begin{pmatrix} \omega_1 + j\gamma_{r1} & \kappa + jX \\ \kappa + jX^* & \omega_2 + j\gamma_{r2} \end{pmatrix} \equiv j \begin{pmatrix} \tilde{\omega}_1 & \tilde{\omega}_{12} \\ \tilde{\omega}_{21} & \tilde{\omega}_2 \end{pmatrix}, \quad (10)$$

where κ measures the near-field coupling and X represents the coupling mediated by the continuum between the two **closed, uncoupled channel** resonances of frequencies ω_1 and ω_2 . Following the calculation in [5,10], X can be expressed as

$$X = \sqrt{\gamma_{r1}\gamma_{r2}} e^{j\psi}, \quad (11)$$

where the phase angle ψ describes the relative phase of the coupling with the open channel and in general with the two ports (up and down). The eigenvalues **of both diagonal frequency and decay matrices of the Hamiltonian at the BIC point, defined by**

$$\hat{H}^r(\mathbf{k}_{\text{BIC}}) = \hat{\Omega} + j\hat{\Gamma}_r = \begin{pmatrix} \tilde{\omega}_+ & 0 \\ 0 & \tilde{\omega}_- \end{pmatrix} + j \begin{pmatrix} \tilde{\gamma}_+ & 0 \\ 0 & \tilde{\gamma}_- \end{pmatrix}, \quad (12) \quad \text{and}$$

associated with the collective modes \tilde{A}_+ , \tilde{A}_- , are related to the **uncoupled modes frequency and decay rates by:**

$$\tilde{\omega}_\pm + j\tilde{\gamma}_\pm = (\omega_1 + \omega_2)/2 + j(\gamma_{r1} + \gamma_{r2})/2 \quad (13)$$

$$\pm \frac{1}{2} \sqrt{[(\omega_1 - \omega_2) + j(\gamma_{r1} - \gamma_{r2})]^2 + 4(\kappa + j\sqrt{\gamma_{r1}\gamma_{r2}}e^{j\psi})^2}. \quad (14)$$

This relation allows one to determine the **asymptotic Friedrich-Wintgen's condition as a function of the uncoupled modes frequencies, decay rates and coupling rate among closed channel modes κ :**

$$\kappa(\gamma_{r1} - \gamma_{r2}) = \sqrt{\gamma_{r1}\gamma_{r2}}e^{j\psi}(\omega_1 - \omega_2), \quad (15)$$

$$\psi = m\pi, \quad m \in \mathcal{Z} \quad (16)$$

Substituting $(\gamma_{r1} - \gamma_{r2})$ derived from Eq. (15) in Eq. (14), it is possible to find that the third term with the square root is exactly equal to the second term and cancels, or adds with it, depending on the sign \pm . The dark mode acquires ideally zero radiation loss (say $\tilde{\omega}_-$ without loss of generality). At this condition, the eigenvalues are:

$$\tilde{\omega}_+ + j\tilde{\gamma}_+ = \frac{\omega_1 + \omega_2}{2} + \frac{\kappa(\gamma_{r1} + \gamma_{r2})}{2\sqrt{\gamma_{r1}\gamma_{r2}}e^{j\psi}} + j(\gamma_{r1} + \gamma_{r2}) \quad (17)$$

$$\tilde{\omega}_- + j\tilde{\gamma}_- = \frac{\omega_1 + \omega_2}{2} - \frac{\kappa(\gamma_{r1} + \gamma_{r2})}{2\sqrt{\gamma_{r1}\gamma_{r2}}e^{j\psi}}, \quad \tilde{\gamma}_- = 0 \quad (18)$$

in which the wave of amplitude \tilde{A}_- has no radiative loss and becomes the ideal FW-BIC (ideally dark mode), whereas all radiative loss is transferred to the bright mode \tilde{A}_+ . At this point in momentum space ($\mathbf{k}=\mathbf{k}_{\text{BIC}}$), $\hat{\Omega}$ and $\hat{\Gamma}_r$ are both diagonal, and since $\text{rank}(\hat{\Gamma}_r) = 1$ (only a single independent decay port exists) the resonant states interfere to annihilate the coupling with the radiation channel of the BIC mode, which guarantees energy conservation **as any coupling among the final orthogonal modes asymptotically vanishes** [3].

However, arbitrarily close to the BIC point in the momentum, both modes experience nonzero radiative loss. The modes are coupled with a single independent radiation channel, thus are nonorthogonal since their coupling guarantees energy conservation. This behavior holds true in any real system, particularly with **momentum close** to ideal FW-BICs, referred to as **FW quasi-BICs**. It is worth mentioning that in presence of nonnegligible absorption loss, the modes are always nonorthogonal. **If we perturb the ideal FW-BIC condition by moving in momentum space,** in the representation where $\hat{\Omega}$ is diagonal, in general $\hat{\Gamma}_r$ must have nonzero off-diagonal terms to ensure energy conservation, or similarly, in the representation where $\hat{\Gamma}_r$ is diagonal, $\hat{\Omega}$ must have nonzero off-diagonal terms, $\kappa_{12,21}$, which represent the near-field coupling [10]. This is a key concept that

implies that $\forall \mathbf{k}: \mathbf{k} \simeq \mathbf{k}_{BIC}$, the new perturbed Hamiltonian $\hat{H}^r(\mathbf{k} \simeq \mathbf{k}_{BIC})$ for the final coupled modes, the FW *quasi*-BIC $A_-(\mathbf{k} \simeq \mathbf{k}_{BIC})$ and bright $A_+(\mathbf{k} \simeq \mathbf{k}_{BIC})$ modes, can be represented with non-zero off-diagonal terms in $\hat{\Omega}(\mathbf{k} \simeq \mathbf{k}_{BIC})$, when $\hat{\Gamma}_r$ is diagonal because of energy conservation as described below (**Supplementary Fig. 1a**).

The same non-Hermitian Hamiltonian model can also describe the effect of coupled-resonance-induced transparency due to the interference of nonorthogonal eigenvectors, thus at a different wavevector from the ideal FW-BIC condition. Hsu *et al.* demonstrated that when multiple resonances (two or more) are connected to a single independent decay port, a transparency window, known as coupled-resonance-induced transparency, always occurs regardless of the radiation loss values of the resonances because of the off-diagonal terms [11]. **Therefore, this coupling, also necessary for any FW quasi-BIC point, can give rise in special cases to coupled-resonance-induced transparency.** The condition for EIT can, in principle, occurs also with momentum near the ideal FW-BIC point, i.e., when $\mathbf{k}_{EIT} = \mathbf{k}_{BIC} + \delta\mathbf{k}$ (**Supplementary Fig. 1a**). At the EIT point, the slow light condition increases the photon-matter interaction time, enhancing emission properties.

In the next section we first describe the occurrence of the transparency condition in the far-field representation and its link with the near-field representation. We then consider the perturbation of the Hamiltonian close to the FW-BIC to demonstrate explicitly that the FW quasi-BIC, despite being a quasi-dark mode, can reach the maximum physical limit of local field enhancement under the supercritical coupling condition, thanks to the near-field coupling with its bright partner. We will also validate these considerations derived from TCMT through RCWA simulations of our photonic system.

1.3 Analogue of electromagnetically induced transparency and definition of Supercritical Coupling

The calculations presented here follow references [10,11] for clarity of description, **but with harmonic time dependence convention $\exp(j\omega_{in}t)$** . Let's first restate the TCMT problem by writing the dynamical equations for the two modes that are nonorthogonal in the representation where $\hat{\Omega}(\mathbf{k})$ is diagonal and a **single radiation channel**. Since the representation is changed with respect to Eq. (10), we are considering different symbols for the elements in the matrices, and we adopt this representation only because the condition for EIT emergence is rather simple to show:

$$\frac{d}{dt} \begin{pmatrix} A_1 \\ A_2 \end{pmatrix} = \left[j \begin{pmatrix} \bar{\omega}_1 & 0 \\ 0 & \bar{\omega}_2 \end{pmatrix} - \begin{pmatrix} \bar{\gamma}_{r1} & \gamma_{12} \\ \gamma_{12} & \bar{\gamma}_{r2} \end{pmatrix} - \begin{pmatrix} \gamma_a & 0 \\ 0 & \gamma_a \end{pmatrix} \right] \begin{pmatrix} A_1 \\ A_2 \end{pmatrix} + \begin{pmatrix} d_1 \\ d_2 \end{pmatrix} s^+, \quad (19)$$

$$s^- = c_{21}s^+ + d_1A_1 + d_2A_2. \quad (20)$$

In Eq. (19), off-diagonal terms γ_{12} in the radiative decay matrix must be nonzero for energy conservation if both modes decay in the channel, **meaning that the decay matrix and the frequency matrix cannot have diagonal forms simultaneously** [10, 11]). In Eq. (20), s^- is the transmitted wave and we have, due to the presence of the substrate breaking vertical symmetry, that the direct scattering matrix elements are $c_{11} = -c_{22} = (1 - n)/(1 + n)$, with n index of the substrate, and $c_{12} = c_{21} = 2\sqrt{n}/(1 + n)$. Equation (20) simplifies when the system is mirror symmetric since $n = 1$ [11]. Invoking again energy conservation and time reversal symmetry and using the relations between $\hat{\Gamma}_r$, \hat{C} and \hat{D} :

$$d_{1,2} = j\sqrt{2\bar{\gamma}_{r1,r2}/(n+1)}, \quad (21)$$

$$\gamma_{12} = \sqrt{\bar{\gamma}_{r1}\bar{\gamma}_{r2}}. \quad (22)$$

Let us keep using a mirror symmetric system to determine the condition of induced transparency. The

experimental case is then calculated with RCWA, showing that the condition for induced transparency also holds for vertical asymmetry. The complex transmission coefficient at regime is [10]:

$$t = c_{21} \mp \frac{(c_{11} \pm c_{12}) [j(\omega_{in} - \bar{\omega}_2) + \gamma_a] \bar{\gamma}_{r1} + [j(\omega_{in} - \bar{\omega}_1) + \gamma_a] \bar{\gamma}_{r2}}{[j(\omega_{in} - \bar{\omega}_1) + \gamma_a + \bar{\gamma}_{r1}] [j(\omega_{in} - \bar{\omega}_2) + \gamma_a + \bar{\gamma}_{r2}] - \bar{\gamma}_{r1} \bar{\gamma}_{r2}}, \quad (23)$$

in which $|c_{11} + c_{12}| = |c_{22} - c_{12}|$ and we have already established that absorption is the same for both modes and given by γ_a . The top (bottom) signs are used when both modes are even (odd) with respect to vertical symmetry. In the limit $\gamma_a \ll (\bar{\omega}_1 - \bar{\omega}_2)^2 / \max(\bar{\gamma}_{r1}, \bar{\gamma}_{r2})$, the absorptive decay rate is sufficiently small that the transmission coefficient approaches 1 (**EIT condition**) when the numerator of the second term becomes zero at the transparency frequency ω_t , given by:

$$\omega_{in} = \frac{\bar{\omega}_1 \bar{\gamma}_{r2} + \bar{\omega}_2 \bar{\gamma}_{r1}}{\bar{\gamma}_{r1} + \bar{\gamma}_{r2}} \doteq \omega_t. \quad (24)$$

This condition is always fulfilled when $\bar{\omega}_1 < \omega_{in} < \bar{\omega}_2$ provided that the resonances are sufficiently close, regardless of their radiative damping. In a real system for $\gamma_a \neq 0$, the approximation to this condition is a consequence of the optical theorem, for which t cannot reach ideally 1. Nonetheless, the fast dispersion induced at the transparency frequency leads to an enhancement of the local optical field [12-14]. **Indeed, when the EIT is approached, light is significantly slowed down, which favors light-matter interactions and enhances the optical emission process. With this simple demonstration, we have proved that FW-BIC and EIT can be close in principle in the momentum space. Indeed, the induced transparency arises from the coupling of two optical modes to the same radiation channel, which is also the same framework near FW-BIC.**

While the diagonal frequency matrix representation is useful for finding the transparency condition, the next one will provide more insight about the mode coupling. Let us now rewrite the dynamical equations (19) in the representation where the radiative decay is diagonal. We will indicate the final eigenvector waves at $\mathbf{k} = \mathbf{k}_{\text{EIT}}$ with amplitudes A'_+ and A'_- (not to be confused with the amplitudes \tilde{A}_+ , \tilde{A}_- at the FW-BIC wavevector $\mathbf{k} = \mathbf{k}_{\text{BIC}}$ in Eq. (12)). As mentioned earlier, $\text{rank}(\hat{\Gamma}_r) = \text{rank}(\hat{D}) = 1$. Thus, in its diagonal representation, $\hat{\Gamma}_r$ has only one nontrivial element because the determinant must be zero. It is straightforward to demonstrate that in this equivalent representation (with $c_{21} = 1$):

$$\frac{d}{dt} \begin{pmatrix} A'_+ \\ A'_- \end{pmatrix} = \left[j \begin{pmatrix} \omega'_+ & \kappa'_{12} \\ \kappa'_{12} & \omega'_- \end{pmatrix} - \begin{pmatrix} \gamma'_+ & 0 \\ 0 & 0 \end{pmatrix} - \begin{pmatrix} \gamma'_a & \zeta'_{12} \\ \zeta'_{21} & \gamma'_a \end{pmatrix} \right] \begin{pmatrix} A'_+ \\ A'_- \end{pmatrix} + \begin{pmatrix} d'_1 \\ 0 \end{pmatrix} s^+, \quad (25)$$

$$s^- = s^+ + d'_1 A'_+, \quad (26)$$

where the connection with the previous representation of diagonal frequency matrix is given by:

$$\omega'_+ = \frac{\bar{\omega}_1 \bar{\gamma}_{r1} + \bar{\omega}_2 \bar{\gamma}_{r2}}{\bar{\gamma}_{r1} + \bar{\gamma}_{r2}}, \quad (27)$$

$$\omega'_- = \frac{\bar{\omega}_1 \bar{\gamma}_{r2} + \bar{\omega}_2 \bar{\gamma}_{r1}}{\bar{\gamma}_{r1} + \bar{\gamma}_{r2}}, \quad (28)$$

$$\kappa'_{12} = \frac{(\bar{\omega}_2 - \bar{\omega}_1) \sqrt{\bar{\gamma}_{r1} \bar{\gamma}_{r2}}}{\bar{\gamma}_{r1} + \bar{\gamma}_{r2}} \quad (29)$$

$$\gamma'_+ = \bar{\gamma}_{r1} + \bar{\gamma}_{r2} \quad (30)$$

$$\gamma'_- = 0, \quad (31)$$

$$d'_1 = \sqrt{d_1^2 + d_2^2}. \quad (32)$$

The above relations are useful because they directly state that the transparency frequency $\omega_t = \omega'_-$,

corresponds to the final dark mode. This link is important: at the transparency frequency the fast dispersion slows down light and enhances the local field, which corresponds to the dark mode. While the previous representation involved nonorthogonal modes in which their coupling was expressed in the far-field, in this second representation, we can see that a **nonradiating dark mode** with $\gamma'_- = 0$ is **coupled via a nonzero near-field constant** κ'_{12} to a bright leaky wave with a decay rate $\gamma'_+ = \bar{\gamma}_{r1} + \bar{\gamma}_{r2}$. These identities must not be confused with Eqs. (17-18) that express the relations between diagonal dark and bright modes at the FW-BIC point $\mathbf{k} = \mathbf{k}_{BIC}$ with the original uncoupled modes. Instead, the above equations refer to two different representations of the same modes at fixed and same wavevector $\mathbf{k} = \mathbf{k}_{EIT} \neq \mathbf{k}_{BIC}$. Here, when the drive field is turned off, the dark mode amplitude decays to zero. In the linear regime, exchange energy occurs between the modes. We see below that while the drive field is on, energy flows from the bright mode to the dark mode. As the drive field is turned off, energy flows from the dark mode to the bright mode. Consequently, the dark mode undergoes decay in the far field due to its nearly-zero direct coupling with the radiation channel and its nonzero near-field coupling with the bright mode [15]. In this alternative representation, it is the near-field coupling between a dark mode and the bright mode that gives rise to the transparency condition. This formulation aligns with the general framework used in the subradiant-superradiant model, which illustrates the analogue of electromagnetically induced transparency in photonic and plasmonic systems [12-14].

Supplementary Fig. 1 | Supercritical Coupling Model. **a**, (top) Oscillators scheme at the FW quasi-BIC, with depicted dispersion diagram of the FW-BIC formation *near* the avoided crossing point and with mismatched momentum with respect to the EIT occurring at the avoided crossing (dashed lines are frequencies and solid lines linewidth size). (Bottom) The corresponding Hamiltonians (frequency and decay rate) at the ideal FW-BIC (\mathbf{k}_{BIC}), FW quasi-BIC ($\simeq \mathbf{k}_{BIC}$), and EIT (\mathbf{k}_{EIT}) points. **b-d**, Normalized intensity enhancements G/G_{max} for both dark and bright modes (red and blue solid lines, respectively) compared to the corresponding single resonance intensity enhancements (red and blue dashed lines) as a function of Q_K , for $Q_{R1} = 5 \times 10^9$, $Q_{R2} = 200$, $Q_a = 5000$, with input frequency tuning with the dark (**b**), middle (**c**), and bright frequency (**d**). **e**, Intensity level ratio between the coupled dark mode G_{dark} at supercritical coupling and the single dark mode $G_{single-dark}$ as a function of Q_{R1}/Q_a and Q_{R2}/Q_a : when $Q_{R1}/Q_a = 1$, we find the critical coupling condition, and the coupled dark mode has the same level of enhancement as the single dark mode. When $Q_{R2} \gg Q_a$, there is no advantage ($G_{dark} \rightarrow G_{single-dark}$) because the input channel is unfavorable. In the remaining region of the parameters, $G_{dark} \gg G_{single-dark}$.

Supercritical Coupling. The FW-BIC and classical analogue of EIT formalisms are derived from the same original framework of modes coupled to a single radiation channel: the EIT with non-zero off diagonal terms, whereas the ideal FW-BIC is a limit of this framework with zero off-diagonal terms. Since the EIT occurs at the avoided crossing, FW-BIC must not be at the avoided crossing, which implies that the radiative decay rates of the closed channel modes in Eq. (15) differ, $\gamma_{r1} \neq \gamma_{r2}$. Thus, the ideal FW-BIC is not at the avoided crossing ($\omega_1 = \omega_2$) but is shifted in its vicinity. Both conditions can be fulfilled, in principle, for close wavevectors when, e.g., $\gamma_{r1} \simeq 5 \gamma_{r2}$ (see the simulated linewidths when the modes do not cross each other in Supplementary Fig. 4, orientation angle of the photonic crystal $\phi=45^\circ$). This also means that **the realization of EIT is possible when the involved dark mode is a perturbation of the FW-BIC mode, i.e. exhibits characteristics of a FW quasi-BIC**. While this will be shown using RCWA in our system, let us now explore the consequences for enhancing the local optical field.

As shown in the scheme of **Supplementary Fig. 1a**, let us write explicitly the dynamical equations (12) adding the perturbation to the diagonal representation (FW-BIC point) of the Hamiltonian as we move away from the ideal BIC wavevector towards the EIT point. Since the radiative Q -factor of a BIC scales as $|\mathbf{k} - \mathbf{k}_{BIC}|^{-\alpha}$ with $\alpha \geq 2$, for any wavevector close to the BIC point, $\mathbf{k} = \mathbf{k}_{BIC} + \Delta\mathbf{q} \simeq \mathbf{k}_{BIC}$, it is necessary to admit a finite non-zero decay rate of the dark mode A_- , i.e. $1/\gamma_- = \tau_{R1}$ with $\gamma_- \rightarrow \varepsilon \gtrsim 0$, and as such it is necessary to include a non-zero mode coupling $\kappa_{12} \neq 0$ to guarantee energy conservation as both modes are coupled to a single independent radiation channel. The perturbed Hamiltonian is $\hat{H}^r(\mathbf{k} \simeq \mathbf{k}_{BIC}) = \begin{pmatrix} \omega_+ & \kappa_{12} \\ \kappa_{12} & \omega_- \end{pmatrix} + j \begin{pmatrix} \gamma_+ & 0 \\ 0 & \gamma_- \end{pmatrix}$. It is important to note that the modes are the final coupled modes of the system and not the closed channel modes: their frequencies are considered as shifted with respect to the exact frequencies of bright and dark modes of the FW point $\mathbf{k} = \mathbf{k}_{BIC}$ in Eq. (13), and the finite decay rate of the dark mode turns it into a **quasi-dark mode** (FW **quasi** BIC), and this nonzero coupling to the radiation channel ($\sqrt{2/\gamma_-} = \sqrt{2/\tau_{R1}}$) will imply nonzero near- (κ_{12}) or far-field (γ_{12}) coupling with the shifted bright partner, depending on the representation used. The bright mode has amplitude A_+ , with decay rate $1/\gamma_+ = \tau_{R2} \ll \tau_{R1}$. Generally, the off-diagonal terms can be kept complex to include both near and far-field coupling, but we have verified by RCWA that the coupling is real with good approximation in the next section. Here, we assume the representation with near-field coupling κ_{12} in the off-diagonal terms. All the quantities listed below are intended to be related to the final modes. Now, considering the general dynamical equations with both modes having same losses included all in $\gamma_a = 1/\tau_a$, it is possible to write, $\forall \mathbf{k}: \mathbf{k} \simeq \mathbf{k}_{BIC}$, that:

$$\frac{dA_-}{dt} = j\omega_- A_- - \left(\frac{1}{\tau_a} + \frac{1}{\tau_{R1}}\right) A_- + j\kappa_{12} A_+ + \sqrt{\frac{2}{\tau_{R1}}} S_+, \quad (33)$$

$$\frac{dA_+}{dt} = j\omega_+ A_+ - \left(\frac{1}{\tau_a} + \frac{1}{\tau_{R2}}\right) A_+ + j\kappa_{12} A_- + \sqrt{\frac{2}{\tau_{R2}}} S_+. \quad (34)$$

This set of equations is valid for any system with two modes and one-radiation channel, regardless of the nature of the optical resonances (e.g., plasmonic modes, whispering gallery modes, guided modes, defect modes) and without any approximation on the range of the parameters. Considering $\frac{d}{dt} \rightarrow j\omega_{in}$, and solving for A_- in (33), substituting it in (34), and then substituting the resulting A_+ again in (33), we find, at the steady state, that:

$$\begin{aligned} \frac{A_-(\omega_{in})}{s_+} &= \frac{\sqrt{\frac{2}{\tau_{R1}}}}{j(\omega_{in}-\omega_-)+\frac{1}{\tau_a}+\frac{1}{\tau_{R1}}} + \\ &+ \frac{j/\tau_\kappa \sqrt{\frac{2}{\tau_{R2}}}}{\left[j(\omega_{in}-\omega_-)+\frac{1}{\tau_a}+\frac{1}{\tau_{R1}}\right] \left[j(\omega_{in}-\omega_+)+\frac{1}{\tau_a}+\frac{1}{\tau_{R2}}+\frac{1/\tau_\kappa^2}{j(\omega_{in}-\omega_-)+1/\tau_a+1/\tau_{R1}}\right]} + \\ &- \frac{1/\tau_\kappa^2 \sqrt{\frac{2}{\tau_{R1}}}}{\left[j(\omega_{in}-\omega_-)+\frac{1}{\tau_a}+\frac{1}{\tau_{R1}}\right]^2 \left[j(\omega_{in}-\omega_+)+\frac{1}{\tau_a}+\frac{1}{\tau_{R2}}+\frac{1/\tau_\kappa^2}{j(\omega_{in}-\omega_-)+1/\tau_a+1/\tau_{R1}}\right]}}, \quad (35) \end{aligned}$$

$$\begin{aligned} \frac{A_+(\omega_{in})}{s_+} &= \frac{\sqrt{\frac{2}{\tau_{R2}}}}{j(\omega_{in}-\omega_+)+\frac{1}{\tau_a}+\frac{1}{\tau_{R2}}+\frac{1/\tau_\kappa^2}{j(\omega_{in}-\omega_-)+1/\tau_a+1/\tau_{R1}}} + \\ &+ \frac{j/\tau_\kappa \sqrt{\frac{2}{\tau_{R1}}}}{\left[j(\omega_{in}-\omega_-)+\frac{1}{\tau_a}+\frac{1}{\tau_{R1}}\right] \left[j(\omega_{in}-\omega_+)+\frac{1}{\tau_a}+\frac{1}{\tau_{R2}}+\frac{1/\tau_\kappa^2}{j(\omega_{in}-\omega_-)+1/\tau_a+1/\tau_{R1}}\right]}. \quad (36) \end{aligned}$$

Above, we have explicitly defined the near-field coupling lifetime $\tau_\kappa = \frac{1}{\kappa_{12}}$, and the associated quality factor $\tau_\kappa = 2Q_\kappa/\omega$. We can see that the quasi-dark mode A_- can be excited *via* internal coupling κ_{12} more than what expected from the isolated resonance response of the dark mode, represented by the first term in Eq. (35). In **Supplementary Figs. 1b-d**, the behavior for both mode intensities for a specific set of informative values, $Q_{R1} = 5 \times 10^9$, $Q_{R2} = 200$, $Q_a = 5000$ is plotted to capture the major insight. In **Supplementary Fig. 1b**, the intensity field enhancement

$$G = \frac{|A_\pm|^2}{|s_+/\sqrt{\omega_{in}}|^2 V_{eff}} \quad (37)$$

is plotted for both modes [solid red line for the dark A_- and blue line for bright A_+ using full expressions (35,36)] and shows that the dark mode on resonance ($\omega_{in} = \omega_-$) reaches the maximum limit of field enhancement possible in a real-world resonator with nonradiative loss, $G_{max} = Q_a/V_{eff}$, even if

$$Q_{R1} \gg Q_a, \quad (38)$$

which would be impossible in case of a single dark resonance, *i.e.*, not coupled to another wave (dashed red line). This condition occurs at the **supercritical coupling point** defined by:

$$\bar{\tau}_\kappa = \sqrt{\tau_{R2}\tau_a}, \quad (39)$$

or

$$\bar{Q}_\kappa = \sqrt{Q_{R2}Q_a}. \quad (40)$$

Indeed, assuming $\tau_{R1} \gg \tau_a, \tau_{R2}, \tau_\kappa$ and $\tau_{R2} < \tau_a$ in Eq. (35), and considering $\omega_{in} = \omega_-$ (on resonance with the dark mode) and $|\omega_{in} - \omega_+| \simeq 2\kappa_{12} = 2/\tau_\kappa$ (the coupling affects the split in frequencies, thus the pump is shifted from the bright mode when on resonance with the dark one), the relation simplifies

in:

$$\frac{A_-(\omega_{in}=\omega_-)}{s_+} \rightarrow \left[\frac{j/\tau_\kappa \sqrt{\frac{2}{\tau_{R2}}}}{j/(\tau_\kappa \tau_a) + 1/\tau_a^2 + 1/(\tau_{R2} \tau_a) + 1/\tau_\kappa^2} \right]_{\tau_\kappa = \sqrt{\tau_{R2} \tau_a}} \rightarrow j\sqrt{\tau_a/2}, \quad (41)$$

in which the first two terms in the denominator have been neglected since smaller when $\tau_{R2} < \tau_a$. The above relation proves that the dark mode intensity enhancement $G = \left| \frac{A_-(\omega_{in}=\omega_-)}{s_+/\sqrt{\omega_{in}}} \right|^2 \frac{1}{V_{eff}} = Q_a/V_{eff} = G_{max}$, *i.e.* it can reach the maximum imposed by nonradiative losses even in extreme situations of mismatched quality factors. It is worth mentioning that when $Q_\kappa \rightarrow \infty$ ($\kappa_{12} \rightarrow 0$), we again obtain the correct case of uncoupled resonances, and the dark mode field goes to the level it could gain if it were isolated (dashed red line). Indeed, in the plot, we have imposed that the near-field coupling rate affects the spectral separation among resonances as it is proportional to their distance: $\omega_\pm = \omega_o \pm \kappa_{12} = \omega_o [1 \pm 1/(2Q_\kappa)]$. Thus, for $Q_\kappa \rightarrow \infty$, the resonant frequencies coincide and cross. Even when out of perfect spectral tuning, the maximum gain achieved by the quasi-dark mode A_- is orders of magnitudes larger than what possible in a single dark mode, as shown in **Supplementary Fig. 1c-d**. In case $\omega_{in} = \omega_o = 1/2(\omega_+ + \omega_-)$, the optimum shifts to larger $Q_\kappa^* \simeq Q_a$. When $Q_{R2} \rightarrow Q_a$ and $\omega_{in} = \omega_-$ the bright mode is critically coupled to the pump but there is still energy going into the dark mode up to $0.3G_{max}$ at a certain $Q_\kappa^* \lesssim \bar{Q}_\kappa = Q_a$. Furthermore, by inspecting the ratio between the solid red line and the dashed red line, it is possible to appreciate how, even if Q_κ does not reach the optimum, the intensity of the coupled dark resonance is orders of magnitude larger than that of the single resonance.

The supercritical coupling mechanism guarantees the possibility of achieving the maximum level of local field enhancement when the coupling (Q_κ) is optimally tuned, and always in the highest Q -factor mode, even under the conditions of coupling, for both bright and dark modes, which would be unfavorable in the case of single isolated modes. To give an example, let be $Q_{R2} = 10^3 \ll Q_a = 10^6 \ll Q_{R1} = 10^{10}$, thus none of the modes matches Q_a , on the contrary, they have completely unmatched quality factors. If $Q_\kappa = \sqrt{10^3 \times 10^6} \simeq 3 \times 10^4$ (say $V_{eff} = 1$ for brevity), the dark mode reaches the maximum intensity enhancement $G_{max} = 10^6$ despite the single dark resonance level of intensity enhancement would be only 10^2 , *i.e.* four orders of magnitude less, as it is shown in the main **Fig. 1e**. In addition, the supercritical coupling condition is independent from the highest- Q factor resonance, unlike the critical coupling condition ($Q_{R1} = Q_a$) [16], converges to the critical coupling result if $Q_{R1} \rightarrow Q_a$, and can ensure a larger level of enhancement in the dark mode, with a considerable advantage over the single dark resonance case, even with Q_{R2} and Q_κ vary over a considerably large range of values. This is shown for fixed $\bar{Q}_\kappa = \sqrt{Q_{R2} Q_a}$ in **Supplementary Fig. 1e**.

This mechanism holds true for all wavevectors that span the range from a FW quasi-BIC to the EIT point, with correspondingly varied values of the parameters involved (coupled mode frequencies, decay rates, and near-field coupling). Far from this momentum region, the mode coupling becomes progressively negligible (as it can be easily calculated numerically) and the isolated single mode response is restored.

Turning to the parallel with coupled-resonance induced transparency, we understand that at the dark mode frequency $\omega_t = \omega'_-$ (Eqs. 24, 28) where the transparency window occurs, fast dispersion leads to slow light and an enhanced field that with suitable coupling between modes, could reach the maximum field enhancement of the system, as indicated by supercritical coupling. **We recall that EIT is not a necessary condition of the FW mechanism**, although their proximity in

momentum space, set on purpose, widens the wavevector span over which the field is enhanced.”

In order to clarify this last point to the reviewer, we can evaluate the group delay for $\omega \simeq \omega'_-$, given by $\tau_{GD} = -d/d\omega[\arg(A'_-(\omega))]$, where we can approximate the spectral amplitude profile of the transparency window as

$$A'_-(\omega) \simeq A'_0 \exp\left[-j\frac{(\omega-\omega'_-)}{\gamma'_-}\right],$$

which gives $\tau_{GD} = 1/\gamma'_-$, and that is only limited by nonradiative loss since $\gamma'_- \rightarrow \gamma_a$ in a real world resonator with diverging radiative quality factor. This finally determines the maximum achievable field intensity enhancement due to slow light, since the group delay measures the cavity lifetime besides a factor 2, $\tau_{GD} = 2\tau_{R1}$ [Hosseini, S. E., Karimi, A., & Jahanbakht, S. (2018), “Q-factor of optical delay-line based cavities and oscillators, Optics Communications, 407, 349-354.”] A different equivalent approach is also shown in Appendix D of paper [Shi, W., Gu, J., Zhang, X., Xu, Q., Han, J., Yang, Q., ... & Zhang, W. (2022), Terahertz bound states in the continuum with incident angle robustness induced by a dual period metagrating, Photonics Research, 10(3), 810-819], where the group delay is equally found.

The intensity enhancement is given by the number of photons accumulated per optical cycle in the resonator due to the delay normalized to the number of incident photons. Recalling that the quality factor is the ratio between stored energy and lost energy per optical cycle, we have that the maximum $G = (\omega\tau_{GD})/2V_{eff} \rightarrow Q_a/V_{eff}$, as associated with the transparency window of nonradiative loss-limited width $\gamma'_- \rightarrow \gamma_a$. This means that the dark mode could gain maximum enhancement, but this can occur only by the interference with the bright partner.

A summary of the coupled-mode model of the perturbed Hamiltonian detailed in SM is also described in the main paper from **line 74**. It is noteworthy to mention that the main text has been shortened to 70% of the previous version at the request of the editor. Only the significant variations pertaining the reviewer’s concerns have been highlighted in red.

Main Manuscript line 74: “The system is described by the non-Hermitian Hamiltonian $\hat{H}_k = \hat{\Omega}(k) + j\hat{\Gamma}_r(k)$, which models transverse electric-like (TE-like) and magnetic-like (TM-like) modes coupled to a single independent radiation channel, thus originally nonorthogonal^{24,32}. In the energy-momentum space, these modes evolve and eventually approach a FW-BIC at a specific wavevector $k = k_{BIC}$ if the FW condition that diagonalizes $\hat{H}_{k_{BIC}}$ is satisfied by the parameters. The initial modes 1 and 2 have radiative loss rates γ_{r1} and γ_{r2} . The coupled final modes of wavelengths λ_- and λ_+ split apart because of strong coupling (**Fig. 1b, c**). One of these waves becomes a perfect dark mode (ideal FW-BIC) with zero linewidth $\gamma_- = 0$ and diverging lifetime $\tau_{R1} = 1/\gamma_- = 2Q_{R1}/\omega_1 = \infty$, **at some wavevector k_{BIC} near the avoided crossing**. The bright mode acquires all radiative losses with $\gamma_+ = \gamma_{r1} + \gamma_{r2}$, which provides a final mode with low Q_{R2} . At the asymptotic FW condition, $\hat{\Omega}(k_{BIC})$ and $\hat{\Gamma}_r(k_{BIC})$ are simultaneously diagonalized, resulting in orthogonal modes. This is allowed by energy conservation of input field and modes because the dark mode totally decouples from the radiation channel. However, **for wavevectors close to but not at the FW-BIC, the perturbed FW quasi-BIC ($\gamma_- \neq 0$) takes on nonzero coupling with the radiation channel ($\sqrt{2\gamma_-}$), thus the perturbed Hamiltonian $\hat{H}_{k \simeq k_{BIC}}$ must be represented with nonzero off-diagonal terms κ_{12} in $\hat{\Omega}(k \simeq k_{BIC})$ to obey energy conservation^{27,32}. When two modes are coupled with a single radiation channel, also coupled-resonance-induced transparency can take place. This is the analogue of electromagnetically induced transparency (EIT) in photonic/plasmonic systems and can provide exceptionally slow light and enhanced local optical field³³. For suitable PCNS geometries, EIT and ideal FW-BIC may occur with small phase mismatch ($k_{EIT} = k_{BIC} + \delta k$) (**Fig. 1c**). Essentially, the quasi-BIC may evolve to the transparency frequency of the EIT process. Temporal-coupled mode theory (TCMT) is extensively discussed in Supplementary Sec. 1.2-1.3, where the final mode amplitudes for the dark**

(FW quasi-BIC) and bright partners are explicitly provided in Supplementary Eqs. (35,36). The intensity enhancement of final modes G , normalized to the maximum intensity enhancement $G_{max} = Q_a/V_{eff}$, is plotted in **Fig. 1e** in an extreme case of unmatched quality factors, unsatisfactory scenario for single (uncoupled) resonances. Having defined the near-field coupling quality factor as $Q_\kappa = \omega/(2\kappa_{12}) = \tau_\kappa\omega/2$, an optimum condition that we term supercritical coupling can be reached when $\bar{Q}_\kappa = \sqrt{Q_{R2}Q_a}$, which avoids the bottleneck of narrow input radiation channel ($Q_{R1} = 10^{10}$) by diverting the low- Q_{R2} mode input energy to the dark mode. The maximum level of local field enhancement, always in the highest Q -factor mode, can be reached even in highly unfavored conditions for single isolated modes. **Supplementary Fig. 1** shows off-resonance behavior and parameters for which the coupled dark mode is advantaged. At the EIT transparency frequency (dark mode), the field is enhanced because of slow light, possibly reaching maximal level as explained by supercritical coupling of dark mode.”

Thus, **Figure 1** has been modified accordingly, as reported here below with the improved panels **1c** and **1e**, explaining that EIT and FW-BIC are not the same wavevector and showing the supercritical coupling for a particularly representative condition with mismatched quality factors where the quasi-BIC reaches the maximum possible gain.

Fig. 1 | Principle of supercritical coupling and directive upconversion emission. a, Layout of the PCNS unit cell geometry and collimated upconversion radiation generated through supercritical edge BIC coupling. **b**, Schematic depicting internally coupled interfering resonances and their external coupling to the far-field. **c**, Scenario where the coupled waves λ₋ and λ₊ (dashed lines) and the linewidths (solid lines) shift apart due to intercoupling κ₁₂: λ₋ becomes a FW quasi-BIC close in momentum to the EIT point, with laser wavelength λ_{in} set to λ₋. **d**, The energy-momentum dispersion of the system is tailored to achieve FW quasi-BIC and EIT with minimal phase mismatch δk at the laser wavelength λ_{in}. **e**, Normalized intensity enhancement for bright and dark modes with highly mismatched parameters, Q_{R2} = 10³ ≪ Q_a = 10⁶ ≪ Q_{R1} = 10¹⁰. The tuning of Q_κ serves the key

factor enabling the supercritical coupling condition. In this condition, a significant portion of the far-field energy is diverted to the near-field drive, effectively exciting the high- Q_{R1} FW quasi-BIC (denoted by the red solid line) up to the maximum achievable level, G_{\max} . This level surpasses the threshold of the single dark mode (denoted by the red dashed line) by several orders of magnitude. When $Q_k \rightarrow \infty$ ($\kappa_{12} \rightarrow 0$), the dark mode field decreases to the single resonance threshold.

Comment #3 Given that (i) the developed supercritical coupling regime cannot be analytically reproduced from the given model, (ii) the developed model is based on certain unphysical assumptions, I cannot recommend the manuscript for publication in Nature in any form. My overall impression that the supercritical coupling regime is an invalid concept in general, as it contradicts the general physical limitation of linear systems with harmonic excitation to be excited more efficiently than in the standard critical coupling regime (see Ref.[23] of the main text). Despite that, the experimental results of the manuscript are of very high technical quality, the demonstration of strong resonant directive upconversion emission from the sample edge is of high practical importance and the presented extended numerical data analysis supports the data. Thus, I would possibly recommend the manuscript for a more technical journal, like Nature Communications, in case the authors do a major revision by completely removing the supercritical coupling story from the text and focusing on the experimental results fitting them within a more standard theory.

As an alternative option, if the authors manage to carefully explain their results with a revised accurate theory based on correct assumptions, and indeed uncover that the UCNP emission enhancement is driven by a new physical concept, this paper might be still a candidate for publication in Nature. For this, they would need to show that their concept works in at least two representations of the Hamiltonian, as they all are equivalent.

Once again, we would like to express our gratitude to the reviewer for thorough evaluation, invaluable suggestions, constructive feedback, and critical insights, which have largely contributed to the enhancement of the technical presentation in this revised manuscript. In our responses to **Comments #1 and #2**, we have described how we improved our model by incorporating exact analytical solutions and establishing proper validity limits, as well as determining the maximum field enhancement. **Our work highlights the validity of the concept of supercritical coupling**, demonstrating that the maximum field enhancement can be reached, even within the dark mode. This surpasses by orders of magnitude the attainable value without coupled-mode theory and extends beyond the critical coupling, which can be regarded as a special case within our extended coupled-mode model. Moreover, our model elucidates the mechanism behind the enhanced dark mode field at the transparency frequency of EIT, in accordance with slow-light condition. The proximity of EIT and FW-BIC results in a finite range of wavevectors where the dark mode can reach its maximum field, thereby enhancing the applicability of high Q resonators across a variety of wavevectors, extending beyond a single wavevector.

Leveraging this mechanism of supercritical coupling for dark-mode pumping holds significant promise for a broad range of applications, especially in scenarios where in-plane symmetry breaking for quasi BICs is not possible, and or where knowledge of the nonradiative loss to match radiative coupling is unavailable. Importantly, our model can be applied to various other physical systems **featuring dark modes, thus serving as inspiration for new experiments and technologies.**

To facilitate the reviewer in verifying our results, we have included a Matlab code below, which can be executed in MathWorks Matlab to reproduce our findings. This code can be directly pasted into the MATLAB command view or a new script tab. Additionally, it can be utilized in the freely available online version of Matlab, accessible at <https://it.mathworks.com/products/matlab-online.html>, following the creation of an account and selection of the free version.

```

%% Matlab code
clear all

% desired quality factors and behavior to be checked, that can be changed before running the code
QR1 = 1*10^(10);
Qa= 1.*10^(6);
QR2= 1*10^(3);
Qk = linspace(1,10^8,100000); % near-field coupling quality factor
MAX=ones(1,100000); %auxiliary

% hereafter the units are such that the central frequency w0, between dark mode of frequency wd and bright
%mode of frequency wb, is w0 = 1;

w=-1./(2.*Qk); %(TUNING the frequency w ON THE DARK MODE wd)
wd=-1./(2.*Qk);
wb=1./(2.*Qk);

%in units w0 = 1 the time constants are:
Tk = 2.*Qk;
Ta=2*Qa;
TR1= 2*QR1;
TR2=2*QR2;

%to plot the vertical line at the maximum Qk=sqrt(Qa*QR2)
x = [sqrt(Qa*QR2)].*ones(1,100);
y = linspace(10^-8,1,100);

%dark mode normalized intensity
Int1 =(1/Qa).*abs( (sqrt(2/TR1)./(1i.*(w-wd)+ 1/Ta+1/TR1)) ...
+ 1i.*( ((1./Tk).*(sqrt(2/TR2))) ./ ((1i.*(w-wd)+1/Ta +1/TR1).*(1i.*(w-wb)+1/Ta +1/TR2
+(1./Tk.^2)./(1i.*(w-wd)+1/Ta+1/TR1)))) ...
- ( ((1./Tk).^2.*(sqrt(2/TR1))) ./ ((1i.*(w-wd)+1/Ta +1/TR1).^2.*(1i.*(w-wb)+1/Ta +1/TR2
+(1./Tk.^2)./(1i.*(w-wd)+1/Ta+1/TR1)))) ).^2;

%bright mode normalized intensity
Int2 =(1/Qa).*abs( (sqrt(2/TR2)./(1i.*(w-wb)+1/Ta+1/TR2 +(1./Tk.^2)./(1i.*(w-wd)+1/Ta + 1/TR1))) ...
+ 1i.*( ((1./Tk).*(sqrt(2/TR1))) ./ ((1i.*(w-wd)+1/Ta +1/TR1).*(1i.*(w-wb)+1/Ta +1/TR2
+(1./Tk.^2)./(1i.*(w-wd)+1/Ta+1/TR1)))) ).^2;

% single dark and bright (uncoupled) mode normalized intensity
Iso1 = (1/Qa).*abs( (sqrt(2/TR1)./(1i.*(w-wd)+ 1/Ta+1/TR1))).^2;
Iso2 = (1/Qa).*abs( (sqrt(2/TR2)./(1i.*(w-wb)+ 1/Ta+1/TR2))).^2;

% Plot
figure,
loglog(Qk,Int1,'color',[1 0 0],LineWidth=4)
hold on
loglog(Qk,Int2,'color','#48C5EA',LineWidth=4)
hold on
loglog(Qk,Int1+Int2,'color',[0 0 0],LineWidth=1)
hold on
loglog(Qk,Iso1,'color',[1 0 0],'LineWidth',2,'LineStyle','--')
hold on
loglog(Qk,Iso2,'color','#48C5EA','LineWidth',2,'LineStyle','--')
hold on
loglog(Qk,MAX,'color',[0.7 0.7 0.7],'LineWidth',1,'LineStyle','-.')
hold on
loglog(x,y,'Color','k','LineWidth',0.5)
hold off

xlabel('$ Q \kappa$', 'Interpreter', 'latex')
ylabel('$G/G_{\rm max}$', 'Interpreter', 'latex')
range=[min(Qk) max(Qk) 10^(-8) 2];
axis(range)
pbaspect([1 1.5 1])
set(gca, 'Layer', 'top');
set(gca, 'fontsize',16, 'FontWeight', 'normal', 'LineWidth',1.2);

```

```

grid 'off'
box 'on'
ax=gca;
title(['$Q_{\rm R1}=$',num2str(QR1,'%g') ',    $Q_{\rm R2}=$',num2str(QR2,'%g') ',    $Q_{\rm
a}=$',num2str(Qa,'%g') ],'Interpreter','latex','FontSize',20)
ax.FontSmoothing = 'on'
ax.TickLength = [0.03 0.03]
set(gcf,'Position',[20 1 800 800])
legend("Dark","Bright","Sum","Single Dark Resonator","Single Bright Resonator","$G_{\rm max}$ level",
'Interpreter','latex')
legend('boxoff')
legend('FontSize',18)
legend('FontName','Arial')
legend('Location','northoutside')

set(gca,'TickDir','out');
xtickangle(45)

```

Furthermore, we have enclosed an additional file (Wolfram Mathematica Notebook “.nb”) named “Mathematica_Notebook_for_Reviewer.nb”, which can be executed in Mathematica to reproduce our findings and evaluate the solutions. We have also enclosed a printed version of the Notebook, named “Printed_Version_Mathematica_Notebook_for_Reviewer.pdf”. In this, we reported the functions describing the mode amplitudes (Supplementary Eqs. 35,36), showed that they exchange symmetrically when exchanging the meaning of the parameters (dark to bright and viceversa) as it should be, and finally that their difference is zero within numerical approximation when both have same parameters as it should be.

Comment #4: Below, I list other more minor technical comments. I also attach an additional file providing a detailed analysis of different Hamiltonian representations, explaining, why the physical meaning of the model parameters in the authors' model is invalid and showing that the supercritical coupling regime does not follow from the TCMT equations.

We thank again the reviewer for the additional file provided. We have thoroughly addressed all comments regarding calculations as outlined in *Comments #1-#3*.

Other comments:

Comment #4.1 The authors discuss the importance of vertical asymmetry for realization of FW-BIC and strong coupling between TE and TM modes, citing Refs. [R. Mermet-Lyauoz, et al., arXiv:1905.03868 (2019); Phys Rev B 61, 2090–2101 (2000)]. I disagree with authors and note that a true FW-BIC with zero radiative losses requires complete vertical symmetry as was established in pioneering papers [Hsu, C.W. et al. Nature, 499(7457), pp.188-191 (2013); Zhen, B. et al. Physical review letters, 113(25), p.257401 (2014)]. In Ref. [Phys Rev B 61, 2090–2101 (2000)] the authors did not show that the losses of the dark FW mode reach complete zero at the avoided resonance crossing, as they did numerical simulations only. Ref. [R. Mermet-Lyauoz, et al., arXiv:1905.03868 (2019)] raised an extended discussion in the community back to 2019 and was concluded to be misleading, thus it cannot be a reliable source of information (please note that it was not published in a peer-reviewed journal since 2019). I also agree that from the point of view of experimental applications, a true FW-BIC with zero radiation losses and a quasi-FW-BIC with near-zero radiation losses are indistinguishable. Still, the authors emphasise the role of asymmetry here "the vertical asymmetry, which promotes the interference of vector TE-like and TM-like modes and the formation of FW quasi-BIC, depends on the supporting substrate and cladding layer" which is a drawback rather than a benefit for FW-BIC Q factor value. I would suggest the authors to revise this statement. On the other hand, I agree that the strength of TE-TM coupling can be increased with the increase of the vertical asymmetry.

We understand the reviewer's argument and have removed the sentence accordingly.

Comment #4.2 In response to my comment, the authors added a comparison to symmetry-protected BICs (sp-BICs) to Supp.Mat. I disagree with the comparison, as they state that sp-BICs are not formed due to interference compared to FW-BICs. Actually, sp-BICs can be also understood as a complete destructive interference between two non-orthogonal coupled modes and the same Eq.(15) from SM can be applied to them. In this case, however, $\gamma_1 = \gamma_2$ due to symmetry considerations, thus Eq.(15) is satisfied for $\omega_1 = \omega_2$. Therefore, I would suggest the authors to clarify more why the observed effect of enhanced directive UCNP emission cannot be realized for sp-quasi-BICs for nearly-normal incidence.

We understand the reviewer's argument about symmetry-protected BICs. We acknowledge the distinction in the mechanism compared to our discussion, as sp-BICs often involves an interference process between external input s_+ and output waves s_- that cancel each other out, resulting in zero radiative coupling. However, this mechanism described by the reviewer differs from what we have examined. In our work, we emphasize the necessity of energy transfer occurring within a couple of bright and dark modes, which is necessary for achieving supercritical coupling. In the case of sp-BICs, there is typically only one isolated dark mode within the physical system and its associated transmittance spectrum. The FW-BIC can be moved to a symmetry-protected point, for instance, at normal incidence. This represents an interesting avenue for further investigation and improvement.

Comment #4.3 The authors provide a detailed explanation of the relation of the observed effects and EIT regime. In general, I agree with their explanations, however, I would note that the EIT is a more general effect that can be observed without a BIC for any two coupled modes with close frequencies and lifetimes. The authors' description makes an impression that the EIT is always driven by the BIC which is not true.

We agree with the reviewer, and we explicitly mentioned in the revised version that EIT is not a necessary condition in the main paper (**line 211**):

“Discussion. The mechanism of near-field coupling between FW quasi-BIC and bright mode can divert the input source from the bright to the dark mode, breaking the limits of single-dark mode coupling with orders-of-magnitude improvement, a condition referred to as supercritical coupling. This phenomenon is also related to the EIT process, which can arise from similar coupling and aligns with it, since slow light at the dark mode is associated with superior field enhancement. However, the occurrence of a transparency window is not a requirement.”

Also in SM at the end of Sec. 1.3:

“We recall that EIT is not a necessary condition of FW mechanism, although their proximity in momentum space, set on purpose, widens the wavevector span over which the field is enhanced.”

Comment #4.4 The authors say that the excitation Q factor is usually very large even for large numerical aperture of the excitation objective, as follows from their own observations in previous works. I would not agree that it is a general case. In general, the excitation Q factor can be quite low for quasi-BICs as due to the presence of beam components with high k-vector the effective excited mode is an average in k-space in the vicinity of the quasi-BIC k-vector. If the quasi-BIC Q factor decreases quickly in its vicinity, the averaged value would be sufficiently smaller, reaching ~ 100 for objectives with $NA > 0.1$.

We would like to emphasize that the excitation size has been thoroughly studied in our previous research on BICs, and we have found that the influence of the excitation size is typically negligible

compared to other fundamental factors, even when using a large numerical aperture. We have demonstrated this in multiple studies, including examples in [*J. Phys. Chem. C* **122**, 19738–19745 (2018); *ACS Nano* **14**, 15417–15427 (2020)].

Regarding the influence of sample size on the enhancement of the local optical field, which is the main focus of this work, a formal analysis of sample size influence was performed in a study by Kodigala et al. [*Lasing action from photonic bound states in continuum*, *Nature* **541**, 196–199 (2017)] on lasing action from photonic BICs.”

In the SM text, we did not mention details about the excitation size affected by the input numerical aperture, while we discussed the role of the bandwidth in Q-excitation. The argument of the excitation spot size was omitted from SM, and we included one statement in the Section of “**Excitation Quality Factor**”

“However, it is worth mentioning that the conclusion regarding the enhancement factor is not affected by this circumstance since it is compensated by comparing the signal between the structured region and the unstructured film.”

We have considered the excitation size in our previous research on BICs and found that the influence of the excitation size is typically negligible compared to other fundamental factors, even when using a large numerical aperture. Wavevector matching would be an advantage, as the reviewer correctly suggests since it allows the Q factor to scale quickly. However, a tradeoff must be considered since most mechanisms involving BIC (lasing, high harmonic generation, etc.) require a large photon density for which focusing is mandatory. Only an extremely powerful laser would guarantee such a high photon density without focusing, surely not a tunable laser. Focusing loses energy in other modes, but the local density of photons at the right wavevector still increases by orders of magnitude.

Referee #2 (Remarks to the Author):

The authors have provided comprehensive responses to all the questions raised. In my opinion, considering the quality of their answers, the paper is suitable for acceptance in the journal Nature. I also have some additional comments, which are as follows:

Error bars should be defined in all figures, and the authors should include a sentence describing how it measures it. For example, Fig 2(d), Fig (2c), fig 4(d)

Additional comments

Figure 2 lambda-in is unclear to me and should be better illustrated.

We thank again the reviewer for the thoughtful comments and positive feedback on our revisions. We greatly appreciate the reviewer’s suggestions for improving our work.

We have included in the revised manuscript definition of error bars (standard deviation of intensity) and lambda-in (input source tuned to the spectral position of the BIC).

Referee #3 (Remarks to the Author):

In the revised iteration, the authors have diligently addressed all the suggestions provided, leading to a marked enhancement in the overall quality of the manuscript. I am impressed by the incorporation of novel data in Figure 4, encompassing Iso-frequency far-field and near-field intensity mappings, alongside the demonstration of collimated emissions spanning diverse visible ranges on a centimeter scale. Following a comprehensive review of the revised manuscript, I agree with its publication in Nature.

We would like to express our gratitude for the time the reviewer invested in the review of our manuscript. We appreciate the reviewer's acknowledgement of the significance of our approach for enhancing upconversion and the novelty of directional control of light emission.

Reviewer Reports on the Second Revision:

Referees' comments:

Referee #1 (Remarks to the Author):

The authors answered all my questions in the revised version of the paper. I can recommend the paper for publication in Nature in the current form